## OPEN

# Polygenic prediction of educational attainment within and between families from genome-wide association analyses in 3 million individuals

Aysu Okbay[1,197,198 ✉], Yeda Wu[2], Nancy Wang[3], Hariharan Jayashankar[3], Michael Bennett[3], Seyed Moeen Nehzati[4], Julia Sidorenko[2], Hyeokmoon Kweon[1], Grant Goldman[3], Tamara Gjorgjieva[3], Yunxuan Jiang[5], Barry Hicks[5], Chao Tian[5], David A. Hinds[5], Rafael Ahlskog[6], Patrik K. E. Magnusson[7], Sven Oskarsson[6], Caroline Hayward[8], Archie Campbell[9,10], David J. Porteous[9,10,11], Jeremy Freese[12], Pamela Herd[13], 23andMe Research Team*, Social Science Genetic Association Consortium*, Chelsea Watson[4], Jonathan Jala[4], Dalton Conley[14], Philipp D. Koellinger[1,15], Magnus Johannesson[16], David Laibson[17], Michelle N. Meyer[18], James J. Lee[19], Augustine Kong[20], Loic Yengo[2,198], David Cesarini[3,21,22,198], Patrick Turley[23,24,198], Peter M. Visscher[2,198 ✉], Jonathan P. Beauchamp[25,198], Daniel J. Benjamin[3,4,26,198 ✉] and Alexander I. Young[4,26,197,198 ✉]

We conduct a genome-wide association study (GWAS) of educational attainment (EA) in a sample of ~3 million individuals and identify 3,952 approximately uncorrelated genome-wide-significant single-nucleotide polymorphisms (SNPs). A genome-wide polygenic predictor, or polygenic index (PGI), explains 12–16% of EA variance and contributes to risk prediction for ten diseases. Direct effects (i.e., controlling for parental PGIs) explain roughly half the PGI's magnitude of association with EA and other phenotypes. The correlation between mate-pair PGIs is far too large to be consistent with phenotypic assortment alone, implying additional assortment on PGI-associated factors. In an additional GWAS of dominance deviations from the additive model, we identify no genome-wide-significant SNPs, and a separate X-chromosome additive GWAS identifies 57.

E A is an important dimension of socioeconomic status that features prominently in research by social scientists, epidemiologists and other medical researchers. EA is strongly related to a range of health behaviors and outcomes, including mortality[1]. For this reason, and because EA can be measured accurately at low cost, cohort studies used in genetic epidemiology and medical research routinely measure participants' EA.

The most recent GWAS meta-analysis of EA had a combined sample size of ~1.1 million individuals[2]. Here we report and analyze

results from an updated meta-analysis of EA in a combined sample nearly three times larger ($N = 3,037,499$). The increase comes from expanding the sample for the association analyses from 23andMe from ~365,000 to ~2.3 million genotyped research participants. As before, our core analysis is a GWAS of autosomal SNPs. Our updated meta-analysis identifies 3,952 approximately uncorrelated SNPs at genome-wide significance compared to 1,271 in the previous study. The larger sample size yields more accurate effect-size estimates that allow us to construct a genome-wide PGI (also called

[1]Department of Economics, School of Business and Economics, Vrije Universiteit Amsterdam, Amsterdam, the Netherlands. [2]Institute for Molecular Bioscience, University of Queensland, Brisbane, QLD, Australia. [3]National Bureau of Economic Research, Cambridge, MA, USA. [4]UCLA Anderson School of Management, Los Angeles, CA, USA. [5]23andMe, Inc., Sunnyvale, CA, USA. [6]Department of Government, Uppsala University, Uppsala, Sweden. [7]Swedish Twin Registry, Department of Medical Epidemiology and Biostatistics, Karolinska Institutet, Stockholm, Sweden. [8]MRC Human Genetics Unit, Institute of Genetics and Cancer, University of Edinburgh, Western General Hospital, Edinburgh, UK. [9]Centre for Genomic and Experimental Medicine, Institute of Genetics and Cancer, University of Edinburgh, Western General Hospital, Edinburgh, UK. [10]Usher Institute, University of Edinburgh, Edinburgh, UK. [11]Centre for Cognitive Ageing and Cognitive Epidemiology, University of Edinburgh, Edinburgh, UK. [12]Department of Sociology, Stanford University, Stanford, CA, USA. [13]McCourt School of Public Policy, Georgetown University, Washington, DC, USA. [14]Department of Sociology, Princeton University, Princeton, NJ, USA. [15]Robert M. La Follette School of Public Affairs, University of Wisconsin-Madison, Madison, WI, USA. [16]Department of Economics, Stockholm School of Economics, Stockholm, Sweden. [17]Department of Economics, Harvard University, Cambridge, MA, USA. [18]Center for Translational Bioethics and Health Care Policy, Geisinger Health System, Danville, PA, USA. [19]Department of Psychology, University of Minnesota Twin Cities, Minneapolis, MN, USA. [20]Big Data Institute, Li Ka Shing Centre for Health Information and Discovery, University of Oxford, Oxford, UK. [21]Department of Economics, New York University, New York, NY, USA. [22]Center for Experimental Social Science, New York University, New York, NY, USA. [23]Department of Economics, University of Southern California, Los Angeles, CA, USA. [24]Center for Economic and Social Research, University of Southern California, Los Angeles, CA, USA. [25]Interdisciplinary Center for Economic Science and Department of Economics, George Mason University, Fairfax, VA, USA. [26]Human Genetics Department, UCLA David Geffen School of Medicine, Los Angeles, CA, USA. [197]These authors contributed equally: Aysu Okbay, Alexander I. Young. [198]These authors jointly supervised this work: Aysu Okbay, Loic Yengo, David Cesarini, Patrick Turley, Peter M. Visscher, Jonathan P. Beauchamp, Daniel J. Benjamin, Alexander I. Young. *Lists of authors and their affiliations appear at the end of the paper. ✉e-mail: daniel.benjamin@gmail.com; a.okbay@vu.nl; peter.visscher@uq.edu.au; alextisyoung@gmail.com

**Table 1 | Comparison of previous large-scale GWASs of EA**

| | Additive GWAS, autosomes | | | | | | Additive GWAS, X chromosome | | | | | | | Dominance GWAS, autosomes | | | |
| | | SNPs | | | PGI R² | | | SNPs | | | PGI R² (C + T, P <1) | | | | SNPs | | |
| | N | No. of SNPs | No. of loci | Mean χ² | LDpred, HapMap3 SNPs | C + T, P < 5 × 10⁻⁸ | N | No. of SNPs | No. of loci | Mean χ² | Male | Female | Pooled | N | No. of SNPs | No. of loci | Mean χ² |
|---|---|---|---|---|---|---|---|---|---|---|---|---|---|---|---|---|---|
| EA1 | 126,559 | 2,310,444 | 4 | 1.24 | 2.64% | 0.03% | - | - | - | - | - | - | - | - | - | - | - |
| EA2-D | 293,723 | 9,256,490 | 74 | 1.46 | 5.81% | 0.46% | - | - | - | - | - | - | - | - | - | - | - |
| EA2-C | 405,072 | 9,918,450 | 162 | 1.63 | 6.91% | 0.93% | - | - | - | - | - | - | - | - | - | - | - |
| EA3 | 1,131,881 | 10,016,266 | 1,271 | 2.91 | 10.09% | 4.03% | 694,894 | 205,865 | 10 | 2.60 | 0.04% | 0.00% | 0.01% | - | - | - | - |
| EA4 | 3,037,499 | 10,675,380 | 3,952 | 4.90 | 13.28% | 7.18% | 2,713,033 | 211,581 | 57 | 5.24 | 0.29% | 0.10% | 0.19% | 2,574,253 | 5,870,596 | 0 | 1.00 |

Summary overview of GWASs meta-analyses of educational attainment. No. of SNPs is the number of markers included in the final GWAS meta-analysis of number of years of schooling completed; no. of Loci is the number of approximately independent SNPs that reached genome-wide significance; and mean χ² is the average test statistic for SNPs with MAF >1% and N >0.9 × N_max, where N_max is the maximum sample size across all SNPs. PGIs are generated using SNPs available in all GWASs (all five GWASs for autosomal PGI and EA3-EA4 for the X chromosome PGI) and uniform procedures described in the Supplementary Note. C + T stands for clumping and thresholding. The autosomal PGI R² values are sample-size weighted averages of the incremental R² values from the Health and Retirement Study and the National Longitudinal Study of Adolescent to Adult Health. The X chromosome PGI R² values are the incremental R² values from the Health and Retirement Study. The incremental R² is the increase in R² after adding the PGI to a regression of EA on controls (a full set of dummy variables for year of birth, an indicator variable for sex, a full set of interactions between sex and year of birth and the first ten principal components of the genomic relatedness matrix). EA1, Rietveld et al.[6¹] combined meta-analysis of discovery and replication cohorts; EA2-D, Okbay et al.[6²] meta-analysis of discovery and replication cohorts; EA2-C, Okbay et al.[6²] meta-analysis of discovery cohorts; EA3, Lee et al.[²] meta-analysis of discovery cohorts; EA4, current study.

a polygenic score) that has greater prediction accuracy, increasing the percentage of variance in EA explained from 11–13% to 12–16%, depending on the validation sample, an increase of approximately 20%. In meta-analyses of the expanded 23andMe sample and the UK Biobank (UKB)[3], we also conduct an updated GWAS of the X chromosome ($N = 2,713,033$) and the first large-scale 'dominance GWAS' (i.e., a SNP-level GWAS of dominance deviations) of EA on the autosomes ($N = 2,574,253$). In our updated X-chromosome GWAS, we increase the number of approximately uncorrelated genome-wide-significant SNPs from 10 to 57. Our dominance GWAS identifies no genome-wide-significant SNPs. Moreover, with high confidence, we can rule out the existence of any common SNPs whose dominance effects explain more than a negligible fraction of the variance in EA. Table 1 summarizes the GWASs conducted in this paper and compares them to previous large-scale GWASs of educational attainment.

The rest of the paper investigates the scope and sources of the PGI's predictive power. We first document that the EA PGI not only predicts a range of cognitive phenotypes, as has been found in previous work[2,4], but also adds nontrivial predictive power for ten diseases we examine, even after controlling for disease-specific PGIs. Next, using a combined sample of ~53,000 individuals with genotyped siblings and ~3,500 individuals with both parents genotyped, we examine the predictive power of the EA PGI controlling for parental EA PGIs. By controlling for parental EA PGIs, we isolate the component of predictive power that is due to direct effects[5], or the causal effects of an individual's genetic material on that individual[6]. For EA and 22 other phenotypes, controlling for the parental EA PGIs roughly halves the EA PGI's association with the phenotype. In contrast, when we examine PGIs for height, body mass index (BMI) and cognitive performance, controlling for parental PGIs has far less impact on their associations with their corresponding phenotype. Thus, the EA PGI stands out as unusual in terms of how much of its predictive power is not due to direct effects.

Finally, we use PGIs to study assortative mating. Using 862 genotyped mate pairs in the UKB and 1,603 pairs in Generation Scotland (GS)[7], we estimate the correlation between mate-pair PGIs for EA, as well as for height. For height, the correlation between mate-pair PGIs is close to that expected under phenotypic assortment (that is, all similarity between mate pairs on the genetic component of the phenotype arises via matching on the phenotype). Once again, EA is different; the correlation between mate-pair PGIs for EA is much larger than one would expect from phenotypic assortment on EA. We find evidence that population structure captured by principal components (PCs) and assortment on cognitive performance explain some, but not all, of the excess mate-pair PGI correlation. These findings shed further light on the EA PGI's predictive power for EA and other phenotypes; the factors on which mate pairs assort that are not EA but are correlated with the EA PGI (e.g., geographic location at courtship age (we speculate)) likely also contribute to the PGI's predictive power.

For a less technical description of the paper and of how it should—and should not—be interpreted, see the frequently asked questions in Supplementary Data 1.

## Results

**Additive GWAS of EduYears in autosomes.** We conducted a sample-size-weighted meta-analysis of association results on EA, measured as number of years of schooling completed (EduYears), by combining three sets of summary statistics: public results from our previous meta-analysis of 69 cohorts ($N = 324,162$, excluding UKB and 23andMe), new association results from 23andMe ($N = 2,272,216$) and new association results from a GWAS we conducted in UKB with an improved coding of the EA measure ($N = 441,121$; Supplementary Note). All analyses were conducted in samples of European genetic ancestries, included controls for

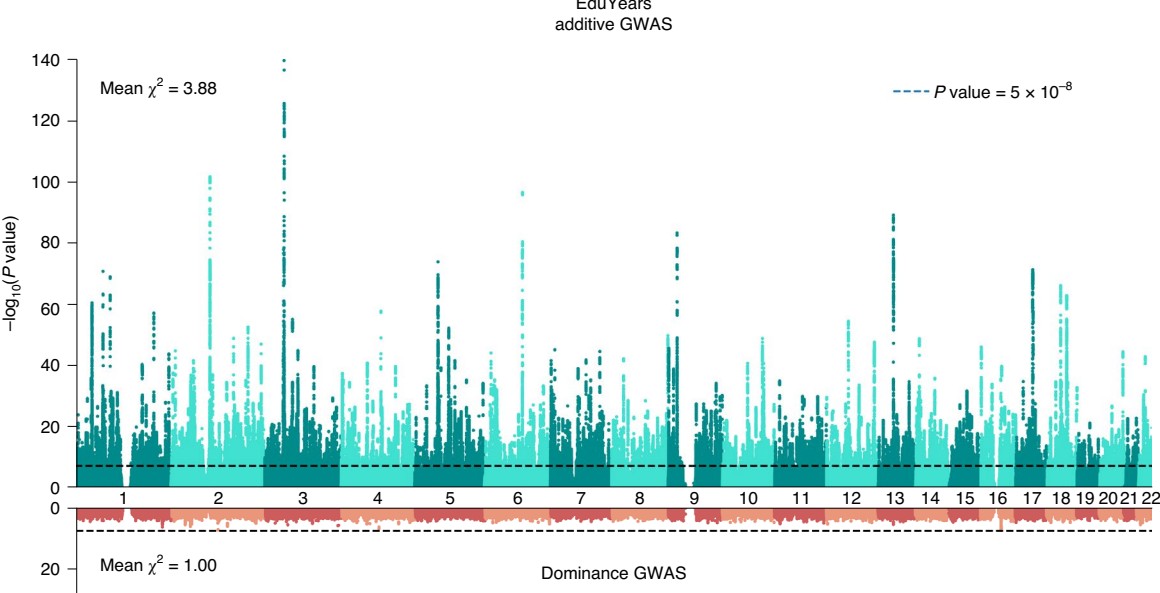

**Fig. 1 | Manhattan plots for the additive and dominance GWASs.** The top graph (green) shows the additive GWAS ($N = 3,037,499$ individuals), and the bottom graph (red) shows the dominance GWAS ($N = 2,574,253$ individuals). The $P$ value and mean $\chi^2$ values are based on inflation-adjusted two-sided $Z$ tests. The $x$ axis is chromosomal position, and the $y$ axis is the significance on a $-\log_{10}$ scale. The dashed line marks the threshold for genome-wide significance ($P = 5 \times 10^{-8}$).

sex, year of birth, their interaction and genetic PCs, and applied a uniform set of quality-control procedures (Supplementary Note contains a comprehensive description). The final meta-analysis contains association results for ~10 million SNPs. The quantile–quantile plot in Extended Data Fig. 1 shows that the $P$ values deviate strongly from the uniform distribution. According to the linkage disequilibrium (LD) score regression[8] intercept (1.66), confounding accounts for 7% of the inflation, similar to previous GWAS of EA (ref. [2]) (Extended Data Fig. 2 shows the LD score plot). The Manhattan plot in Fig. 1 and many of our subsequent analyses are based on test statistics adjusted for the LD score intercept.

We identify 3,952 lead SNPs, defined as approximately uncorrelated (pairwise $r^2 < 0.1$) variants with an association $P$ value below $5 \times 10^{-8}$. At the stricter threshold[9] of $P < 1 \times 10^{-8}$, the number declines to 3,277 (Supplementary Table 1; Supplementary Note contains a description of the clumping algorithm). To assess the sensitivity of our conclusions about the number of independent SNPs, we conducted a conditional and joint (COJO) multiple-SNP analysis[10]. This analysis identified 2,925 SNPs (Supplementary Table 2); 41 of these are in LD ($r^2 > 0.1$) with other COJO lead SNPs and may represent secondary associations within a locus. Adjusted for the winner's curse, we find that the effects of our lead SNPs are consistently quite small. On average, an additional copy of the reference allele of the median SNP is associated with 1.4 weeks more schooling: the effects at the 5th and 95th percentiles (in absolute value) are 0.9 and 3.5 weeks, respectively (Supplementary Note contains details on these calculations). We also examined the out-of-sample replicability of the lead SNPs identified in the most recent previous meta-analysis[2]. In the independent 23andMe data, the replication record is broadly in line with theoretical predictions derived from an empirical Bayesian framework described in the Supplementary Note (Extended Data Fig. 3).

**Biological annotation.** To compare results from biological annotation of our meta-analysis to that of the most recent previous meta-analysis, we applied stratified LD score regression[11] to both sets of summary statistics using a recent set of SNP annotations[12]. The results are very similar across the two meta-analyses, but standard errors are smaller when using the current meta-analysis

results, as expected given the larger sample size (Supplementary Fig. 1a–d). Notably, we replicate the unexpected result of relatively weak enrichment of genes highly expressed in glial cells (astrocytes and oligodendrocytes) relative to neurons.

**X-chromosome GWAS results.** To update the previous X-chromosome analysis, we conducted a sample-size-weighted meta-analysis of mixed-sex association results from UKB and 23andMe ($N = 2,713,033$) for ~200,000 SNPs on the X chromosome (Extended Data Fig. 4). We identified 57 lead SNPs with estimated effects in the range 1 to 3 weeks of schooling. Our findings are fully consistent with earlier conclusions: SNP heritability due to the X chromosome of 0.4% and (using sex-stratified association analyses in the UKB) a male–female genetic correlation on the X chromosome close to unity ($r_g = 0.94$, s.e. $= 0.03$).

**Dominance GWAS.** We conducted a GWAS of dominance deviations from the additive model (Supplementary Note) by meta-analyzing summary statistics from association analyses conducted in 23andMe and UKB ($N = 2,574,253$). Theory and evidence from the quantitative genetics literature, including findings from two recent papers[13,14] that estimated dominance SNP heritability across dozens of phenotypes (but not EA), suggest that dominance effects explain at most a very small share of the variance in polygenic phenotypes[15]. Nevertheless, in the behavior genetics literature, when the phenotypic correlation between monozygotic twins is more than twice as large as the phenotypic correlation between dizygotic twins, it remains common practice to attribute the violation of the additive model to dominance variance.

The Manhattan plot from our dominance GWAS is shown in red in the bottom panel of Fig. 1. There are no genome-wide-significant SNPs. Power calculations indicate that, at genome-wide significance, we had 80% power to detect dominance effects with an $R^2$ of 0.0015% (Supplementary Note). Such effect sizes would be over an order of magnitude smaller than the largest additive effects ($R^2 \cong 0.04\%$). Therefore, the absence of genome-wide-significant SNPs suggests that dominance effects of common SNPs, taken individually, are negligibly small.

Next, we turn to the combined dominance effects of common SNPs. Applying an adapted version of LD Score regression to the summary statistics, we estimate a SNP heritability of 0.00015 (s.e. = 0.00024), which is statistically indistinguishable from zero ($P = 0.54$). In the Supplementary Note, we report additional analyses (that rely on different assumptions) that similarly conclude that the combined variance explained by dominance deviations in common SNPs is negligible. Our results do not rule out the possibility that rare SNPs have substantial dominance effects.

Even when the phenotypic variance across individuals explained by dominance is negligible, the combined dominance effects on an individual can be substantial when homozygosity (which is deleterious on average) is increased genome-wide due to inbreeding[16]. This reduction of fitness-related phenotypic values is called directional dominance, or inbreeding depression (ID). We applied a recently developed method that uses dominance GWAS summary statistics to estimate ID[17]. Our estimate implies the offspring of first cousins have on average ~1.0 fewer months of EA ($P = 0.04$) than the offspring of unrelated individuals.

**Polygenic prediction.** We assessed empirically how well a PGI derived from the autosomal GWAS of additive variation predicts a host of phenotypes related to EA, academic achievement and cognition. We used three European genetic-ancestry holdout samples from the National Longitudinal Study of Adolescent to Adult Health (Add Health)[18], a representative sample of American adolescents followed into adulthood; the Health and Retirement Study (HRS)[19], a representative sample of Americans over age 50 years; and the Wisconsin Longitudinal Study (WLS)[20], a sample of individuals who graduated from high school in Wisconsin in 1957. Because of the range restriction for EduYears in WLS, we do not use it to evaluate predictive power for EA. Our measure of prediction accuracy is the 'incremental $R^2$', or the gain in coefficient of determination ($R^2$) when the PGI is added as a covariate to a regression of the phenotype on a set of baseline controls (sex, dummy variables for birth year and/or age at assessment, their interactions and ten PCs of the genomic relatedness matrix). All PGIs that we analyze are based on a meta-analysis that excluded Add Health, HRS and WLS.

A PGI constructed using only genome-wide-significant SNPs has an incremental $R^2$ of 9.1% in Add Health and 7.0% in HRS (Extended Data Fig. 5). For all PGI analyses hereafter, unless stated otherwise, we use a PGI generated from HapMap3 SNPs using the software LDpred (ref. [21]). This PGI explains 15.8% of the variance in EduYears in Add Health and 12.0% in HRS (Extended Data Fig. 6). The sample-size-weighted mean is 13.3%. Fig. 2a depicts how the predictive power has increased as GWAS sample sizes have increased. Fig. 2b shows that the prevalence of college completion varies a great deal over PGI deciles (Extended Data Fig. 7a,b shows prevalences of high school completion and grade retention). For example, only 7.3% and 6.8% of individuals in the lowest PGI decile have a college degree in Add Health and HRS, respectively, compared to 70.7% and 53.0% in the highest PGI decile. Fig. 2c, which displays scatterplots of individual EA versus PGIs, shows that throughout the PGI distribution, there is substantial variation in EA at the individual level. Thus, although average EA varies substantially across the PGI distribution, the PGI cannot be used to meaningfully predict an individual's EA.

In post hoc analyses, we found that a PGI generated from ~2.5 million pruned common SNPs using the software SBayesR (ref. [22]) is more predictive than our LDpred PGI. It explains 17.0% of the variance in EduYears in Add Health and 12.9% in HRS, with a sample-size-weighted mean of 14.3% (Supplementary Table 3).

We supplemented our analyses of education outcomes with other cognitive and academic achievement outcomes (Extended Data Fig. 6 and Supplementary Table 4). For example, in Add Health, we found that the PGI explains 8.7% of the variation in Peabody verbal test scores and 12.3% in overall grade point average. In WLS, the PGI explains 6.1% of the variation in Henmon–Nelson test scores and 7.7% in high-school-grade percentile rank.

PGIs like ours that are constructed from GWAS in samples of European genetic ancestries are generally found to have much lower predictive power in samples with other genetic ancestries; for example, on average across phenotypes, estimates of relative accuracy (ratio of $R^2$) in African-genetic-ancestry to European-genetic-ancestry samples have been 22% (ref. [23]) and 36% (ref. [24]). When we used our PGI to predict EduYears in samples with African genetic ancestries from the HRS ($N = 2,507$) and Add Health ($N = 1,716$), the incremental $R^2$ was 1.3% (95% confidence interval (CI), 0.6% to 2.2%) and 2.3% (95% CI, 1.1% to 3.7%), implying that the relative accuracies for EA in the HRS and Add Health are only 11% and 15%, respectively. Using the UKB, we find that the relative accuracy is smaller than would be predicted based on population differences in allele frequencies and LD alone (Online Methods), and this discrepancy is greater for EA than has been found in prior work[25] for height, BMI and six other phenotypes (Extended Data Fig. 8 and Supplementary Table 5). The remaining reduction in predictive power is due to factors including epistasis (although epistatic variance is likely small[13,15]), gene–environment interactions and differences between populations in gene–environment correlations, assortative mating and environmental variance.

**Predicting disease risk.** Among individuals of European genetic ancestries in the UKB, we estimated the predictive power of the EA PGI for ten common diseases for which large-scale GWASs have been conducted (Fig. 3). Because disease status is dichotomous, we assess predictive power using Nagelkerke's coefficient of determination[26]. Consistent with prior work that has estimated nonzero genetic correlations between EA and many diseases and health-related phenotypes[27], some using an earlier EA PGI[1,28,29], our EA PGI significantly predicts all ten diseases (all ten $P$ values are smaller than $3 \times 10^{-8}$; Supplementary Table 6). The mean incremental $R^2$ across all ten diseases is 0.63%. This predictive power is nontrivial compared with the average incremental $R^2$ of 1.19% for disease-specific PGIs constructed using summary statistics from large-scale GWASs of the diseases. Moreover, the EA and disease-specific PGIs contribute roughly independently to predicting disease risk; the incremental $R^2$ from adding both PGIs and their interaction to the regression model is typically roughly equal to the sum of the incremental $R^2$ values of each of the two PGIs considered separately. Higher values of the EA PGI correspond to lower relative risk for each of the ten diseases (Extended Data Fig. 9 and Supplementary Tables 7 and 8).

**Within-family analyses.** Our next set of analyses, like related prior work[5,30,31], aimed to isolate the component of the PGI's predictive

---

**Fig. 2 | Polygenic prediction. a**, Predictive power of the EA PGI as a function of the size of the GWAS discovery sample, with expected predictive power shown by the dashed lines (Supplementary Note section 5.5). **b**, Prevalence of college completion by EA PGI decile, with 95% CIs. **c**, Scatterplot of EA PGI (residualized on ten principal components) and EduYears (residualized on sex, a full set of birth-year dummies, their interactions and ten principal components). Prediction samples for all panels are European-ancestry participants in Add Health ($N = 5,653$) and the HRS ($N = 10,843$). All PGIs were constructed from EduYears GWAS results that exclude Add Health and HRS using the software LDpred and assuming a normal prior for SNP effect sizes. Incremental $R^2$ is the difference between the $R^2$ from a regression of EduYears on the PGI and the controls (sex, a full set of birth-year dummies, their interactions and ten principal components) and the $R^2$ from a regression of EduYears on just the controls. The individual-level data plotted in **c** have been jittered by adding a small amount of noise to each observation.

power that is due to direct effects[5,6], or causal effects of an individual's genetic material on that individual. When controls for both parents' PGIs are included, we refer to the coefficient from a regression of an individual's phenotype on the individual's PGI as the direct effect of the PGI; when those controls are omitted, we refer to it as the population effect. (The regression controlling for parental

PGIs gives an equivalent estimate of the direct effect of the PGI as a regression on PGIs constructed from transmitted and nontransmitted parental alleles[5]; Supplementary Note.) The population effect captures the sum of the direct effect, indirect effects from relatives (e.g., genetic influences on parents' education, socioeconomic status and behavior), other gene–environment correlation (i.e., correlation

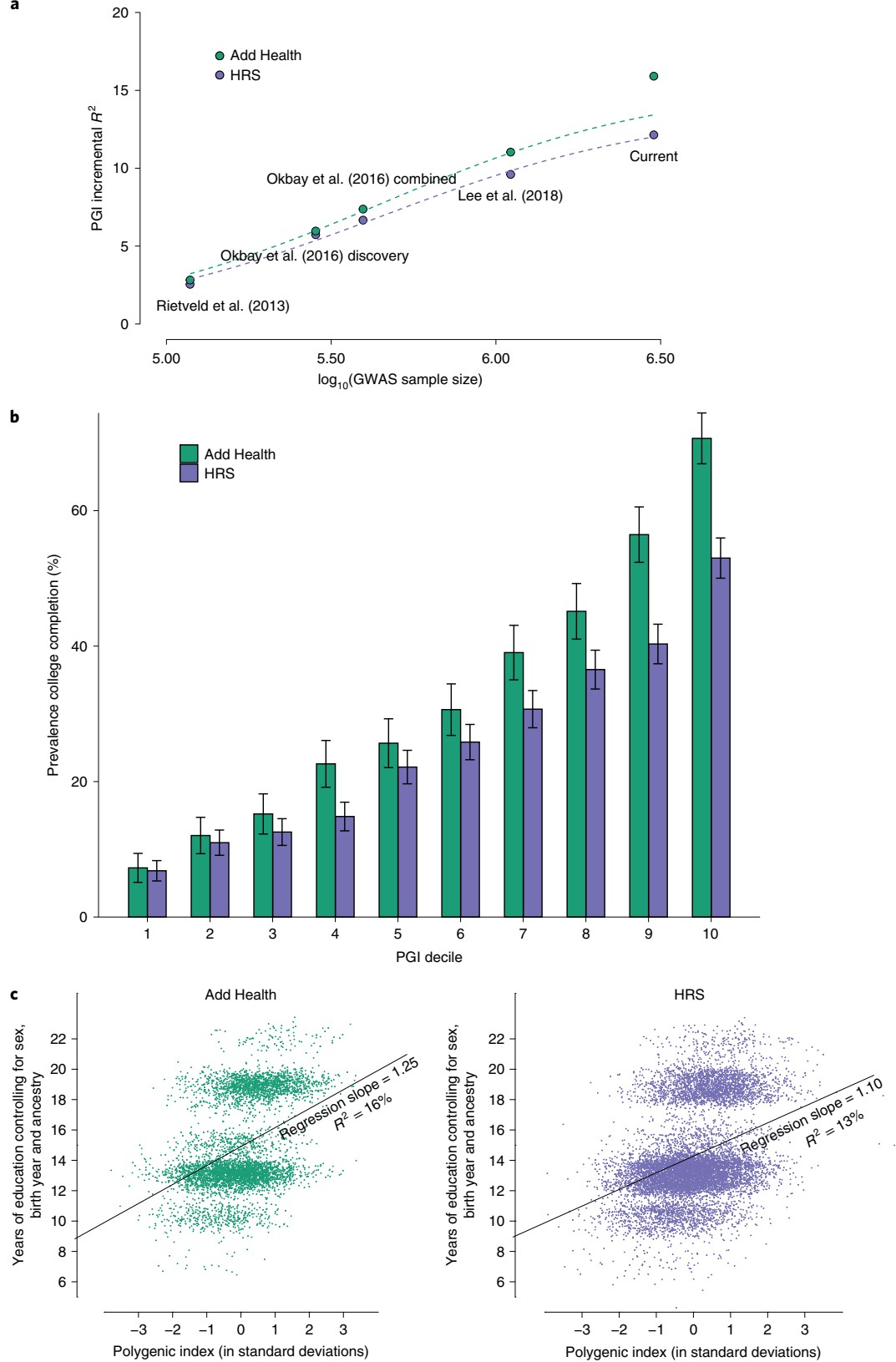

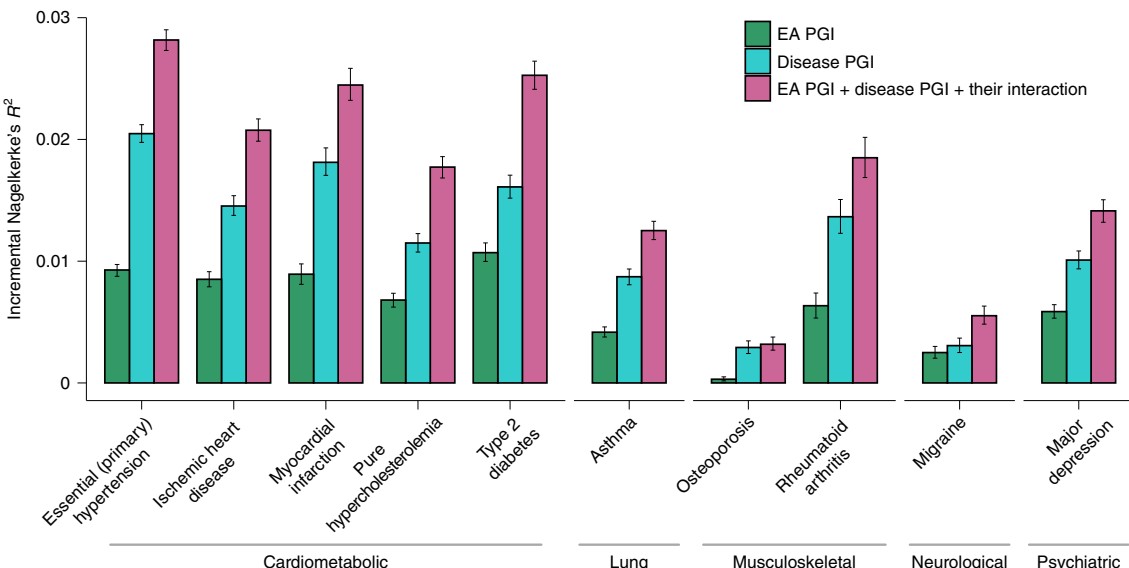

**Fig. 3 | Predictive power of the EA PGI and the disease-specific PGI and their combination for ten diseases in the UKB.** For each disease phenotype, the figure shows the incremental Nagelkerke's $R^2$ from adding the EA PGI, the disease PGI or both PGIs and their interaction to a logistic regression of the disease phenotype on covariates. The covariates are sex, a third-degree polynomial in birth year and their interactions with sex, the first 40 PCs and batch dummies. The error bars represent 95% CIs calculated with the bootstrap percentile method, with 1,000 repetitions.

between genotypes and environmental exposure, with population stratification being one possible cause) and a contribution from the genetic component of the phenotype that would be uncorrelated with the PGI under random mating but becomes correlated with the PGI due to the LD between causal alleles induced by assortative mating (Supplementary Note)[5,32]. Because the PGI is constructed from summary statistics that partly reflect indirect effects and other gene–environment correlation, estimating the direct effect of the PGI is different from estimating the total contribution of direct effects of SNPs[33,34], for which relatedness disequilibrium regression[35] or summary statistics from within-family GWAS[36] could be used.

For this analysis, we used a combined sample of ~53,000 individuals with genotyped siblings and ~3,500 individuals with both parents genotyped (Online Methods and Supplementary Note). Direct-effect estimates from the sibling data may be biased by sibling indirect effects, but estimates of such effects are small, including for some of the phenotypes we study[37]. The data are from the UKB (ref. [3]), GS (ref. [7]) and the Swedish Twin Registry (STR)[38]. We did not have sufficient power to study the diseases from Fig. 3 when restricting to these family samples. We instead analyze a set of 23 health, cognitive and socioeconomic phenotypes, which include cardiometabolic and lung biomarkers related to disease risk (Supplementary Tables 9 and 10).

Fig. 4a (and Supplementary Table 10) shows our meta-analysis estimates of the direct and population effects of the EA PGI. For predicting EA, the ratio of direct to population effect estimates is 0.556 (s.e. = 0.020), implying that $100\% \times 0.556^2 = 30.9\%$ of the PGI's $R^2$ is due to its direct effect. This is smaller than the estimate of 48.9% reported in a previous analysis of Icelandic data[5]. For comparison with EA, we similarly estimate the direct and population effects of PGIs for height, BMI and cognitive performance on their respective phenotypes (Fig. 4a). The ratio of direct to population effect estimates is 0.910 (s.e. = 0.009) for height, 0.962 (s.e. = 0.017) for BMI and 0.824 (s.e. = 0.033) for cognitive performance, implying that 82.8%, 92.5% and 67.9%, respectively, of the PGI $R^2$ values are due to their direct effects (Supplementary Tables 11–13). The EA PGI has by far the lowest ratio.

We similarly assessed how much of the EA PGI's predictive power for the other 22 phenotypes (other than EA) is due to direct effects.

Fig. 4b shows estimates of the population and direct effects of the EA PGI. Across the phenotypes, the inverse-variance-weighted average ratio of direct to population effects is 0.588 (s.e. = 0.013). This is similar to the ratio of 0.556 for the EA PGI on EA. Thus, both for predicting EA and other phenotypes, a substantial part of the EA PGI's predictive power results from direct effects, but a substantial part results from factors other than direct effects. (For analogous analyses with the PGIs for height, BMI and cognitive performance, see Supplementary Fig. 2a–c, Supplementary Tables 11–13 and Supplementary Note.)

**Assortative mating.** We also use the PGI to study assortative mating. For this analysis, we use data on genotyped mate pairs in the UKB (862 pairs) and GS (1,603 pairs). Under the (commonly assumed) hypothesis of phenotypic assortment—according to which the mate-pair genetic components are independent conditional on the mate-pair phenotypes[39,40]—the mate-pair PGI correlation should equal the product of the mate-pair phenotypic correlation, the correlation between the father's phenotype and PGI and the correlation between the mother's phenotype and PGI. We examined whether correlations between mate-pair EA PGIs fit this model (Fig. 5a), and we performed the same analysis for the height PGI (Fig. 5b). Height provides a useful comparison, because its mate-pair phenotypic correlation (0.290, s.e. = 0.018) and mate-pair PGI correlation (0.106, s.e. = 0.020) are somewhat similar to EA's mate-pair phenotypic correlation (0.430, s.e. = 0.017) and mate-pair PGI correlation (0.175, s.e. = 0.020). (For completeness, Supplementary Table 14 also shows results for the BMI and cognitive performance PGIs, but these are less informative because the mate-pair PGI correlations are not statistically distinguishable from zero.)

For height, phenotypic assortment predicts a mate-pair PGI correlation of 0.087 (s.e. = 0.007) (the gray point in the figure), which is only somewhat smaller than the observed estimate of 0.106 and is contained within the 95% CI. In contrast, for EA, the predicted value of 0.031 (s.e. = 0.004) is much smaller than, and statistically distinguishable from, the mate-pair PGI correlation of 0.175. Phenotypic assortment on EA would also imply that after residualizing the PGI on EA, the mate-pair PGI correlation should fall to zero. In fact, the correlation falls by only 37%, to 0.110 (s.e. = 0.021).

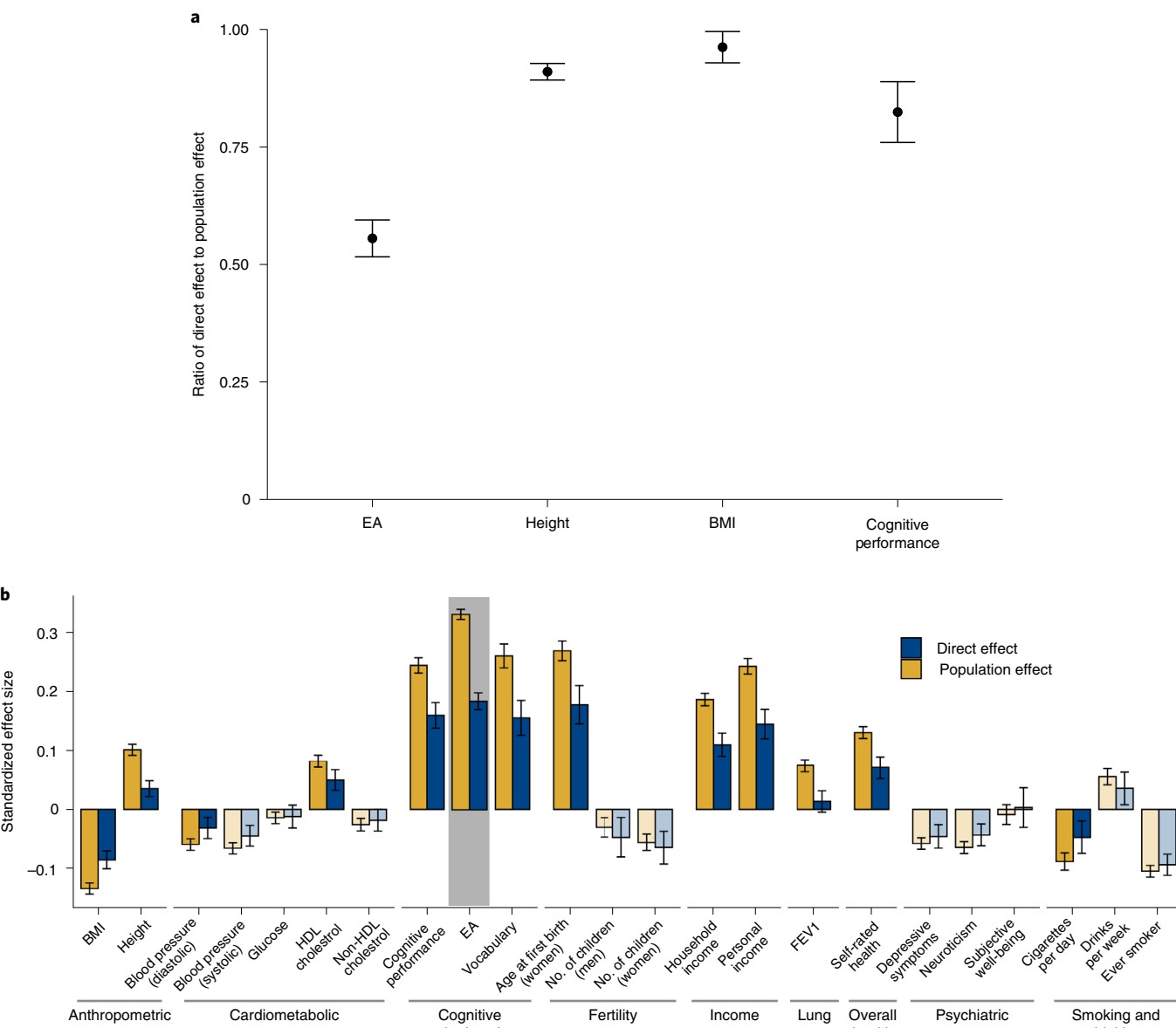

**Fig. 4 | Meta-analysis estimates of direct and population effects of PGIs. a**, For each PGI, the ratio of the direct effect to the population effect on the phenotype from which the PGI was derived. **b**, The effects of the EA PGI on 23 phenotypes. Bars are shaded lighter when the population and direct effects are statistically indistinguishable (two-sided Z test P > 0.05/23, where 23 is the number of phenotypes under study). For both panels, estimates are from meta-analyses of UKB, GS, and STR samples of siblings and trios. Phenotypes and the PGIs are scaled to have variance one, so effects correspond to partial correlation coefficients. Error bars represent 95% CIs. See Supplementary Table 9 for details on phenotypes and Supplementary Tables 10–13 for numerical values underlying this figure. FEV1, forced expiratory volume during the first second; HDL, high-density lipoprotein.

We explore two plausible explanations of the high mate-pair EA PGI correlation. The first is mate pairs tending to share genetic ancestry. Not all forms of social homogamy generate a mate-pair PGI correlation[41], but social homogamy that is related to genetic ancestry (e.g., due to geographic proximity that tracks genetic structure in the population) will do so if there are components of genetic ancestry correlated with the PGI. After residualizing the EA PGI on 40 PCs of the genomic relatedness matrix in addition to EA, we find that the mate-pair PGI correlation falls to 0.091 (s.e. = 0.021). This implies that some, but not most, of the mate-pair PGI correlation is due to assortment on genetic ancestry captured by the PCs (or some factor correlated with the PCs). In the UKB, further adjustment for birth coordinates and the center where participants were assessed (Online Methods) resulted in a slight reduction of the correlation between mate-pair PGIs (Supplementary Table 14), suggesting that

geographic factors not captured by the top 40 PCs also contribute to the high mate-pair EA PGI correlation. The second explanation is assortment on a phenotype or composite of phenotypes that is more strongly correlated with the EA PGI than EA itself. The GS cohort contains high-quality measures of cognitive performance and vocabulary, proxies for plausible candidates of such a composite. In this cohort, after residualizing on these proxies as well as on EA and 40 PCs, the mate-pair PGI correlation is 0.083 (s.e. = 0.027) compared to 0.113 (s.e. = 0.026) when residualizing on EA and PCs alone, which leaves a substantial remainder of the mate-pair PGI correlation unexplained. This remainder is due to assortment on phenotypes correlated with the EA PGI other than EA, cognitive performance and vocabulary—possibly including various personality traits[42–44]—and sources of social homogamy other than genetic ancestry captured by the top 40 PCs—possibly including

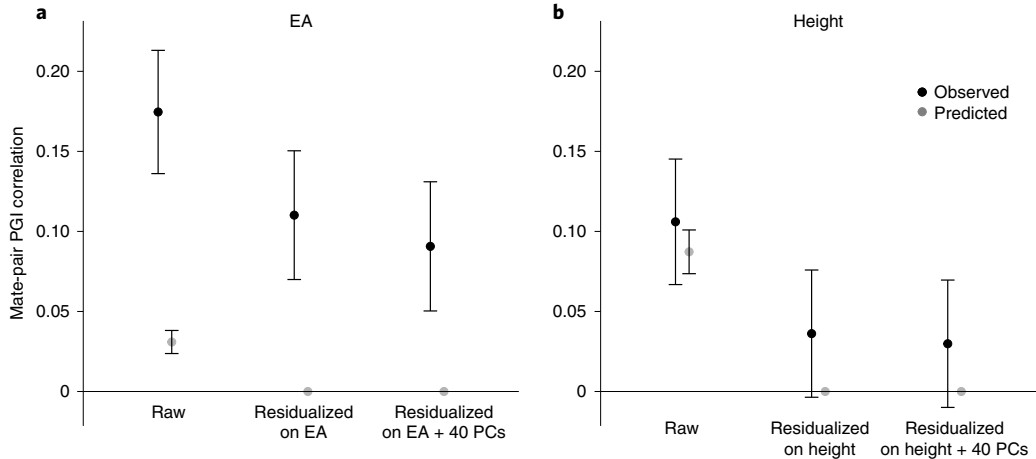

**Fig. 5 | Correlations between mate-pair PGIs. a**, Black dots show the correlation between mate-pair EA PGIs (raw) and the correlation between the residuals of the mate-pair EA PGIs after regressions with the listed regressors. Gray dots show the predicted correlations under phenotypic assortment; that is, all correlations between mate-pair EA PGIs are explained by assortment on EA itself. $N = 2,344$ (861 from UKB and 1,483 from GS). **b**, Analogous but for the height PGI and predictions under phenotypic assortment on height. $N = 2,451$ (858 from UKB and 1,593 from GS). For both panels, error bars represent 95% CIs. See Supplementary Table 14 for numerical values underlying this figure.

geographic location at courtship age[45,46], socioeconomic status and social class[47].

Any factor that contributes to explaining the mate-pair PGI correlation must be correlated with the EA PGI. Therefore, these factors likely contribute to the EA PGI's predictive power for EA and other phenotypes. Moreover, assortative mating on these factors increases the variance of the component of the EA PGI with which they are correlated, which amplifies their contribution to the EA PGI's predictive power.

## Discussion

The results of previous large-scale GWAS of EA have proven useful across many different areas of research, including medicine[48], epidemiology[49,50], psychology[42], economics[51,52] and sociology[47,53,54]. The substantial increase in power from our large sample size will make the summary statistics from the current paper even more useful. Beyond increasing power, the GWAS reported in this paper also included extensive dominance, within-family and assortative mating analyses. These analyses illustrate how, as GWAS have advanced from relatively small samples (by today's standards) that identify just a few SNPs to well-powered analyses of most of the variation from common SNPs, it has become possible to address an ever-increasing set of questions. For example, we find that the EA PGI has predictive power across a broad range of educational, cognitive and health-related phenotypes and diseases. Our results show that this predictive power derives both from direct genetic effects and from gene–environment correlation (likely including indirect genetic effects from relatives), with assortative mating amplifying the predictive power over what would be expected under random mating.

Our findings are also relevant for informing some decades-old debates in the behavior genetics literature. Because the parameters of a general biometric model cannot be separately identified from a small number of phenotypic correlations among different types of relatives, researchers typically have to assume that some of the parameters equal zero in order to estimate other parameters. In the 1970s, for example, researchers from the Birmingham School[55,56], researchers from the Hawaii School[57,58] and the sociologist Sandy Jencks famously came up with strikingly different explanations for a set of kinship correlations on cognitive test scores assembled by Jencks et al.[59]. A careful analysis by Loehlin[60] showed that the three sets of researchers arrived at different explanations for the same

data primarily due to their divergent assumptions about dominance, assortative mating, and special twin environments.

Although our results concern EA rather than cognitive test scores, we believe they are relevant for evaluating the plausibility of some of the assumptions underlying the modeling approaches that have been used to explain familial resemblance in EA and cognitive phenotypes. Three of our findings are especially relevant: (1) dominance variance due to common variants is negligible, (2) much of the predictive power of the EA PGI is not explained by direct effects and (3) the mate-pair PGI correlation is far too strong to be consistent with assortative mating purely on phenotype. Overall, these findings suggest that any model of EA that requires substantial dominance to fit the data, restricts gene–environment correlations to zero or assumes assortative mating is purely based on phenotype is likely to be misspecified. Thus, our analyses demonstrate how results from large-scale GWAS and the resulting PGIs can be used to improve the identifiability of behavior–genetic models.

The sample size of the GWAS of EA reported in this paper is the largest published to date. For some purposes, such as attaining greater predictive power for the PGI, there are clearly diminishing returns. However, even larger samples will enable other analyses that have not yet been adequately powered, such as estimating differences in SNP effect sizes across phenotypes or populations and estimating the fraction of variance explained by epistatic interactions[13].

## Online content

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

**23andMe Research Team**

Michelle Agee[5], Babak Alipanahi[5], Adam Auton[5], Robert K. Bell[5], Katarzyna Bryc[5], Sarah L. Elson[5], Pierre Fontanillas[5], Nicholas A. Furlotte[5], Barry Hicks[5], David A. Hinds[22], Karen E. Huber[5], Aaron Kleinman[5], Nadia K. Litterman[5], Jennifer C. McCreight[5], Matthew H. McIntyre[5], Joanna L. Mountain[5], Carrie A. M. Northover[5], Steven J. Pitts[5], J. Fah Sathirapongsasuti[5], Olga V. Sazonova[5], Janie F. Shelton[5], Suyash Shringarpure[5], Chao Tian[5], Joyce Y. Tung[5], Vladimir Vacic[5] and Catherine H. Wilson[5]

**Social Science Genetic Association Consortium**

Aysu Okbay[1,197,198], Jonathan P. Beauchamp[25,198], Mark Alan Fontana[24,27], James J. Lee[19], Tune H. Pers[28,29], Cornelius A. Rietveld[30,31,32], Patrick Turley[23,24,198], Guo-Bo Chen[33], Valur Emilsson[34,35], S. Fleur W. Meddens[30,36,37], Sven Oskarsson[6], Joseph K. Pickrell[38], Kevin Thom[21], Pascal Timshel[28,29], Ronald de Vlaming[30,31,32], Abdel Abdellaoui[39], Tarunveer S. Ahluwalia[28,40,41], Jonas Bacelis[42], Clemens Baumbach[43,44], Gyda Bjornsdottir[45], Johannes H. Brandsma[46], Maria Pina Concas[47], Jaime Derringer[48], Tessel E. Galesloot[49], Giorgia Girotto[50], Richa Gupta[51], Leanne M. Hall[52,53], Sarah E. Harris[53,54], Edith Hofer[55,56], Momoko Horikoshi[57,58], Jennifer E. Huffman[8], Kadri Kaasik[59], Ioanna P. Kalafati[60], Robert Karlsson[7], Augustine Kong[20], Jari Lahti[59,61], Sven J. van der Lee[56], Christiaan de Leeuw[36,62], Penelope A. Lind[63], Karl-Oskar Lindgren[6], Tian Liu[64], Massimo Mangino[65,66], Jonathan Marten[8], Evelin Mihailov[67], Michael B. Miller[19], Peter J. van der Most[68], Christopher Oldmeadow[69,70], Antony Payton[71,72], Natalia Pervjakova[67,73], Wouter J. Peyrot[74], Yong Qian[75], Olli Raitakari[76], Rico Rueedi[77,78], Erika Salvi[79], Börge Schmidt[80], Katharina E. Schraut[81], Jianxin Shi[82], Albert V. Smith[34,83], Raymond A. Poot[46], Beate St Pourcain[84,85], Alexander Teumer[86], Gudmar Thorleifsson[45], Niek Verweij[87], Dragana Vuckovic[50], Juergen Wellmann[88], Harm-Jan Westra[89,90,91], Jingyun Yang[92,93], Wei Zhao[94], Zhihong Zhu[33], Behrooz Z. Alizadeh[68,95], Najaf Amin[32], Andrew Bakshi[33], Sebastian E. Baumeister[86,96], Ginevra Biino[97], Klaus Bønnelykke[40], Patricia A. Boyle[92,98], Harry Campbell[81], Francesco P. Cappuccio[99], Gail Davies[53,100], Jan-Emmanuel De Neve[101], Panos Deloukas[102,103], Ilja Demuth[104,105], Jun Ding[75], Peter Eibich[106,107], Lewin Eisele[80], Niina Eklund[73], David M. Evans[84,108], Jessica D. Faul[109], Mary F. Feitosa[110], Andreas J. Forstner[111,112], Ilaria Gandin[50], Bjarni Gunnarsson[45], Bjarni V. Halldórsson[45,113], Tamara B. Harris[114], Andrew C. Heath[115], Lynne J. Hocking[116], Elizabeth G. Holliday[69,70], Georg Homuth[117], Michael A. Horan[118], Jouke-Jan Hottenga[39], Philip L. de Jager[91,119,120], Peter K. Joshi[81,121], Astanand Jugessur[122], Marika A. Kaakinen[123], Mika Kähönen[124,125], Stavroula Kanoni[102], Liisa Keltigangas-Järvinen[59], Lambertus A. L. M. Kiemeney[49], Ivana Kolcic[126], Seppo Koskinen[73], Aldi T. Kraja[110], Martin Kroh[106], Zoltan Kutalik[77,78,121], Antti Latvala[51], Lenore J. Launer[127], Maël P. Lebreton[37,128], Douglas F. Levinson[129], Paul Lichtenstein[7], Peter Lichtner[130], David C. M. Liewald[53,100], LifeLines Cohort Study[131]*, Anu Loukola[51], Pamela A. Madden[115], Reedik Mägi[67], Tomi Mäki-Opas[73], Riccardo E. Marioni[53,132,133], Pedro Marques-Vidal[134], Gerardus A. Meddens[135], George McMahon[84], Christa Meisinger[44], Thomas Meitinger[130], Yusplitri Milaneschi[74], Lili Milani[67], Grant W. Montgomery[136], Ronny Myhre[122], Christopher P. Nelson[52,137], Dale R. Nyholt[136,138], William E. R. Ollier[71], Aarno Palotie[91,139,140,141,142,143], Lavinia Paternoster[84], Nancy L. Pedersen[7], Katja E. Petrovic[55], David J. Porteous[9,10,11],

Katri Räikkönen[59,61], Susan M. Ring[84], Antonietta Robino[144], Olga Rostapshova[17,145], Igor Rudan[81], Aldo Rustichini[146], Veikko Salomaa[73], Alan R. Sanders[147,148], Antti-Pekka Sarin[142,149], Helena Schmidt[55,150], Rodney J. Scott[70,151], Blair H. Smith[152], Jennifer A. Smith[68], Jan A. Staessen[153,154], Elisabeth Steinhagen-Thiessen[104], Konstantin Strauch[155,156], Antonio Terracciano[157], Martin D. Tobin[158], Sheila Ulivi[144], Simona Vaccargiu[47], Lydia Quaye[65], Frank J. A. van Rooij[32,159], Cristina Venturini[65,66], Anna A. E. Vinkhuyzen[33], Uwe Völker[117], Henry Völzke[86], Judith M. Vonk[68], Diego Vozzi[144], Johannes Waage[40,41], Erin B. Ware[94,160], Gonneke Willemsen[39], John R. Attia[69,70], David A. Bennett[92,93], Klaus Berger[87], Lars Bertram[161,162], Hans Bisgaard[40], Dorret I. Boomsma[39], Ingrid B. Borecki[110], Ute Bültmann[163], Christopher F. Chabris[164], Francesco Cucca[165], Daniele Cusi[79,166], Ian J. Deary[53,100], George V. Dedoussis[60], Cornelia M. van Duijn[32], Johan G. Eriksson[61,167], Barbara Franke[168], Lude Franke[169], Paolo Gasparini[50,144,170], Pablo V. Gejman[147,148], Christian Gieger[43], Hans-Jörgen Grabe[171,172], Jacob Gratten[33], Patrick J. F. Groenen[173], Vilmundur Gudnason[34,83], Pim van der Harst[87,169,174], Caroline Hayward[8], David A. Hinds[5], Wolfgang Hoffmann[86], Elina Hyppönen[175,176,177], William G. Iacono[19], Bo Jacobsson[42,122], Marjo-Riitta Järvelin[178,179,180,181], Karl-Heinz Jöckel[80], Jaakko Kaprio[51,73,142], Sharon L. R. Kardia[94], Terho Lehtimäki[182,183], Steven F. Lehrer[3,184,185], Patrik K. E. Magnusson[7], Nicholas G. Martin[186], Matt McGue[19], Andres Metspalu[67,187], Neil Pendleton[188,189], Brenda W. J. H. Penninx[74], Markus Perola[67,73], Nicola Pirastu[50], Mario Pirastu[47], Ozren Polasek[81,190], Danielle Posthuma[36,191], Christine Power[176], Michael A. Province[110], Nilesh J. Samani[52,137], David Schlessinger[75], Reinhold Schmidt[55], Thorkild I. A. Sørensen[28,84,192], Tim D. Spector[65], Kari Stefansson[45,83], Unnur Thorsteinsdottir[45,83], A. Roy Thurik[30,31,193,194], Nicholas J. Timpson[84], Henning Tiemeier[32,195,196], Joyce Y. Tung[5], André G. Uitterlinden[32,197], Veronique Vitart[8], Peter Vollenweider[134], David R. Weir[109], James F. Wilson[8,81], Alan F. Wright[8], Dalton C. Conley[14], Robert F. Krueger[19], George Davey Smith[84], Albert Hofman[32], David I. Laibson[17], Sarah E. Medland[63], Michelle N. Meyer[18], Jian Yang[2,33], Magnus Johannesson[16], Peter M. Visscher[2,198], Tõnu Esko[67], Philipp D. Koellinger[1,15], David Cesarini[3,21,22,198] and Daniel J. Benjamin[3,4,26,198]

[27]Center for the Advancement of Value in Musculoskeletal Care, Hospital for Special Surgery, New York, NY, USA. [28]The Novo Nordisk Foundation Center for Basic Metabolic Research, Section of Metabolic Genetics, Faculty of Health and Medical Sciences, University of Copenhagen, Copenhagen, Denmark. [29]Department of Epidemiology Research, Statens Serum Institut, Copenhagen, Denmark. [30]Institute for Behavior and Biology, Erasmus University Rotterdam, Rotterdam, the Netherlands. [31]Department of Applied Economics, Erasmus School of Economics, Erasmus University Rotterdam, Rotterdam, the Netherlands. [32]Department of Epidemiology, Erasmus Medical Center, Rotterdam, the Netherlands. [33]Queensland Brain Institute, University of Queensland, Brisbane, QLD, Australia. [34]Icelandic Heart Association, Kopavogur, Iceland. [35]Faculty of Pharmaceutical Sciences, University of Iceland, Reykjavík, Iceland. [36]Department of Complex Trait Genetics, Center for Neurogenomics and Cognitive Research, Vrije Universiteit Amsterdam, Amsterdam, the Netherlands. [37]Amsterdam Business School, University of Amsterdam, Amsterdam, the Netherlands. [38]New York Genome Center, New York, NY, USA. [39]Department of Biological Psychology, VU University Amsterdam, Amsterdam, the Netherlands. [40]Copenhagen Prospective Studies on Asthma in Childhood, Herlev and Gentofte Hospital, University of Copenhagen, Copenhagen, Denmark. [41]Steno Diabetes Center, Gentofte, Denmark. [42]Department of Obstetrics and Gynecology, Institute of Clinical Sciences, Sahlgrenska Academy, Gothenburg, Sweden. [43]Research Unit of Molecular Epidemiology, Helmholtz Zentrum München, German Research Center for Environmental Health, Neuherberg, Germany. [44]Institute of Epidemiology II, Helmholtz Zentrum München, German Research Center for Environmental Health, Neuherberg, Germany. [45]deCODE Genetics/Amgen, Inc., Reykjavik, Iceland. [46]Department of Cell Biology, Erasmus Medical Center Rotterdam, Rotterdam, the Netherlands. [47]Istituto di Ricerca Genetica e Biomedica U.O.S. di Sassari, National Research Council of Italy, Sassari, Italy. [48]Psychology, University of Illinois, Champaign, IL, USA. [49]Institute for Computing and Information Sciences, Radboud University Nijmegen, Nijmegen, the Netherlands. [50]Department of Medical, Surgical and Health Sciences, University of Trieste, Trieste, Italy. [51]Department of Public Health, University of Helsinki, Helsinki, Finland. [52]Department of Cardiovascular Sciences, University of Leicester, Leicester LE3 9QP, UK. [53]Centre for Cognitive Ageing and Cognitive Epidemiology, University of Edinburgh, Edinburgh, UK. [54]Centre for Genomic and Experimental Medicine, Institute of Genetics and Molecular Medicine, University of Edinburgh, Edinburgh, UK. [55]Department of Neurology, General Hospital and Medical University Graz, Graz, Austria. [56]Institute for Medical Informatics, Statistics and Documentation, General Hospital and Medical University Graz, Graz, Austria. [57]Oxford Centre for Diabetes, Endocrinology & Metabolism, University of Oxford, Oxford, UK. [58]Wellcome Trust Centre for Human Genetics, University of Oxford, Oxford, UK. [59]Institute of Behavioural Sciences, University of Helsinki, Helsinki, Finland. [60]Nutrition and Dietetics, Health Science and Education, Harokopio University, Athens, Greece. [61]Folkhälsan Research Centre, Helsingfors, Finland. [62]Institute for Computing and Information Sciences, Radboud University Nijmegen, Nijmegen, the Netherlands. [63]Quantitative Genetics, QIMR Berghofer Medical Research Institute, Brisbane, Australia. [64]Lifespan Psychology, Max Planck Institute for Human Development, Berlin, Germany. [65]Department of Twin Research and Genetic Epidemiology, King's College London, London, UK. [66]NIHR Biomedical Research Centre, Guy's and St. Thomas' Foundation Trust, London, UK. [67]Estonian Genome Center,

 

University of Tartu, Tartu, Estonia. [68]Department of Epidemiology, University of Groningen, University Medical Center Groningen, Groningen, the Netherlands. [69]Public Health Stream, Hunter Medical Research Institute, New Lambton, NSW, Australia. [70]Faculty of Health and Medicine, University of Newcastle, Newcastle, NSW, Australia. [71]Centre for Integrated Genomic Medical Research, Institute of Population Health, The University of Manchester, Manchester, UK. [72]School of Psychological Sciences, The University of Manchester, Manchester, UK. [73]Department of Health, THL-National Institute for Health and Welfare, Helsinki, Finland. [74]Psychiatry, VU University Medical Center & GGZ inGeest, Amsterdam, the Netherlands. [75]Laboratory of Genetics, National Institute on Aging, Baltimore, MD, USA. [76]Research Centre of Applied and Preventive Cardiovascular Medicine, University of Turku, Turku, Finland. [77]Department of Medical Genetics, University of Lausanne, Lausanne, Switzerland. [78]Swiss Institute of Bioinformatics, Lausanne, Switzerland. [79]Department Of Health Sciences, University of Milan, Milano, Italy. [80]Institute for Medical Informatics, Biometry and Epidemiology, University Hospital of Essen, Essen, Germany. [81]Centre for Global Health Research, Usher Institute of Population Health Sciences and Informatics, University of Edinburgh, Edinburgh, UK. [82]Division of Cancer Epidemiology and Genetics, National Cancer Institute, Bethesda, MD, USA. [83]Faculty of Medicine, University of Iceland, Reykjavik, Iceland. [84]MRC Integrative Epidemiology Unit, University of Bristol, Bristol, UK. [85]School of Oral and Dental Sciences, University of Bristol, Bristol, UK. [86]Institute for Community Medicine, University Medicine Greifswald, Greifswald, Germany. [87]Department of Cardiology, University Medical Center Groningen, University of Groningen, Groningen, the Netherlands. [88]Institute of Epidemiology and Social Medicine, University of Muenster, Muenster, Germany. [89]Divisions of Genetics and Rheumatology, Department of Medicine, Brigham and Women's Hospital, Harvard Medical School, Boston, MA, USA. [90]Partners Center for Personalized Genetic Medicine, Boston, MA, USA. [91]Program in Medical and Population Genetics, Broad Institute of MIT and Harvard, Cambridge, MA, USA. [92]Rush Alzheimer's Disease Center, Rush University Medical Center, Chicago, IL, USA. [93]Department of Neurological Sciences, Rush University Medical Center, Chicago, IL, USA. [94]Department of Epidemiology, University of Michigan, Ann Arbor, MI, USA. [95]Department of Gastroenterology and Hepatology, University of Groningen, University Medical Center Groningen, Groningen, the Netherlands. [96]Institute of Epidemiology and Preventive Medicine, University of Regensburg, Regensburg, Germany. [97]Institute of Molecular Genetics, National Research Council of Italy, Pavia, Italy. [98]Department of Behavioral Sciences, Rush University Medical Center, Chicago, IL, USA. [99]Warwick Medical School, University of Warwick, Coventry, UK. [100]Department of Psychology, University of Edinburgh, Edinburgh, UK. [101]Saïd Business School, University of Oxford, Oxford, UK. [102]William Harvey Research Institute, Barts and The London School of Medicine and Dentistry, Queen Mary University of London, London, UK. [103]Princess Al-Jawhara Al-Brahim Centre of Excellence in Research of Hereditary Disorders (PACER-HD), King Abdulaziz University, Jeddah, Saudi Arabia. [104]The Berlin Aging Study II; Research Group on Geriatrics, Charité – Universitätsmedizin Berlin, Germany, Berlin, Germany. [105]Institute of Medical and Human Genetics, Charité-Universitätsmedizin, Berlin, Germany. [106]German Socio- Economic Panel Study, DIW Berlin, Berlin, Germany. [107]Health Economics Research Centre, Nuffield Department of Population Health, University of Oxford, Oxford, UK. [108]The University of Queensland Diamantina Institute, The Translational Research Institute, Brisbane, QLD, Australia. [109]Survey Research Center, Institute for Social Research, University of Michigan, Ann Arbor, MI, USA. [110]Department of Genetics, Division of Statistical Genomics, Washington University School of Medicine, St. Louis, MO, USA. [111]Institute of Human Genetics, University of Bonn, Bonn, Germany. [112]Department of Genomics,  Life and Brain Center, University of Bonn, Bonn, Germany. [113]Institute of Biomedical and Neural Engineering, School of Science and Engineering, Reykjavik University, Reykjavik, Iceland. [114]Laboratory of Epidemiology, Demography, National Institute on Aging, National Institutes of Health, Bethesda, MD, USA. [115]Department of Psychiatry, Washington University School of Medicine, St. Louis, MO, USA. [116]Division of Applied Health Sciences, University of Aberdeen, Aberdeen, UK. [117]Interfaculty Institute for Genetics and Functional Genomics, University Medicine Greifswald, Greifswald, Germany. [118]Manchester Medical School, The University of Manchester, Manchester, UK. [119]Program in Translational NeuroPsychiatric Genomics, Departments of Neurology & Psychiatry, Brigham and Women's Hospital, Boston, MA, USA. [120]Harvard Medical School, Boston, MA, USA. [121]Institute of Social and Preventive Medicine, Lausanne University Hospital (CHUV), Lausanne, Switzerland. [122]Department of Genes and Environment, Norwegian Institute of Public Health, Oslo, Norway. [123]Department of Genomics of Common Disease, Imperial College London, London, UK. [124]Department of Clinical Physiology, Tampere University Hospital, Tampere, Finland. [125]Department of Clinical Physiology, University of Tampere, School of Medicine, Tampere, Finland. [126]Public Health, Medical School, University of Split, Split, Croatia. [127]Neuroepidemiology Section, National Institute on Aging, National Institutes of Health, Bethesda, MD, USA. [128]Amsterdam Brain and Cognition Center, University of Amsterdam, Amsterdam, the Netherlands. [129]Department of Psychiatry and Behavioral Sciences, Stanford University, Stanford, CA, USA. [130]Institute of Human Genetics, Helmholtz Zentrum München, German Research Center for Environmental Health, Neuherberg, Germany. [131]LifeLines Cohort Study, University of Groningen, University Medical Center Groningen, Groningen, the Netherlands. [132]Department of Economics, University of Toronto, Toronto, ON, Canada. [133]Medical Genetics Section, Centre for Genomic and Experimental Medicine, Institute of Genetics and Molecular Medicine, University of Edinburgh, Edinburgh, UK. [134]Department of Internal Medicine, Internal Medicine, Lausanne University Hospital (CHUV), Lausanne, Switzerland. [135]Tema BV, Hoofddorp, the Netherlands. [136]Molecular Epidemiology, QIMR Berghofer Medical Research Institute, Brisbane, QLD, Australia. [137]NIHR Leicester Cardiovascular Biomedical Research Unit, Glenfield Hospital, Leicester, UK. [138]Institute of Health and Biomedical Innovation, Queensland Institute of Technology, Brisbane, QLD, Australia. [139]Analytic and Translational Genetics Unit, Massachusetts General Hospital, Boston, MA, USA. [140]Stanley Center for Psychiatric Research, Broad Institute of MIT and Harvard, Cambridge, MA, USA. [141]Psychiatric & Neurodevelopmental Genetics Unit, Department of Psychiatry, Massachusetts General Hospital, Boston, MA, USA. [142]Institute for Molecular Medicine Finland (FIMM), University of Helsinki, Helsinki, Finland. [143]Department of Neurology, Massachusetts General Hospital, Boston, MA, USA. [144]Medical Genetics, Institute for Maternal and Child Health IRCCS "Burlo Garofolo", Trieste, Italy. [145]Social Impact, Arlington, VA, USA. [146]Department of Economics, University of Minnesota Twin Cities, Minneapolis, MN, USA. [147]Department of Psychiatry and Behavioral Sciences, NorthShore University HealthSystem, Evanston, IL, USA. [148]Department of Psychiatry and Behavioral Neuroscience, University of Chicago, Chicago, IL, USA. [149]Public Health Genomics Unit,  National Institute for Health and Welfare, Helsinki, Finland. [150]Research Unit for Genetic Epidemiology, Institute of Molecular Biology and Biochemistry, Center of Molecular Medicine, General Hospital and Medical University, Graz, Austria. [151]Information Based Medicine Stream, Hunter Medical Research Institute, New Lambton, NSW, Australia. [152]Medical Research Institute, University of Dundee, Dundee, UK. [153]Research Unit Hypertension and Cardiovascular Epidemiology, Department of Cardiovascular Science, University of Leuven, Leuven, Belgium. [154]R&D VitaK Group, Maastricht University, Maastricht, the Netherlands. [155]Institute of Genetic Epidemiology, Helmholtz Zentrum München, German Research Center for Environmental Health, Neuherberg, Germany. [156]Institute of Medical Informatics, Biometry and Epidemiology, Chair of Genetic Epidemiology, Ludwig Maximilians-Universität, Munich, Germany. [157]Department of Geriatrics, Florida State University College of Medicine, Tallahassee, FL, USA. [158]Department of Health Sciences and Genetics, University of Leicester, Leicester, UK. [159]Department of Internal Medicine, Erasmus Medical Center, Rotterdam, the Netherlands. [160]Research Center for Group Dynamics, Institute for Social Research, University of Michigan, Ann Arbor, MI, USA. [161]Platform for Genome Analytics, Institutes of Neurogenetics & Integrative and Experimental Genomics, University of Lübeck, Lübeck, Germany. [162]Neuroepidemiology and Ageing Research Unit, School of Public Health, Faculty of Medicine, The Imperial College of Science, Technology and Medicine, London, UK. [163]Department of Health Sciences, Community & Occupational Medicine, University of Groningen, University Medical Center Groningen, Groningen, the Netherlands. [164]Autism and Developmental Medicine Institute, Geisinger Health System, Lewisburg, PA, USA. [165]Istituto di Ricerca Genetica e Biomedica (IRGB), Consiglio Nazionale delle Ricerche, c/o Cittadella Universitaria di Monserrato, Monserrato, Cagliari, Italy. [166]Institute of Biomedical Technologies, Italian National Research Council, Segrate (Milano), Italy. [167]Department of General Practice and Primary Health Care, University of Helsinki, Helsinki, Finland. [168]Departments of Human Genetics and Psychiatry, Donders Centre for Neuroscience, Nijmegen, the Netherlands. [169]Department of Genetics, University Medical Center Groningen, University of Groningen, Groningen, the

Netherlands. [170]Experimental Genetics Division, Sidra, Doha, Qatar. [171]Department of Psychiatry and Psychotherapy, University Medicine Greifswald, Greifswald, Germany. [172]Department of Psychiatry and Psychotherapy, HELIOS-Hospital Stralsund, Stralsund, Germany. [173]Econometric Institute, Erasmus School of Economics, Erasmus University Rotterdam, Rotterdam, the Netherlands. [174]Durrer Center for Cardiogenetic Research, ICIN-Netherlands Heart Institute, Utrecht, the Netherlands. [175]Centre for Population Health Research, School of Health Sciences and Sansom Institute, University of South Australia, Adelaide, Australia. [176]South Australian Health and Medical Research Institute, Adelaide, SA, Australia. [177]Population, Policy and Practice, UCL Institute of Child Health, London, UK. [178]Department of Epidemiology and Biostatistics, MRC-PHE Centre for Environment & Health, School of Public Health, Imperial College London, London, UK. [179]Center for Life Course Epidemiology, Faculty of Medicine, University of Oulu, Oulu, Finland. [180]Unit of Primary Care, Oulu University Hospital, Oulu, Finland. [181]Biocenter Oulu, University of Oulu, Oulu, Finland. [182]Fimlab Laboratories, Tampere, Finland. [183]Department of Clinical Chemistry, University of Tampere, School of Medicine, Tampere, Finland. [184]School of Policy Studies, Queen's University, Kingston, Ontario, Canada. [185]Department of Economics, New York University Shanghai, Pudong, Shanghai, China. [186]Genetic Epidemiology, QIMR Berghofer Medical Research Institute, Brisbane, QLD, Australia. [187]Institute of Molecular and Cell Biology, University of Tartu, Tartu, Estonia. [188]Centre for Clinical and Cognitive Neuroscience, Institute Brain Behaviour and Mental Health, Salford Royal Hospital, Manchester, UK. [189]Manchester Institute Collaborative Research in Ageing, University of Manchester, Manchester, UK. [190]Faculty of Medicine, University of Split, Croatia, Split, Croatia. [191]Department of Clinical Genetics, VU Medical Centre, Amsterdam, the Netherlands. [192]Institute of Preventive Medicine, Bispebjerg and Frederiksberg Hospitals, The Capital Region, Frederiksberg, Denmark. [193]Montpellier Business School, Montpellier, France. [194]Panteia, Zoetermeer, the Netherlands. [195]Department of Psychiatry, Erasmus Medical Center, Rotterdam, the Netherlands. [196]Department of Child and Adolescent Psychiatry, Erasmus Medical Center, Rotterdam, the Netherlands. [197]Department of Internal Medicine, Erasmus Medical Center, Rotterdam, the Netherlands.

## Methods

This article is accompanied by a Supplementary Note with further details.

**Coding the EduYears phenotype.** As in previous GWAS[2,61,62], the EduYears phenotype was coded by mapping the highest level of education that a respondent achieved to an International Standard Classification of Education 1997 category and then imputing a years-of-education equivalent for each International Standard Classification of Education 1997 category. Details on cohort-level phenotype measures, genotyping and imputation are in Supplementary Table 15.

Our phenotype coding was unchanged from previous GWAS, except in the UKB. UKB participants with a qualification of 'NVQ or HND or HNC or equivalent' (National Vocational Qualification, Higher National Diploma and Higher National Certificate, respectively) but no college or university degree were previously coded as having 19 years of education[2,62], but this classification overstates their average years of schooling (Supplementary Note section 1 and Supplementary Fig. 3). We therefore recoded EduYears for these participants as the age they reported leaving full-time education minus five. We dropped holders of a National Vocational Qualification/Higher National Diploma/Higher National Certificate/equivalent who reported leaving full-time education before age 12 years (fewer than 50 individuals).

In previous GWAS, individuals younger than 30 years when EA was measured were excluded to ensure that almost everyone had completed formal schooling. In the 23andMe GWAS for the current paper, ~16% of the individuals are aged 16–29 years. To explore the effect of including these individuals, we conducted a simulation using the UKB data (Supplementary Note section 1.2). The results indicate that the inclusion of individuals aged younger than 30 years in the 23andMe GWAS is unlikely to have materially affected our meta-analysis results.

**Additive GWAS.** For our additive GWAS of EduYears, we meta-analyzed three sets of summary statistics: publicly available results from Lee et al.[2] that exclude 23andMe and UKB ($N = 324,162$), new association results from 23andMe ($N = 2,272,216$) and new association results from a GWAS we conducted in UKB with the identical methodology as in Lee et al. but with the improved coding of EduYears described above ($N = 441,121$). All cohort-level analyses were restricted to European-genetic-ancestry individuals who passed the cohort's quality-control filters and, except in 23andMe as described above, whose EA was measured at an age of at least 30 years. We did not run sex-stratified analyses for the autosomal meta-analysis, because there is compelling evidence from our prior work that the male–female genetic correlation for EduYears is close to one. For example, the Okbay et al.[62] data yield an estimate of 0.98 (s.e. = 0.029).

To the new 23andMe and UKB results, we applied a quality-control protocol similar to the one described previously[62] and implemented in the EasyQC R package but updated to a more recent reference panel and adjusted to account for the large GWAS sample sizes (Supplementary Note section 2.2.5 and Supplementary Table 16). Using the software METAL (ref. [63]), for all SNPs that passed the quality-control thresholds in the new 23andMe and UKB results, we conducted a sample-size-weighted meta-analysis of these new results with the 69 results files from Lee et al.[2] (all except 23andMe and UKB). After the meta-analysis, we inflated the standard errors by the square root of the intercept ($\sqrt{1.663}$) from an LD score regression[8] estimated from the meta-analysis summary statistics.

We selected the set of approximately independent genome-wide-significant SNPs using the same iterative clumping algorithm used previously[2] and implemented in Plink (ref. [64]), with a pairwise $r^2$ cutoff of 0.1 and no physical distance cutoff (Supplementary Note section 2.2.6 and Supplementary Table 1). We assessed the sensitivity of our conclusions about the number of lead SNPs with a COJO multiple-SNP analysis[10] using the implementation in the GCTA software[65] (Supplementary Note section 2.2.7), with SNPs farther than 100 Mb apart assumed to have zero correlation. We applied our clumping algorithm to classify each of the COJO lead SNPs as either primary (if retained by the algorithm) or secondary (if eliminated) (Supplementary Table 2).

**X-chromosome analyses.** We conducted separate association analyses of the X-chromosome SNPs in UKB and 23andMe (Supplementary Note section 3). The 23andMe analysis ($N = 2,272,216$) was conducted in a pooled male–female sample using a 0/2 genotype coding for males. The UKB analysis ($N = 440,817$) was an inverse-variance-weighted meta-analysis (assuming 0/2 genotype coding to match the 23andMe analysis) of sex-stratified association analyses conducted using BOLT-LMM v2.3.4 (ref. [66]). Following Supplementary Note section 4.1 of Lee et al., we used the sex-stratified UKB analyses to estimate the X-chromosome SNP heritability for males and females, as well as the male–female genetic correlation (Supplementary Note section 3.1, Supplementary Table 17).

We performed a sample-size-weighted meta-analysis of the 211,581 SNPs that were available in both UKB and 23andMe, passed the quality control filters (Supplementary Note section 3.3 and Supplementary Table 16) and had a sample size greater than 500,000. To adjust for uncontrolled-for population stratification, we inflated the standard errors by the square root of the LD score intercept from an autosomal meta-analysis of UKB and 23andMe ($\sqrt{1.666}$). We selected the set of approximately independent genome-wide-significant SNPs using the same clumping algorithm as in the additive GWAS (Supplementary Note section 2.2.6).

**Dominance GWAS.** We conducted a sample-size-weighted meta-analysis for 5,870,596 autosomal SNPs that passed quality control filters and were available in both the 23andMe ($N = 2,272,216$) and UKB ($N = 302,037$) summary statistics. Similar to the additive GWAS, after the meta-analysis, we inflated the standard errors by the square root of the intercept from an LD score regression. We used LD scores that account for the faster decay of information from tagged SNPs as a function of LD for dominance effects (e.g., Hivert et al.[13]). The LD score regression was restricted to the set of HapMap3 SNPs, and the dominance LD scores were estimated using the 1000 Genomes phase 1 reference sample[67].

We decomposed the variance in the estimated dominance effect sizes into shares due to true signal of dominance genetic variance and sampling variation (Supplementary Note section 4.5 and Supplementary Table 18). We also conducted a series of preregistered replication exercises (https://osf.io/uegqv/) to assess whether the estimates of the dominance effects for various subsets of SNPs are consistent across UKB and 23andMe (Supplementary Note sections 4.6 and 8 and Supplementary Table 19).

To estimate ID for EA, we used ldscdom software, which implements a recently developed method[17] that uses GWAS summary statistics to obtain an estimate of the slope from the regression of the phenotype of interest (EA) on the inbreeding coefficient across individuals. Supplementary Note section 4.7 provides details, and Supplementary Table 20 shows the estimates of ID for each cohort separately, as well as the inverse-variance-weighted meta-analysis of these two estimates.

**Polygenic prediction.** From a GWAS meta-analysis that omits Add Health, HRS and WLS, the SNP weights for our main PGIs were obtained using LDpred (v. 1.0.11)[21], assuming a Gaussian prior for the distribution of effect sizes and restricting to HapMap3 SNPs. LD patterns were estimated in a sample of 14,028 individuals and 1,214,408 HapMap3 SNPs from the public release of the Haplotype Reference Consortium reference panel[68]. The PGIs were obtained in Plink2 (ref. [69]) by multiplying the genotype probabilities at each SNP by the corresponding estimated posterior mean calculated by LDpred and then summing over all included SNPs (Supplementary Note section 5.1 and Supplementary Table 4). We also constructed a PGI for the African-genetic-ancestry individuals in HRS and Add Health using the same LDpred weights (Supplementary Table 21).

The 'clumping and thresholding' PGIs with $P$ value cutoffs of $5 \times 10^{-8}$, $5 \times 10^{-5}$, $5 \times 10^{-3}$ and 1 (i.e., all SNPs) were made in Plink2 (ref. [69]) using the clumping algorithm described in the section 'Additive genome-wide-association study meta-analysis' and the procedure described above. The SNP weights were set equal to the coefficient estimates from the meta-analysis (Supplementary Table 3).

The SNP weights for the SBayesR (ref. [22]) PGI were obtained using GCTB software[70]. We assume four components in the finite mixture model, with initial mixture probabilities $\pi = (0.95, 0.02, 0.02, 0.01)$ and fixed $\gamma = (0.0, 0.01, 0.1, 1)$, where $\gamma$ is a parameter that constrains how the SNP-effect-size variance scales in each of the four distributions. LD was estimated in 2,865,810 pruned common variants from the full UKB European-genetic-ancestry ($N \approx 450,000$) dataset from Lloyd-Jones et al.[22]. Weights were obtained for 2,548,339 of these SNPs that overlapped with the summary statistics after excluding the major histocompatibility complex region. PGIs were constructed in Plink2 (ref. [69]) by multiplying the genotype probabilities at each SNP by the corresponding estimated posterior mean calculated by SBayesR and then summing over all included SNPs (Supplementary Table 3).

We analyzed how well the PGIs predict a host of phenotypes related to educational attainment, academic achievement and cognition (Supplementary Note section 5.2). All regressions include controls for year of birth or age at assessment, sex, their interactions and the first ten PCs of the variance–covariance matrix of the genomic relatedness matrix. In our analyses of grade point average outcomes in Add Health, we also controlled for high-school fixed effects (Supplementary Note section 5.3).

To evaluate prediction accuracy, we first regress the phenotype on the controls listed above without the PGI. Next, we rerun the regression but with the PGI included. For quantitative phenotypes, our measure of predictive power is the incremental $R^2$, or the difference in $R^2$ between the regressions with and without the PGI. For binary outcomes, we proceed similarly but calculate the incremental Nagelkerke $R^2$ from a Probit regression. We obtained 95% CIs around the incremental (Nagelkerke) $R^2$ values by performing a bootstrap with 1,000 repetitions.

**Expected prediction accuracy of the EA PGI.** We calculate the expected prediction accuracy of the EA PGI using a generalization of de Vlaming et al.[71]. The expected coefficient of determination, $R^2$, can be expressed as the following function of the discovery sample size, $N$:

$$E\left(R^2\right) = \frac{A}{B + 1/N}.$$

Although $A$ may vary by prediction sample, $B$ does not. We estimate $A$ and $B$ by nonlinear least squares using data from Add Health and HRS. More details of this calculation can be found in Supplementary Note section 5.5.

**Analysis of European genetic ancestries to African genetic ancestries relative accuracy in UKB.** We used a method that was recently developed by Wang et al.[25]

to investigate the factors contributing to the substantial loss of prediction accuracy of the EduYears PGI in samples of African genetic ancestries. We define the European genetic ancestries to African genetic ancestries relative accuracy (RA) as

$$RA_{E \to A} = \frac{R_{AFR}^2}{R_{EUR}^2},$$

where $R_{AFR}^2$ and $R_{EUR}^2$ are prediction accuracies of PGIs derived from a GWAS conducted in European-genetic-ancestry populations. To facilitate comparability with Wang et al.'s results for eight other phenotypes, we extended their original analyses to also include EduYears. We thus performed a GWAS of HapMap3 SNPs (1,365,446 SNPs) in a sample of European-genetic-ancestry individuals in UKB ($N$ = 425,231). We identified 507 approximately independent genome-wide-significant SNPs (using the LD clumping algorithm implemented in Plink (ref.[64]), setting the window size equal to 1 Mb and the LD $r^2$ threshold to 0.1). We then used these 507 SNPs to generate PGIs and evaluate their accuracy in UKB holdout samples of African-genetic-ancestry individuals ($N$ = 6,514) and European-genetic-ancestry individuals ($N$ = 10,000). To compare our empirical estimate of RA to the RA predicted by the model, we used genotypes from 503 European-genetic-ancestry and 504 African-genetic-ancestry participants in the 1000 Genomes Project to estimate genetic-ancestry-specific MAF and LD correlations between all candidate causal variants (defined as any SNP within a 100-kb window of a genome-wide-significant SNP whose squared correlation with the genome-wide-significant SNP is above 0.45). Following Wang et al., we then substituted these estimates into their equation (2) (Supplementary Table 5 and Extended Data Fig. 8).

**Prediction of disease risk from the EA PGI.** The EA PGI was constructed using LDpred (v.1.0.11) (ref. [21]) as described above but using the summary statistics of a meta-analysis of EA that excludes UKB. Disease-specific PGIs were constructed using summary statistics from GWAS conducted among participants of European genetic ancestries for nine phenotypes (Supplementary Table 22). The PGI for coronary artery disease was used to predict two diseases: ischemic heart disease and myocardial infarction. For all phenotypes other than migraine, we generated weights using LDpred and constructed the PGI using Plink1.9. LDpred was run using the same settings and Haplotype Reference Consortium reference data used for the EA PGI. For migraine, only SNPs with association $P$ value < $10^{-5}$ were available in the summary statistics, so we generated the PGI using clumping and thresholding. Disease phenotypes were generated based on UKB Category 1712 and Data Field 41270 (Supplementary Note section 6.1.2 and Supplementary Tables 23 and 24).

For the various diseases, we computed the predictive power of (1) the EA PGI, (2) the disease-specific PGI and (3) these two PGIs together with their interaction (Supplementary Table 6). Our measure of predictive power is the incremental Nagelkerke's $R^2$ of adding the variable(s) to a logistic regression of the disease phenotype on sex, a third-degree polynomial in birth year and interactions with sex, the first 40 PCs and batch dummies. 95% CIs around the incremental Nagelkerke's $R^2$ were obtained by performing a bootstrap with 1,000 repetitions.

We also computed the odds ratio for selected diseases by deciles of the EA PGI in UKB (Supplementary Tables 7 and 8). Odds ratios and 95% CIs were estimated using logistic regression while controlling for covariates (Supplementary Note section 6.2.1).

**Comparing direct and population effects.** To compare the direct effect of the PGI on various phenotypes to its population effect, we used data on siblings and trios from UKB (ref.[3]), GS (ref. [7]) and STR (ref. [38]). In both UKB and GS, first-degree relatives were identified using KING with the "–related–degree 1" option[72]. For parent–offspring relations, the parent was identified as the older individual in the pair. We removed 621 individuals from GS that had been previously identified by GS as being also present in UKB (Supplementary Note section 7.3).

We analyzed PGIs for EA and cognitive performance in all three samples and height and BMI only in UKB and GS. PGIs were made using GWAS results that exclude GS, STR and all related individuals of up to third degree from UKB (Supplementary Note section 7.3), following the LDpred PGI pipeline described in Supplementary Note section 5.1.

We selected 23 phenotypes related to education, cognition, income and health (Supplementary Table 9) available in at least one of the datasets. For each phenotype in each dataset, we first regressed the phenotype onto sex and age, age² and age³ and their interactions with sex. In addition, for UKB, we included as covariates the top 40 genetic PCs provided by UKB and the genotyping array dummies[3]. For GS and STR, we included the top 20 genetic PCs (Supplementary Note section 5.3 explains how the PCs were created). We then took the residuals from the regression of the phenotype on the covariates and normalized the residual variance within each sample separately so that the phenotypic residual variance was 1 in each sex in the combined sample of siblings and individuals with both parents genotyped. The PGIs of the phenotyped individuals were also normalized to have variance 1 in the same sample. Thus, effect estimates correspond to (partial) correlations, and their squares to proportions of phenotypic variance explained.

We give an overview of the statistical analyses performed here, with details in Supplementary Note section 7.4. In the siblings, we regressed individuals'

phenotypes onto the difference between the individual's PGI and the mean PGI among the siblings in that individual's family and the mean PGI among siblings in that family. In trios, we regressed phenotypes onto the individual's PGI and the individual's father's and mother's PGIs. In both the siblings and trios, we used a linear mixed model to account for relatedness in the samples. We meta-analyzed the results from the siblings and trios, accounting for covariance between the estimates from the sibling and trio samples from the same datasets. We applied a transformation to the meta-analysis that accounts for assortative mating to estimate the population effect of the PGI and the difference between the direct and population effects.

**Analysis of assortative mating.** We identified mate pairs in UKB (862 mate pairs) and GS (1603 mate pairs) by identifying genotyped parents of genotyped individuals within each sample. Let $r_y$ denote the phenotypic correlation between mate pairs, and let $r_p$ and $r_m$ denote the correlations between the phenotype and PGI for the father and mother, respectively. The correlation between the mate-pair PGIs should be equal to $r_y r_p r_m$ if the correlation is explained by assortative mating on the phenotype alone, and the relationship between the PGI and the phenotype is linear. To test the model of phenotypic assortment, we estimated the expected correlation between mate-pair PGIs by estimating $r_y$, $r_p$ and $r_m$. We estimated the standard error of the product of $r_y$, $r_p$ and $r_m$ using 1,000 bootstrap samples where we sampled over the mate pairs. We also estimated the correlation between the residual of the father's PGI after regression onto the father's phenotype and the residual of the mother's PGI after regression onto the mother's phenotype, which should be zero under phenotypic assortment if the relationship between phenotype and PGI is linear. We performed further analyses adjusting for genetic PCs, birth coordinates, UKB assessment center, cognitive performance and vocabulary to test whether assortative mating on factors related to ancestry, geography and cognition explained the mate-pair PGI correlations (Supplementary Note section 9).

**Reporting Summary.** Further information on research design is available in the Nature Research Reporting Summary linked to this article.

## Data availability
GWAS summary statistics can be downloaded from http://www.thessgac.org/data subject to a terms of use to encourage responsible use of the data. We provide association results for all SNPs that passed quality-control filters in autosomal, X chromosome and dominance GWAS meta-analyses that exclude the research participants from 23andMe. SNP-level summary statistics from analyses based entirely or in part on 23andMe data can only be reported for up to 10,000 SNPs. For the complete dominance GWAS meta-analysis, which includes 23andMe, clumped results for the 1,000 SNPs with the smallest $P$ values are provided. For the complete autosomal and X chromosome GWAS meta-analyses, respectively, clumped results for the 8,618 and 141 SNPs with $P$ < $10^{-5}$ are provided; this $P$ value threshold was chosen such that the total number of SNPs across the analyses that include data from 23andMe does not exceed 10,000. The full GWAS summary statistics from 23andMe will be made available through 23andMe to qualified researchers under an agreement with 23andMe that protects the privacy of the 23andMe participants. Please visit https://research.23andme.com/collaborate/#dataset-access/ for more information and to apply to access the data.

## Code availability
The following software packages were used for data analysis: Python version 3.7.4 with packages pandas 0.25.1, scipy 1.3.1, numpy 1.17.2, matplotlib 3.1.1 and argparse 1.1 (https://anaconda.com); R version 4.0.3 with packages EasyQC 9.2, plotrix 3.7.8, tidyr 1.1.3 and readstata13 0.9.2, and R version 3.6 with packages ggplot2 3.3 and fmsb 0.7 (https://www.r-project.org); GCTA 1.93.2beta (https://yanglab.westlake.edu.cn/software/gcta/#Overview); GCTB 2.03 (https://cnsgenomics.com/software/gctb/#Overview); Stata 16.1 (https://www.stata.com); Plink1.9 (https://www.cog-genomics.org/plink/1.9); Plink2 (https://www.cog-genomics.org/plink/2.0); LDpred 1.0.11 (https://github.com/bvilhjal/ldpred); METAL release 2011-03-25 (https://genome.sph.umich.edu/wiki/METAL_Documentation); BOLT-LMM 2.3 (https://alkesgroup.broadinstitute.org/BOLT-LMM/BOLT-LMM_manual.html); LDSC 1.0.1 (https://github.com/bulik/ldsc); and SNIPar (https://github.com/AlexTISYoung/SNIPar/tree/EA4).

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

## Acknowledgements

We thank E.M. Tucker-Drob for helpful comments and J. Zeng for help with the SBayesR software. This research was carried out under the auspices of the Social Science Genetic Association Consortium. The analyses reported in the paper fall under National Bureau of Economic Research institutional review board protocols 19_434, 19_465 and 20_041. This paper uses cohort-level data from Okbay et al.[62], and information about studies participating in that study can be found in the Additional Acknowledgements Supplementary section of that paper. Per Social Science Genetic Association Consortium policy, we acknowledge the authors of that paper, listed below, as collaborators. 23andMe research participants provided informed consent and participated in the research online, under a protocol approved by the external Association for the Accreditation of Human Research Protection Programs-accredited institutional review board, Ethical & Independent Review Services. Participants were included in the analysis on the basis of consent status as checked at the time data analyses were initiated. We would like to thank the research participants and employees of 23andMe for making this work possible. We gratefully acknowledge the contributions of members of 23andMe's Research Team, whose names are listed below. The research has also been conducted using the UKB Resource under application numbers 11425 and 12505. Informed consent was obtained from UKB subjects. Ethical approval for the GS: Scottish Family Health Study was obtained from the Tayside Committee on Medical Research Ethics (on behalf of the National Health Service). H.J, M.B., D. Cesarini and P.T. were supported by the Ragnar Söderberg Foundation (E42/15 to D. Cesarini); A.O. and P.K. by the European Research Council (consolidator grant 647648 EdGe to P.K.); H.J., M.B., S.M.N., T.G., C.W., J.J., M.N.M., D. Cesarini, P.T., J.P.B., D.J.B. and A.I.Y. by Open Philanthropy (grant 010623-00001 to D.J.B.); R.A. and S.O. by Riksbankens Jubileumsfond (grant P18-0782:1 to S.O.); N.W., G.G., C.W., L.Y. and D.J.B. by the National Institute on Aging (NIA)/National Institutes of Health (NIH) (grants R24-AG065184 and R01-AG042568 to D.J.B.); D.J.B. by the NIA/NIH (grant R56-AG058726 to T. Galama); P.T. by the NIA/National Institute on Mental Health (grants R01-MH101244-02 and U01-MH109539-02 to B. Neale); J.S. and P.M.V. by the Australian Research Council (grant FL180100072 to P.M.V.); and Y.W., L.Y. and P.M.V. by the National Health and Medical Research Council (grant GNT113400 to P.M.V.). The study was also supported by Netherlands Organisation for Scientific Research VENI (grant 016.Veni.198.058 to A.O.); the F.G. Meade Scholarship and UQ Research Training Scholarship from the University of Queensland Senate (Y.W.); the Swedish Research Council (grant 2019-00244 to S.O.); an MRC University Unit Programme Grant (MC_UU_00007/10, QTL in Health and Disease, to C.H.); the Swedish Research Council (grant 421-2013-1061 to M.J.); Pershing Square Fund of the Foundations of Human Behavior (D.L.); the Li Ka Shing Foundation (A.K.); the Australian Research Council (grant DE200100425 to L.Y.); the NIA/NIH (grant K99-AG062787-01 to P.T.); the Government of Canada through Genome Canada and the Ontario Genomics Institute (grant OGI-152 to J.P.B.); the Social Sciences and Humanities Research Council of Canada (J.P.B.); and the Australian Research Council (P.M.V.).

## Author contributions

A.O., L.Y., D. Cesarini, P.T., P.M.V., J.P.B., D.J.B. and A.I.Y. designed and oversaw the study. A.O. was the study's lead analyst, responsible for GWAS, quality control, meta-analyses, analyzing the predictive power of the PGI for EA and cognition outcomes and creating the PGIs used in other analyses (except for the disease PGIs). M.B. and H.K. conducted the recoding of the educational attainment measure in the UKB. A.O. and J.P.B. performed the GWAS replication. J.P.B. calculated the winner's-curse-adjusted effect sizes. L.Y. conducted the analysis of predicted and actual PGI accuracy in the African-genetic-ancestry sample in the UKB. H.J. ran the bioinformatics analysis, under J.J.L.'s guidance. A.O., N.W., L.Y. and J.P.B. conducted the dominance GWAS meta-analysis. A.O., J.S. and P.M.V. oversaw and ran the X chromosome meta-analysis. Y.W. analyzed the predictive power of the PGI for disease phenotypes. S.M.N., R.A., S.O. and A.I.Y. conducted the within-family analyses. H.J., D. Cesarini and A.I.Y. conducted the assortative mating analyses. Besides the contributions explicitly listed above, N.W., H.J., M.B., G.G. and T.G. assisted for several subsections. C.W. coordinated data organization, and J.J. organized the computing infrastructure. D. Conley, P.D.K., M.J., D.L. and M.N.M. provided important input and feedback on various aspects of the study design. All authors contributed to and critically reviewed the manuscript.

## Competing interests

Y.J., B.H., C.T., D.A.H. and the members of the 23andMe Research Team are current or former employees of 23andMe, Inc. All other authors declare no competing interests.

## Additional information

**Extended data** is available for this paper at https://doi.org/10.1038/s41588-022-01016-z.

**Correspondence and requests for materials** should be addressed to Aysu Okbay, Peter M. Visscher, Daniel J. Benjamin or Alexander I. Young.

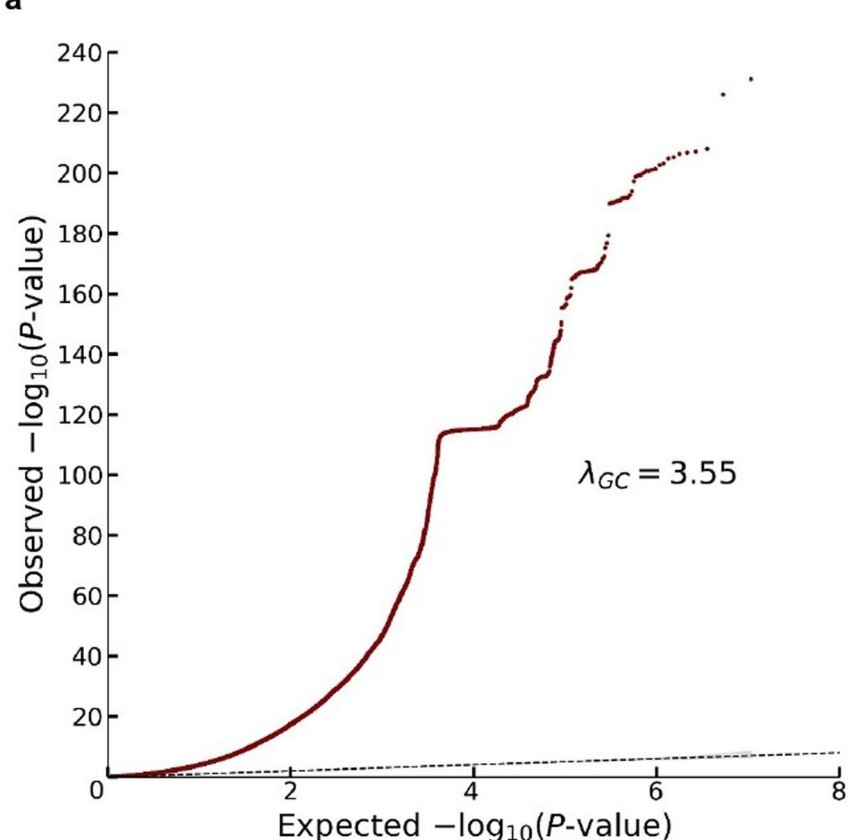

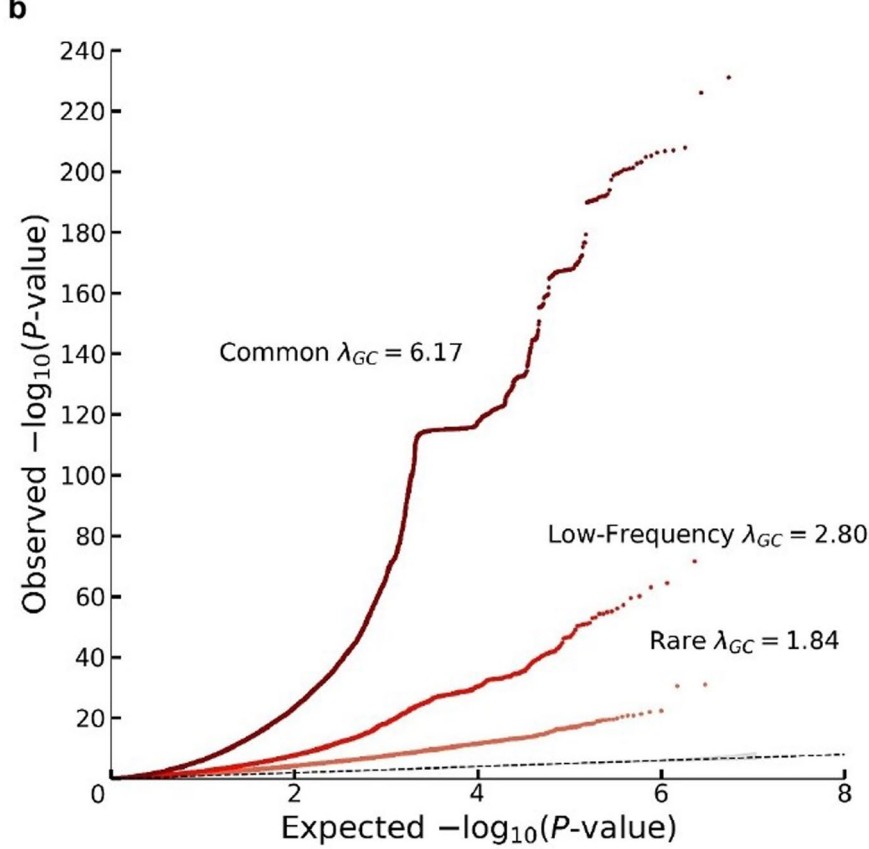

**Extended Data Fig. 1 | See next page for caption.**

**Extended Data Fig. 1 | Quantile-quantile plots for the additive GWAS meta-analysis.** The panels display Q-Q plots, which show the -$\log_{10}$($P$-values) based on a two-sided $Z$-tests for **(a)** all SNPs and **(b)** SNPs grouped by minor allele frequency (MAF): rare (<1%), low frequency (1–5%) and common (>5%). The plots and $\lambda_{GC}$ numbers are based on the unadjusted GWAS summary statistics (that is with standard errors that were *not* inflated by the square root of the estimated LD Score intercept). The dotted line represents the expected -$\log_{10}$($P$-values) under the null hypothesis. The (barely visible) gray shaded areas in the Q-Q plots represent the 95% confidence intervals under the null hypothesis. The flat horizontal region in the plots is an inversion region in chromosome 17 (17q21.31).

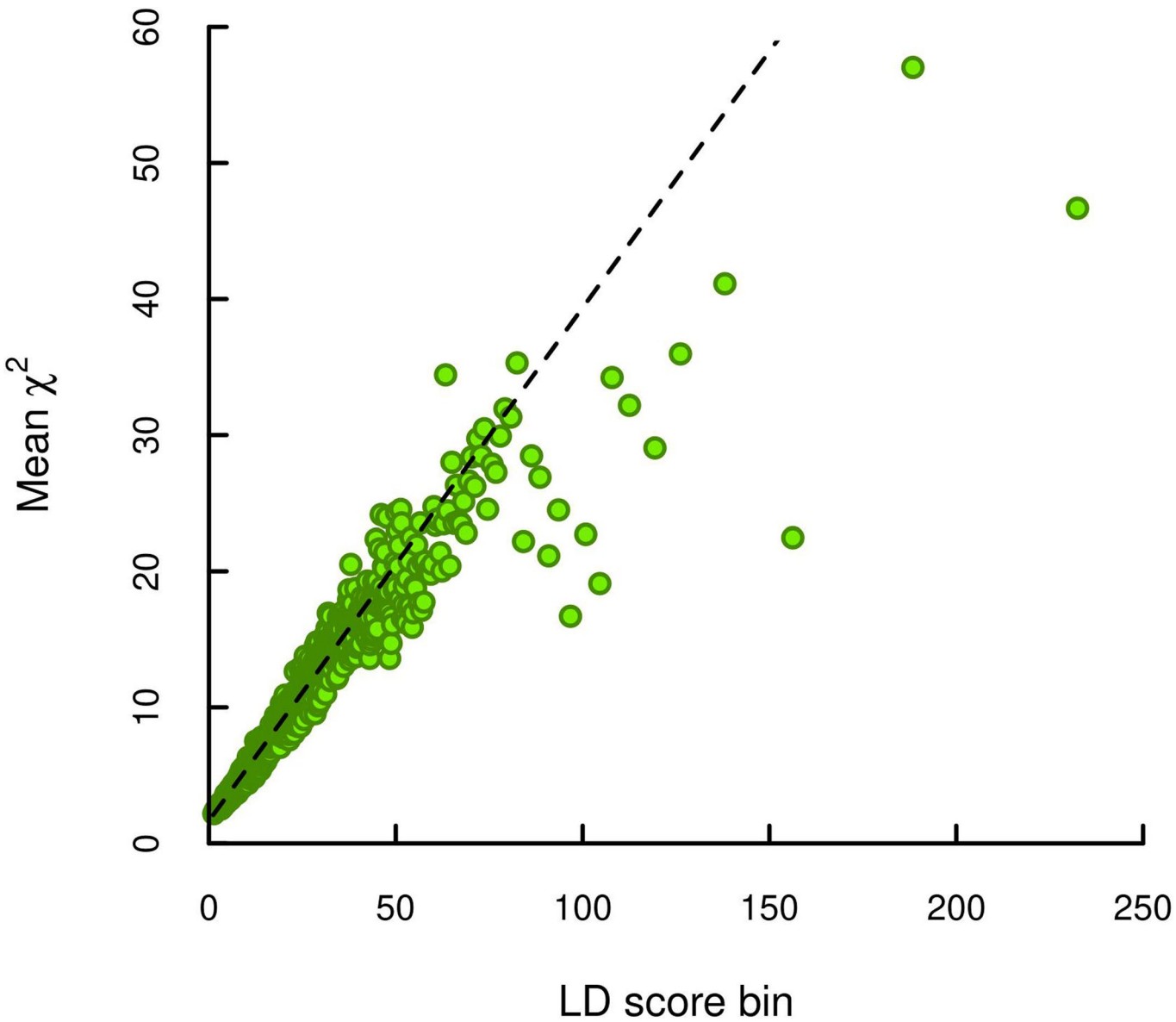

**Extended Data Fig. 2 | LD score plot from the additive GWAS meta-analysis.** Each point represents an LD score quantile containing 1000 SNPs (except for the last quantile, which contains 709). The *x* and *y* coordinates of each point are the mean LD score and the mean statistic of SNPs in that quantile. The LD score regression intercept is 1.663, suggesting that biases due to stratification or cryptic relatedness explain roughly 7% of the inflation in test statistics (see Supplementary Note section 2.2.6).

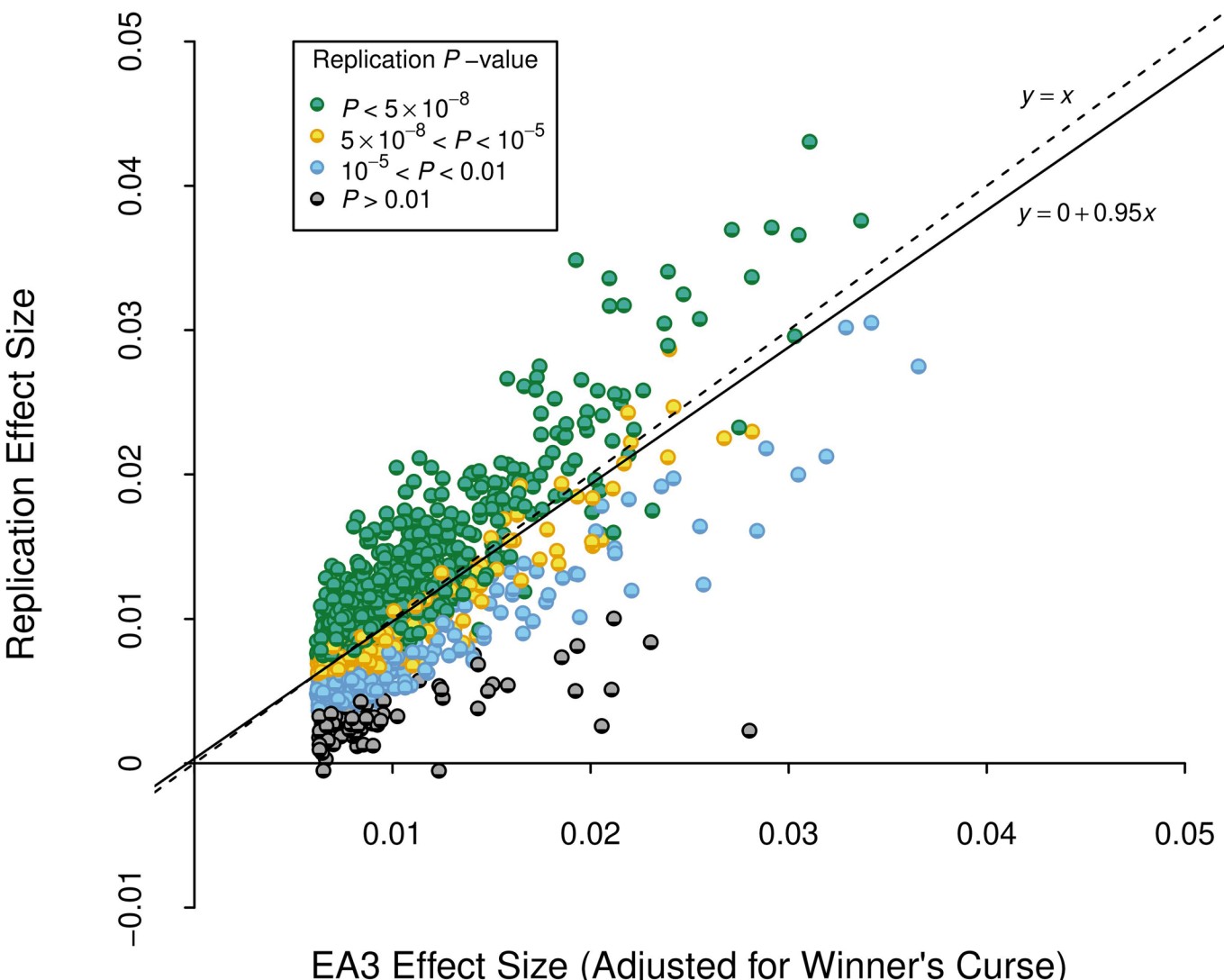

**Extended Data Fig. 3 | Replication of EA3 lead SNPs.** We examined the out-of-sample replicability of the 1,504 lead SNPs identified at genome-wide significance in a version of our previously published GWAS meta-analysis of *EduYears* (EA3), with the UKB GWAS in that analysis replaced by a UKB GWAS that uses the new phenotype coding explained in Supplementary Note section 1.1. Prior to clumping, we dropped SNPs that had a sample size smaller than 80% of the maximum sample size in the updated EA3 data ($N_{EA3,max}$ = 1,130,819), or that had a sample size in the new data smaller than 80% of the maximum sample size of the new data ($N_{new,max}$ = 2,272,216). The *x* axis is the winner's-curse-adjusted estimate of the SNP's effect size in the updated EA3 study (calculated using shrinkage parameters estimated using summary statistics from EA3). The *y* axis is the SNP's effect size estimated from the subsample of our data that did not contribute to the EA3 GWAS. All effect sizes are from a regression where the phenotype has been standardized to have unit variance. The reference allele is chosen to be the allele estimated to increase EA in EA3. The dashed line is the identity, and the solid line is the fitted regression line. *P*-values are based on two-sided *Z*-tests.

**a**

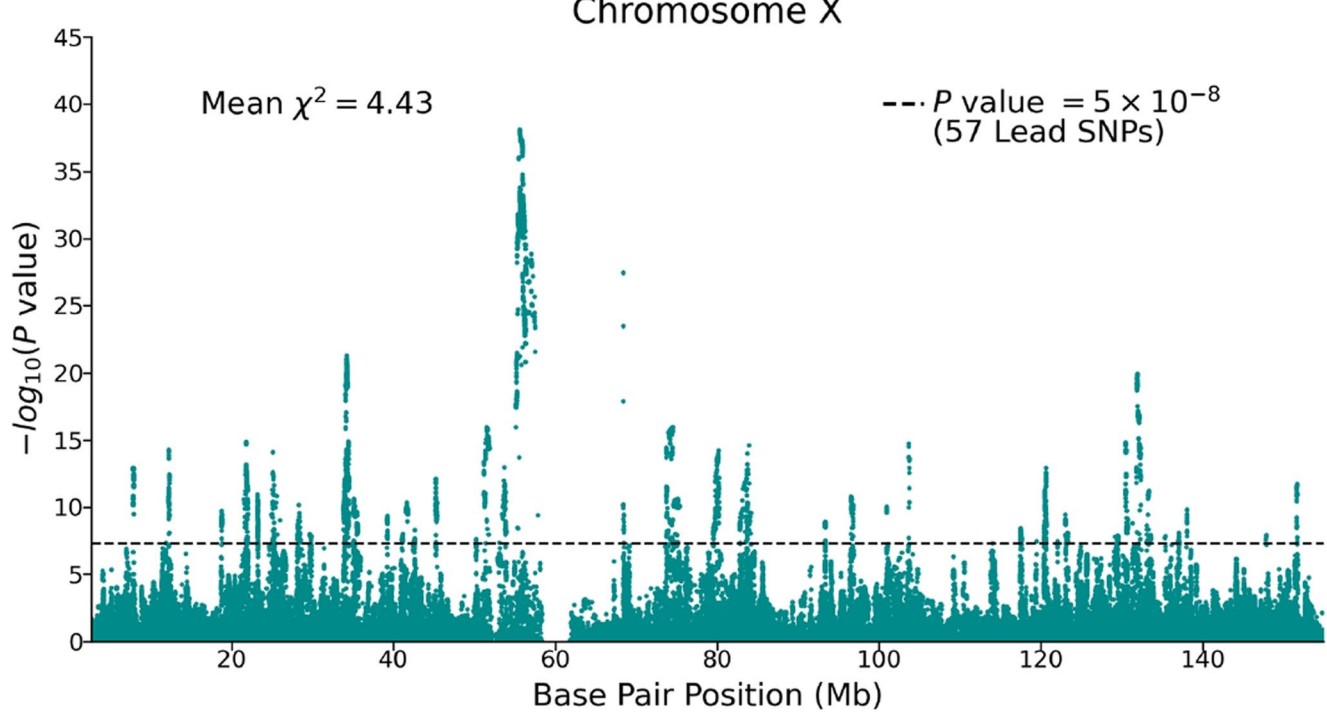

**b**

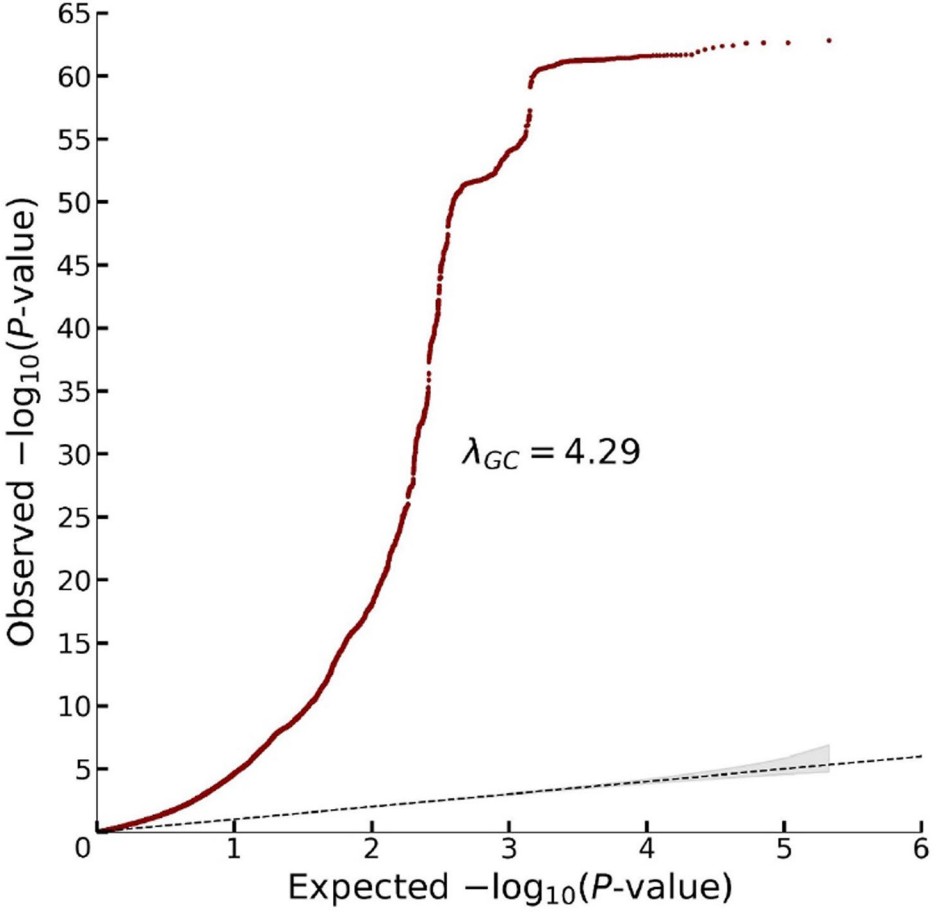

**Extended Data Fig. 4 | See next page for caption.**

**Extended Data Fig. 4 | Meta-analysis of X chromosome SNPs ($N = 2,713,033$ individuals).** The meta-analysis was conducted by combining summary statistics from (pooled-sex) association analyses conducted in UK Biobank ($N = 440,817$ individuals) and 23andMe ($N = 2,272,216$ individuals); see Supplementary Note section 3.4 for details. Panel **(a)**: Manhattan plot, in which $P$ values are based on summary statistics adjusted for inflation using the LD score intercept estimated from an autosomal association analysis of UKB and 23andMe. The solid line indicates the threshold for genome-wide significance ($P = 5 \times 10^{-8}$ based on a two-sided $Z$-test adjusted for multiple comparisons). Panel **(b)**: Q-Q plot, in which $P$ values are based on unadjusted $Z$-test statistics. The dotted line represents the expected $-\log_{10}(P\text{-values})$ under the null hypothesis. The (barely visible) gray shaded area in represents the 95% confidence intervals under the null hypothesis.

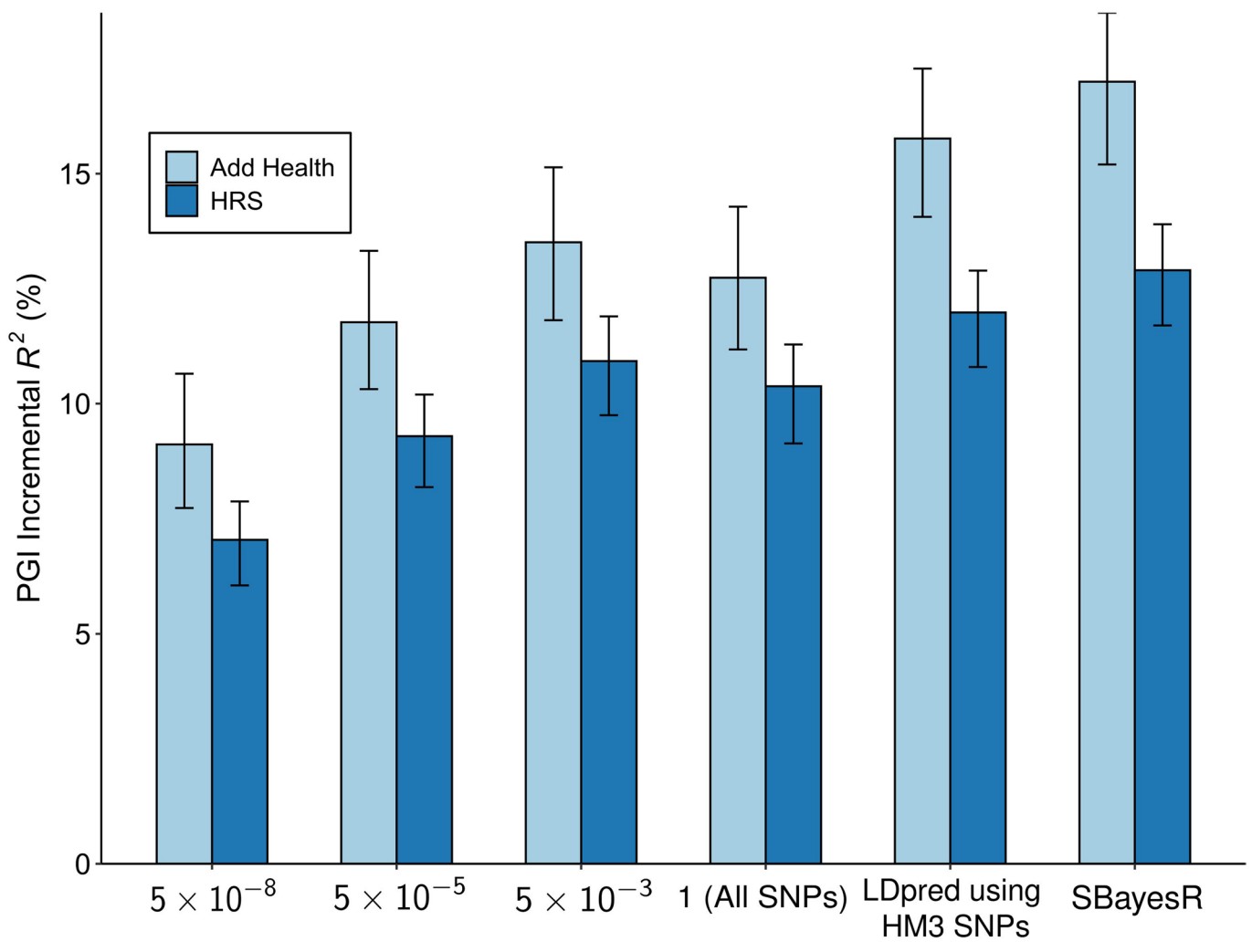

**Extended Data Fig. 5 | Predictive power of the EduYears PGI as a function of pruning at different *P* value thresholds.** Each bar represents the incremental $R^2$ with error bars showing the 95% confidence intervals bootstrapped with 1,000 iterations each. Each clumping and thresholding PGI is based on a set of approximately independent SNPs identified using the clumping algorithm defined in **Supplementary Note** section 2.2.6. For *HRS* ($N = 10,843$ individuals) and *Add Health* ($N = 5,653$ individuals) respectively, the number of SNPs included in the PGI is (with *P* value threshold in parentheses): 3,806 and 3,843 ($5 \times 10^{-8}$); 10,852 and 10,897 ($5 \times 10^{-5}$); 33,159 and 32,693 ($5 \times 10^{-3}$); 281,087 and 247,329 (1); 1,137,480 and 1,170,675 (All HapMap3 SNPs, LDpred); 2,540,570 and 2,548,339 (SBayesR). *P*-values are based on two-sided *Z*-tests. Incremental $R^2$ is the difference between the $R^2$ from a regression of *EduYears* on the PGI and the controls (sex, birth-year dummies, their interactions, and 10 PCs) and the $R^2$ from a regression of *EduYears* on just the controls.

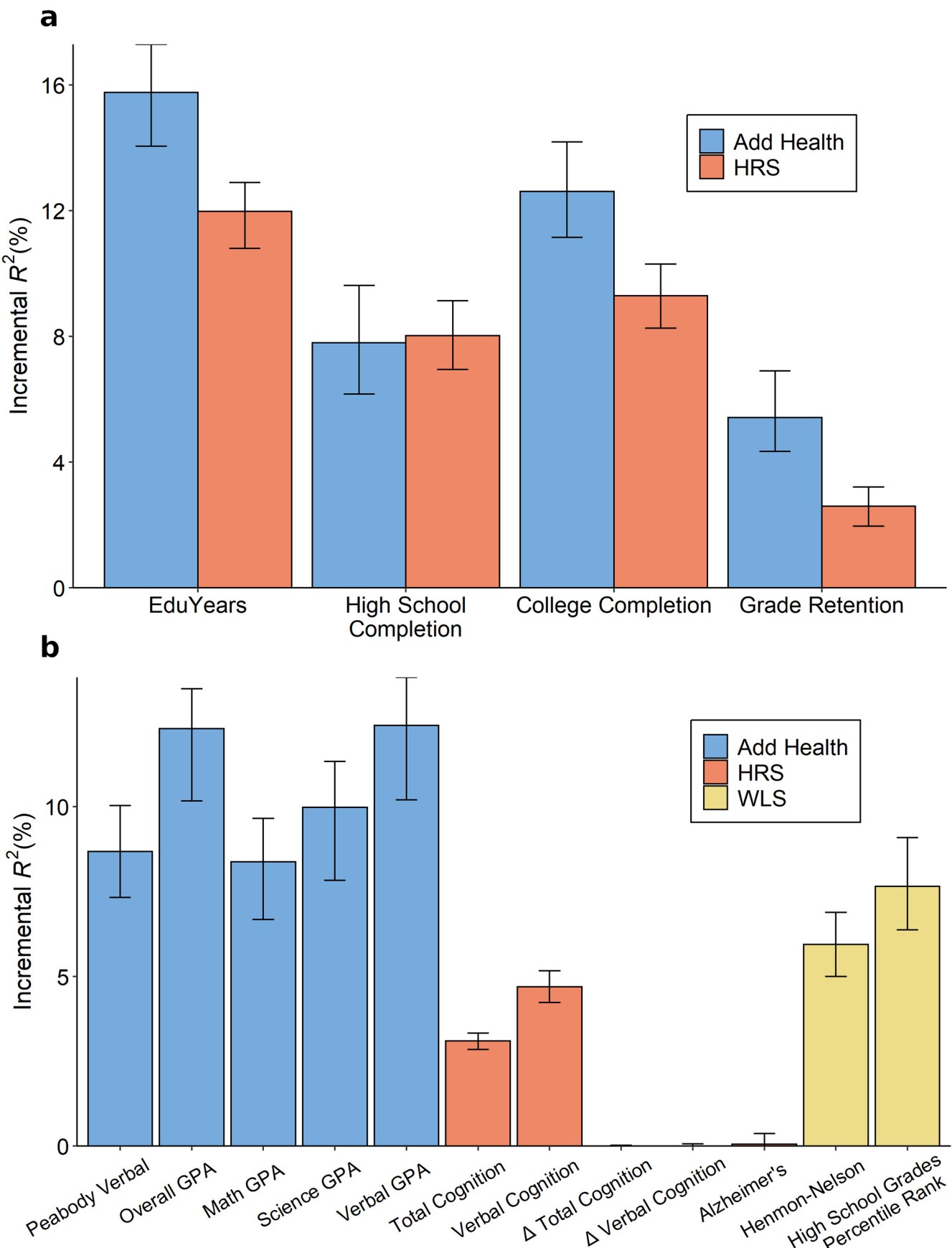

**Extended Data Fig. 6 | PGI prediction in Add Health, HRS and WLS.** Predictive power of the PGI constructed from the current *EduYears* GWAS results in three independent prediction cohorts: *Add Health* ($N = 5,653$), *HRS* ($N = 10,843$), and *WLS* ($N = 8,395$). For binary phenotypes, the y-axis is incremental Nagelkerke $R^2$. Panel **(a)**: Results for education phenotypes available in *Add Health* and *HRS*. Panel **(b)**: Results for cognitive and academic achievement phenotypes available in either *Add Health*, *HRS* or *WLS*. "Δ Total Cognition" and "Δ Verbal Cognition" are wave to wave changes in total and verbal cognition. In both panels, error bars show 95% confidence intervals for the incremental $R^2$, bootstrapped with 1000 iterations each. The number of individuals in the prediction sample for each regression can be found in Supplementary Table 4.

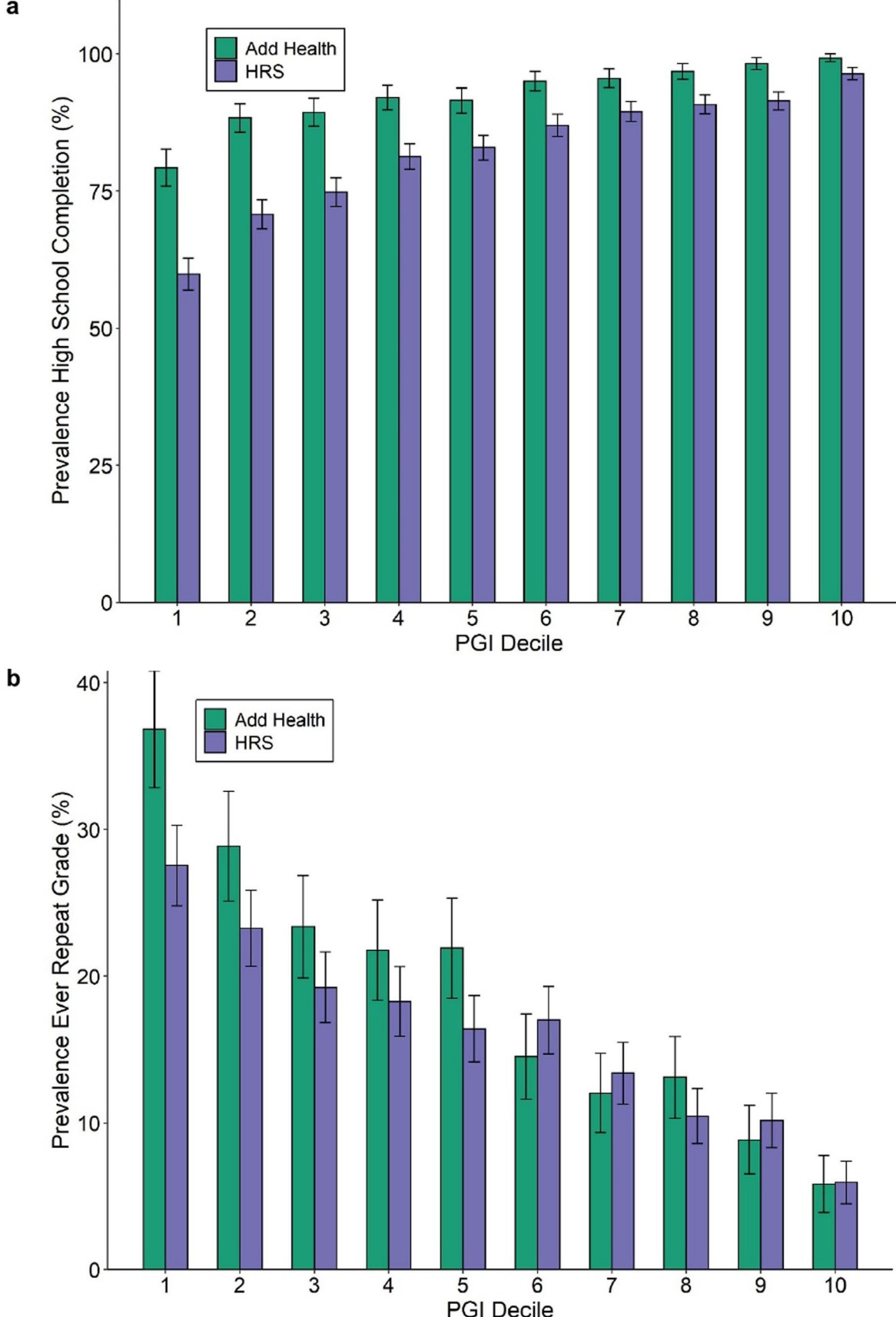

**Extended Data Fig. 7 | Prevalence of schooling outcomes by EduYears PGI decile.** Each decile contains approximately 1,085 respondents in *HRS* and 565 in *Add Health*. Total sample sizes for these phenotypes in each prediction cohort are in Supplementary Table 4. Decile 1 contains the lowest PGI values; decile 10, the highest. Error bars show 95% confidence intervals. Panel **(a)**: High school completion. Panel **(b)**: Grade retention.

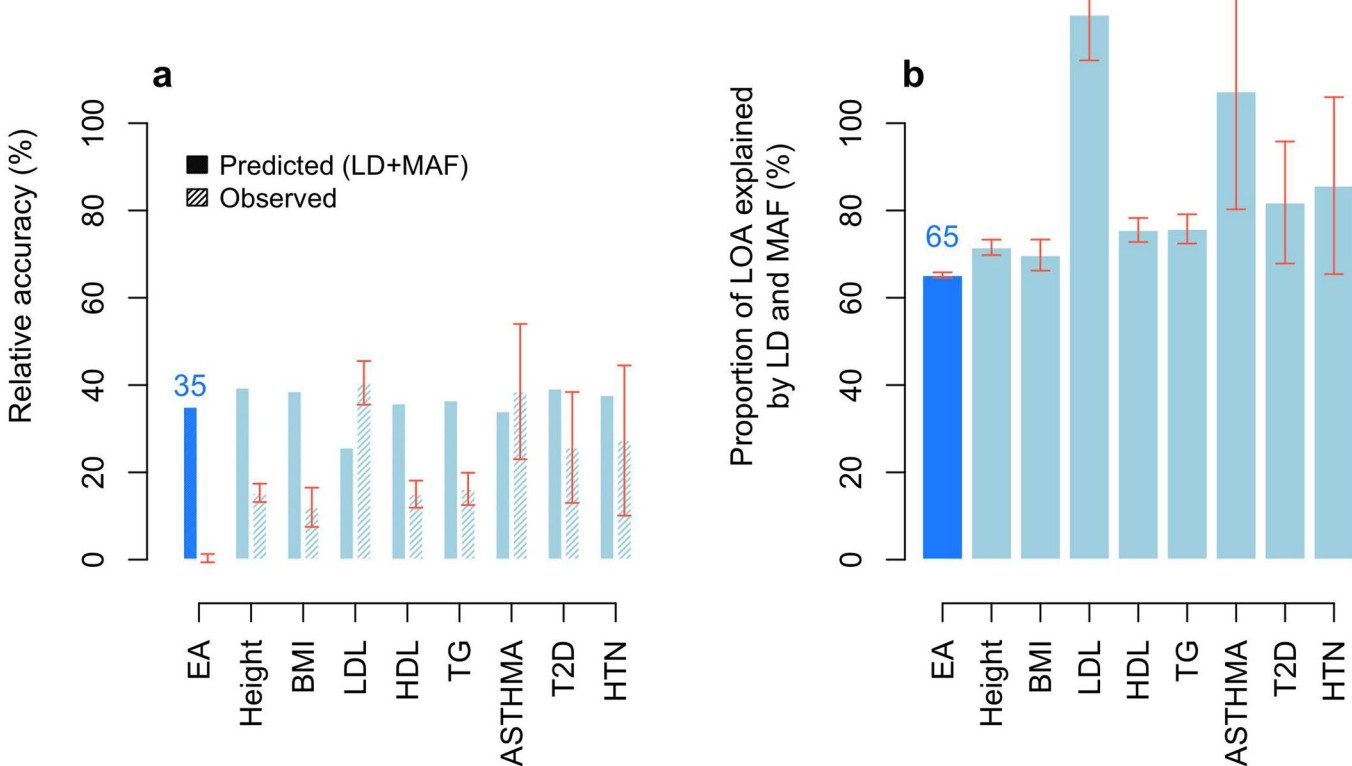

**Extended Data Fig. 8 | European genetic ancestries to African genetic ancestries relative accuracy.** Panel **(a)** plots the relative accuracy (RA) with error bars representing confidence intervals with $+/-1$ standard error. Panel **(b)** plots the proportion of the loss of accuracy (LOA) explained by LD and MAF calculated as $100\% \times (1 - RA_{pred(LD+MAF)})/(1 - RA_{obs})$ with error bars representing confidence intervals with $+/-1$ standard error. RA refers to the European genetic ancestries to African genetic ancestries ratio of prediction accuracies ($R^2$) of PGIs trained in a large sample of European-genetic-ancestry UKB participants ($N = 425,231$). The accuracy in European-genetic-ancestry participants was assessed in a holdout sample of 10,000 unrelated individuals, while the accuracy in African-genetic-ancestry participants was assessed in a holdout sample of 6,514 unrelated individuals. Phenotype labels: EA (Educational Attainment), Height (standing height), BMI (body mass index), LDL (low-density lipoprotein cholesterol), HDL (high-density lipoprotein cholesterol), TG (triglycerides), ASTHMA (diagnosed asthma), T2D (diagnosed type 2 diabetes) and HTN (diagnosed hypertension). See Supplementary Note section 7 in Wang et al. for additional details. Data underlying this Figure are reported in Supplementary Table 5.

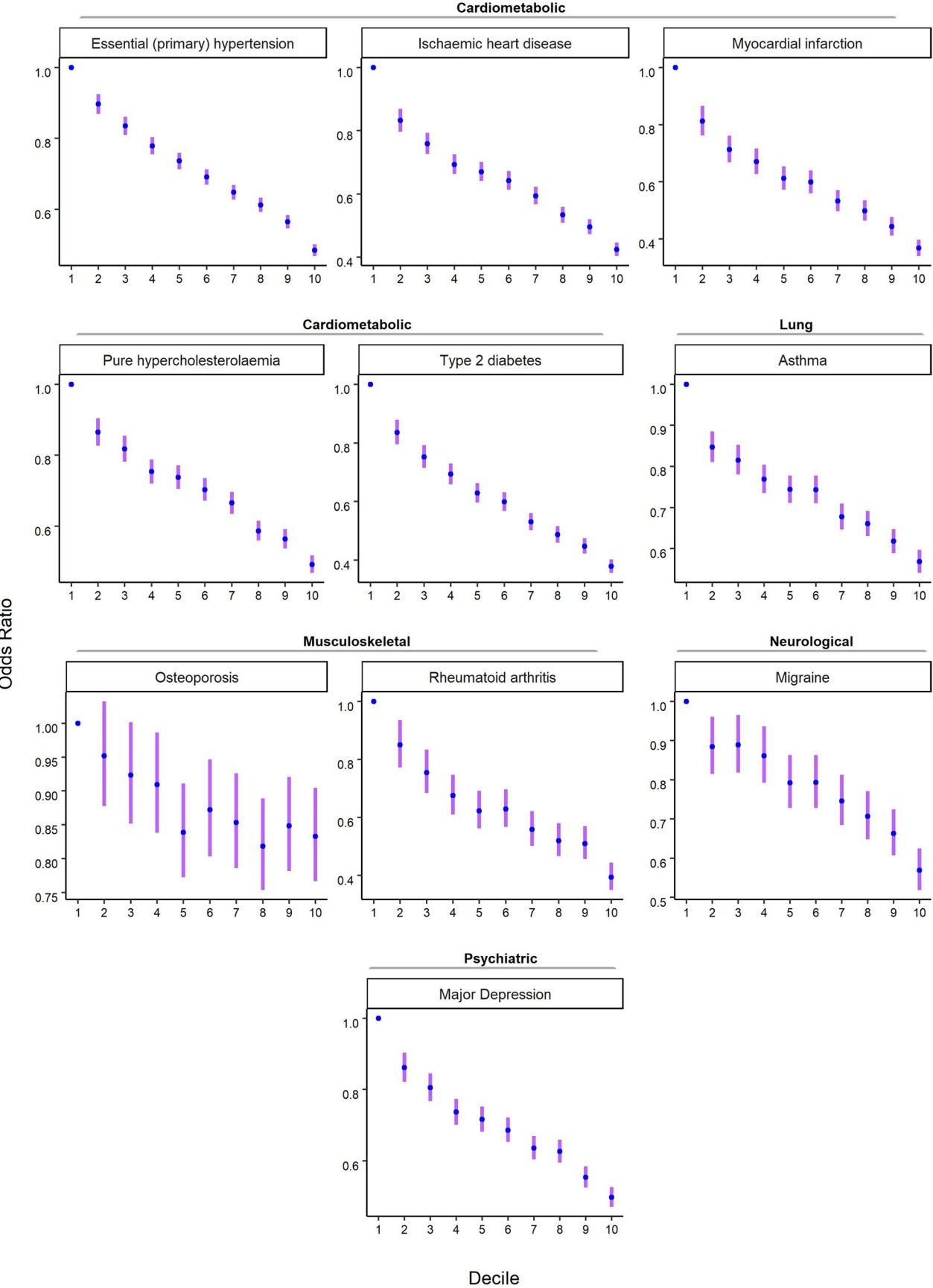

**Extended Data Fig. 9 | Odds ratio for selected diseases by deciles of the EA PGI in the UKB.** The EA PGI was discretized into deciles (1 = lowest, 10 = highest), and nine dummy variables were created to contrast each of deciles 2-10 to decile 1 as the reference. Odds ratio and 95% confidence intervals (the error bars) were estimated using logistic regression while controlling for covariates (sex, a third-degree polynomial in birth year and interactions with sex, the top 40 PCs, and batch dummies).

# nature research

# Reporting Summary

Nature Research wishes to improve the reproducibility of the work that we publish. This form provides structure for consistency and transparency in reporting. For further information on Nature Research policies, see our Editorial Policies and the Editorial Policy Checklist.

## Statistics

For all statistical analyses, confirm that the following items are present in the figure legend, table legend, main text, or Methods section.

| n/a | Confirmed | |
|---|---|---|
| ☐ | ☒ | The exact sample size (*n*) for each experimental group/condition, given as a discrete number and unit of measurement |
| ☐ | ☒ | A statement on whether measurements were taken from distinct samples or whether the same sample was measured repeatedly |
| ☐ | ☒ | The statistical test(s) used AND whether they are one- or two-sided <br> *Only common tests should be described solely by name; describe more complex techniques in the Methods section.* |
| ☐ | ☒ | A description of all covariates tested |
| ☐ | ☒ | A description of any assumptions or corrections, such as tests of normality and adjustment for multiple comparisons |
| ☐ | ☒ | A full description of the statistical parameters including central tendency (e.g. means) or other basic estimates (e.g. regression coefficient) AND variation (e.g. standard deviation) or associated estimates of uncertainty (e.g. confidence intervals) |
| ☐ | ☒ | For null hypothesis testing, the test statistic (e.g. *F*, *t*, *r*) with confidence intervals, effect sizes, degrees of freedom and *P* value noted <br> *Give P values as exact values whenever suitable.* |
| ☐ | ☒ | For Bayesian analysis, information on the choice of priors and Markov chain Monte Carlo settings |
| ☒ | ☐ | For hierarchical and complex designs, identification of the appropriate level for tests and full reporting of outcomes |
| ☐ | ☒ | Estimates of effect sizes (e.g. Cohen's *d*, Pearson's *r*), indicating how they were calculated |

*Our web collection on statistics for biologists contains articles on many of the points above.*

## Software and code

Policy information about availability of computer code

| Data collection | No software was used for data collection. |
|---|---|
| Data analysis | The following software packages were used for data analysis: Python version 3.7.4 with packages pandas 0.25.1, scipy 1.3.1, numpy 1.17.2, matplotlib 3.1.1 and argparse 1.1 (https://anaconda.org); R version 4.0.3 with packages EasyQC 9.2, plotrix 3.7.8, tidyr 1.1.3 and readstata13 0.9.2, R version 3.6 (https://www.r-project.org); GCTA 1.93.2beta (https://yanglab.westlake.edu.cn/software/gcta/#Overview); GCTB 2.03 (https://cnsgenomics.com/software/gctb/#Overview); Stata 16.1 (https://www.stata.com); PLINK 1.9 (https://www.cog-genomics.org/plink/1.9); PLINK 2 (https://www.cog-genomics.org/plink/2.0); LDpred 1.0.11 (https://github.com/bvilhjal/ldpred); METAL release 2011-03-25 (https://genome.sph.umich.edu/wiki/METAL_Documentation); BOLT-LMM 2.3 (https://alkesgroup.broadinstitute.org/BOLT-LMM/BOLT-LMM_manual.html); LDSC 1.0.1 (https://github.com/bulik/ldsc); SNIPar (https://github.com/AlexTISYoung/SNIPar/tree/EA4). |

For manuscripts utilizing custom algorithms or software that are central to the research but not yet described in published literature, software must be made available to editors and reviewers. We strongly encourage code deposition in a community repository (e.g. GitHub). See the Nature Research guidelines for submitting code & software for further information.

## Data

Policy information about availability of data

All manuscripts must include a data availability statement. This statement should provide the following information, where applicable:

- Accession codes, unique identifiers, or web links for publicly available datasets
- A list of figures that have associated raw data
- A description of any restrictions on data availability

GWAS summary statistics can be downloaded from http://www.thessgac.org/data subject to a Terms of Use to ensure responsible use of the data. We provide association results for all SNPs that passed quality-control filters in autosomal, X chromosome, and dominance GWAS meta-analyses that excludes the research

# Field-specific reporting

Please select the one below that is the best fit for your research. If you are not sure, read the appropriate sections before making your selection.

☐ Life sciences   ☒ Behavioural & social sciences   ☐ Ecological, evolutionary & environmental sciences

For a reference copy of the document with all sections, see nature.com/documents/nr-reporting-summary-flat.pdf

# Behavioural & social sciences study design

All studies must disclose on these points even when the disclosure is negative.

| | |
|---|---|
| Study description | This is a genome-wide association study (GWAS) meta-analysis of educational attainment (EA) in a sample of ~3 million individuals. All data used in this study (genetic and phenotype data) are quantitative. |
| Research sample | The research sample consists of ~3 million individuals from 71 research cohorts. We meta-analyzed three sets of summary statistics: publicly available results from Lee et al. (2018) that exclude 23andMe and UKB (N = 324,162), new association results from 23andMe (N = 2,272,216), and new association results from a GWAS we conducted in UKB with an improved coding of the EA measure (N = 441,121).   The large study sample was required for us to have sufficient statistical power in detecting single nucleotide polymorphisms (SNPs) with small effect sizes and for our follow-up analyses. |
| Sampling strategy | We obtained the largest sample we could. |
| Data collection | Data collection was performed independently by each participating cohort. |
| Timing | Data was collected from multiple cohorts with variable data collection periods. |
| Data exclusions | All observations reporting less than seven years of schooling were dropped to exclude outliers (there were fewer than 50 such observations; see Supplementary Note 1.1.4). |
| Non-participation | No participants dropped out or declined participation. |
| Randomization | Participants were not allocated into experimental groups. |

# Reporting for specific materials, systems and methods

We require information from authors about some types of materials, experimental systems and methods used in many studies. Here, indicate whether each material, system or method listed is relevant to your study. If you are not sure if a list item applies to your research, read the appropriate section before selecting a response.

## Materials & experimental systems

| n/a | Involved in the study |
|---|---|
| ☒ | Antibodies |
| ☒ | Eukaryotic cell lines |
| ☒ | Palaeontology and archaeology |
| ☒ | Animals and other organisms |
| ☐ | ☒ Human research participants |
| ☒ | Clinical data |
| ☒ | Dual use research of concern |

## Methods

| n/a | Involved in the study |
|---|---|
| ☒ | ChIP-seq |
| ☒ | Flow cytometry |
| ☒ | MRI-based neuroimaging |

# Human research participants

Policy information about studies involving human research participants

| | |
|---|---|
| Population characteristics | See above. |

| Recruitment | Recruitment strategies were particular to each cohort. |
| Ethics oversight | All analyses are on anonymized, secondary data. Nonetheless, the analyses reported in the paper fall under National Bureau of Economic Research IRB protocols 19_434, 19_465, and 20_041. |

Note that full information on the approval of the study protocol must also be provided in the manuscript.

