## [Peer Review File · Nature Genetics]

Peer Review Information

Manuscript Title: Polygenic prediction of educational attainment within and between families from genome-wide association analyses in 3 million individuals

Corresponding author name(s): Professor Daniel Benjamin

Reviewer Comments & Decisions:

Decision Letter, initial version:
--

16th Jun 2021

Dear Professor Benjamin,

Your Article, "Polygenic prediction within and between families from a 3-million-person GWAS of educational attainment" has now been seen by 3 referees. You will see from their comments below that while they find your work of interest, some important points are raised. We are interested in the possibility of publishing your study in Nature Genetics, but would like to consider your response to these concerns in the form of a revised manuscript before we make a final decision on publication.

To guide the scope of the revisions, the editors discuss the referee reports in detail within the team, with a view to identifying key priorities that should be addressed in revision. As you will see from these comments, all referees have identified aspects of the analyses and the discussion that need to be improved. Some of the claims need to be better supported by additional analyses. Please address all referees' points as thoroughly as possible.

We therefore invite you to revise your manuscript taking into account all reviewer and editor comments. Please highlight all changes in the manuscript text file. At this stage we will need you to upload a copy of the manuscript in MS Word .docx or similar editable format.

*2) If you have not done so already please begin to revise your manuscript so that it conforms to our Article format instructions, available [here](http://www.nature.com/ng/authors/article_types/index.html). Refer also to any guidelines provided in this letter.

[REDACTED]

We hope to receive your revised manuscript within three to six months. If you cannot send it within this time, please let us know.

Sincerely,

Wei Li, PhD
Senior Editor

Nature Genetics
One New York Plaza, 47th Fl.
New York, NY 10004, USA
www.nature.com/ng

Reviewers' Comments:

Reviewer #1:

Remarks to the Author:

This 4th iteration of the GWAS of educational attainment by the SSGAC is a significant update to the previous version mainly in sample size. There are some incremental gains in variant discovery, polygenic prediction, and knowledge regarding the genetic architecture of educational attainment. Further refinement of direct/indirect components of polygenic score associations is a contribution. However, two significant issues in GWAS of educational attainment are only partly addressed. The data are there. But more interpretation is needed to realize the potential for impact.

The first issue that requires more attention is the much smaller effect size for the polygenic score in Black vs. White HRS participants. The difference in R-squared is roughly an order of magnitude (PDF page 7, para 3). Two obvious potential sets of causes of this difference are (1) measurement error arising from differences in LD between GWAS SNPs and causal variants in the HRS Black sample vs. the White sample; and (2) gene-environment interactions arising from social-environmental constraints on phenotypic variance that are different in the HRS Black sample vs. the White sample. In past iterations of this GWAS, the authors could be forgiven for navigating around the thorny question of the relative contributions of these (and perhaps other) sources of low portability. But now it's time to take the question seriously. If these authors want to continue advancing the cause of genetic prediction algorithms for human socioeconomic attainments—and validating them in US samples, no less—they are going to have to confront the question of why their prediction algorithms work so poorly in a specific segment of the population that faces major barriers to socioeconomic attainment. I don't think the authors have to provide a definitive answer. But they have to at least frame the possibilities and offer us some kind of analysis.

Three approaches that could be taken are:

First, the prediction cohorts have been used by other researchers (and some of these authors) to study social forces that constrain attainments of Black vs. White Americans. Pick some of these dimensions and show us how R² varies across them within the Black samples of HRS and Add Health. Birth cohort, region of birth, as well as more granular features of the environment are all available for analysis.

Second, there is certainly heterogeneity in the extent of European ancestry within the samples of Black Americans. If portability is all about measurement error, then R² should scale with genetic similarity of the target to the discovery population. Does it?

Third, this group has expended significant effort projecting what gains in polygenic prediction R² should look like as GWAS sample sizes increase. An important observation in recent GWAS is that R²

gains in Euro-descent samples are somewhat less than expected. What about in non-Euro descent samples? (and how does this compare to polygenic scores for stature or BMI?)

The second issue is easier to deal with and has to do with the question of what traits genetic correlates of educational attainment influence on their way to producing differences in years of schooling. Since educational attainment is not a trait, per se, this question has always been in the background. Cognitive abilities are certainly a leading candidate. However, it is notable here that polygenic prediction R² values for measurements of cognitive abilities are roughly ½ of those for educational attainment phenotypes. A second, more striking observation is that spousal assortment on the polygenic score is not well explained by phenotypic assortment on education and cognition, nor is it explained by assortment on genetic ancestry features (pdf page 9 para 3). The implication of this observation is that GWAS of education are turning up genetic signal for other traits. What might these be? There have been several papers on the network of phenotypes linked with the education polygenic score beyond cognition and education. What do these papers suggest?

A few other, more minor comments:

The treatment of dominance effects in the main text was surprisingly extensive. Given the main result is that there is little evidence for dominance effects, this might be shifted to the supplement to make space for the issues raised above.

In a twin model, dominance is not distinguished from epistasis. Given the sample size, analysis of epistasis seems possible. At least the authors should offer some explanation of why they explore only dominance here.

It would be helpful to provide some discussion of how the within-sib GWAS analysis reported here relates to the one reported by Howe et al. (2021 BioRxiv), on which a number of these authors are also listed as coauthors.

Reviewer #2:

Remarks to the Author:

In the manuscript "Polygenic prediction within and between families from a 3-million-person GWAS of educational attainment" Okbay et al. investigate the genetics of educational attainment (EA). The authors QC-ed and analyzed the data with care. They use replication samples which validate the initial results. They did a good job of looking at possible confounders including stratification, assortative mating etc. This paper makes an important contribution to the genetics of EA and its part in the ongoing discussion about the "nature of nurture". But we raise a few moderate issues and several minor issues.

Moderate and more major points:

First, the authors provide very few, if any, comments on what is a possible route for the indirect effect of EA on seemingly unrelated traits. We believe that the authors can at least work in the literature that shows that low SES is a significant risk factor in numerous diseases.

Second, LDscore intercept is notably larger than what we are used to seeing. I think this deserves more comments on the possible causes and comparisons with other GWAS (including some of the

previous EA GWAS) in terms of lambda 1000.

Third, while they must have dealt with this elsewhere, is it true that all their data lists total years of education or do they get clumped data (8-11 years, finished high school, some college, graduated college, etc.). If the latter, how do they deal with different "clumping" of YOE where the meaning of the thresholds likely change between samples and across birth cohorts? One of us has fitted multiple threshold models in pairs of relatives to such clumped YOE data in a large longitudinal sample covering 4 decades where YOE was rising rapidly. The model fitted terribly indicating that the "meaning" of the different categories was not constant across birth cohorts. How do they treat "technical schools" etc? We searched and were unable to find a discussion of these concerns in their material.

Fourth, given that parental PGIs are the (equal weighted) sum of transmitted and untransmitted alleles, can they be more specific about how their "controlling for parental PGI" is or is not equivalent to transmitted vs untransmitted analyses of Kong et al?

Fifth, should they be discussing "assortative mating" or "spousal resemblance?" Do they have data on spousal correlations at marriage and if not, how can they justify their title? In their analyses of the causes of the high correlation, they don't seem to consider "spousal interaction", that is that one individual's EA could directly impact on the EA of their spouse by providing financial support, encouragement, etc. Doesn't that possibility merit discussion?

Minor issues

On line 162 "relatively weak enrichment" should probably read better as "weaker enrichment". Re: genes highly expressed in glial cells, what (MSigDB) pathway/gene sets they tend to significantly load on?

They should explain in their legend all abbreviations used in their tables.

What exactly is "social homogamy based on genetic relatedness"? That deserves some explanation in the text.

They write "lead SNPs corresponds to 1.4 weeks of schooling per allele." While cute, this is hardly a sustainable or sensible interpretation.

They write "Our findings are fully consistent with earlier conclusions: SNP heritability due to the X chromosome of 0.4% and (using sex stratified association analyses in the UK Biobank) a male-female genetic correlation close to unity ($r = 0.94$)." Is this correlation just for the X-chromosome or the entire genome?

In figure 3 they use the term "depression." It appears they mean major depression. It should be listed as such as generic "depression" can mean a variety of different things – e.g. self-report current symptoms above some cut-point.

There might not be room, but can they say something about the expected gain in predictive power as a function of sample size. Should it be linear as a function of log to base 10 sample size?

Reviewer #3:

Remarks to the Author:

The manuscript by Okbay et al. represents the largest GWAS effort of educational attainment (EA), with a 3-million predominantly European ancestry subjects. They reported 3,952 approximately uncorrelated genome-wide-significant SNPs. The polygenic score or polygenic index (PGI) explains 12-16% of EA variation and also other diseases. These results represent the increment advance of the previous study conducted by the same group (ref 2) and should be insightful for other large GWAS. The manuscript is well written. However, it is a little disappointed about how much biology we can learn from this large GWAS effort. Some statistical analyses may need careful investigation.

Comments:

- 1) The authors isolated the direct and indirect effects of the PGI in the population effect by controlling for both parents' PGIs in a regression model. They estimated 30.9% of the EA direct effect, which is significantly smaller than the estimate of 48.9% reported in the Icelandic data (Ref 4). In Discussion, they claimed that "much of the predictive power of the EA PGI is not explained by direct effect" (Page 10). However, it seems the regression approach used in this study is different from the previous study. Since parents transmit 50% of their genomes to offspring, adjusting for parents' PGIs likely underestimates the direct effects in this study. Thus, this claim seems unconvinced. It would be more reasonable to apply similar analysis approach of Ref 4.
- 2) The authors performed assortative mating analysis and tried to answer how much PGI prediction power is due to assortative mating. They found that the observed spousal PGI correlation is substantially higher than the predicted PGI correlation, which is the product of the spousal phenotype correlation and both father and mother's phenotype and PGI correlations. In Supplementary Table 12, father and mother's phenotype and PGI correlations are missed. There are no standard errors of predicted PGI correlations listed. By controlling the ancestry, the spousal PGI correlation is much reduced. Furthermore, it is not clear why the spousal phenotype correlation reported in literature is much higher than that in the current study (0.412). Therefore, the predicted PGI correlation may be underestimated.
- 3) In the assortative mating analysis, the residuals of the father's and mother's PGIs were calculated after regression on their top 20 genetic principal components. Are the top 20 genetic principal components sufficient to control the effect by population stratification?
- 4) The PGI was calculated by GWAS summary statistics of both males and females. However, it is known that gender is associated with EA, also year of birth. Have gender and year of birth and their interaction been including in the analysis? It seems more reasonable to calculate male and female specific PGIs by using male and female specific summary statistics. Then, the correlation between EA and father's PGI can be less biased than that using father's PGI calculated using the summary statistics of males and females combined. This may be another reason of underestimating predicted spousal PGI correlation. The conclusion that "the spousal PGI correlation is far too strong to be consistent with assortative mating purely on phenotype" may need additional analysis.
- 5) Page 10, line 348. What does gene-environment correlation mean? Does it suggest gene-environment interaction or something else?
- 6) Reference 2 and 8 are the identical.

Author Rebuttal to Initial comments

Decision Letter

16th June 2021

Dear Professor Benjamin,

Your Article, "Polygenic prediction within and between families from a 3-million-person GWAS of educational attainment" has now been seen by 3 referees. You will see from their comments below that while they find your work of interest, some important points are raised. We are interested in the possibility of publishing your study in Nature Genetics, but would like to consider your response to these concerns in the form of a revised manuscript before we make a final decision on publication.

To guide the scope of the revisions, the editors discuss the referee reports in detail within the team, with a view to identifying key priorities that should be addressed in revision. As you will see from these comments, all referees have identified aspects of the analyses and the discussion that need to be improved. Some of the claims need to be better supported by additional analyses. Please address all referees' points as thoroughly as possible.

We therefore invite you to revise your manuscript taking into account all reviewer and editor comments. Please highlight all changes in the manuscript text file. At this stage we will need you to upload a copy of the manuscript in MS Word .docx or similar editable format.

...

Sincerely,

Wei Li, PhD
Senior Editor
Nature Genetics

Response Documents

1. Cover Letter
2. Response to R1.
3. Response to R2.
4. Response to R3.

Cover Letter

Dear Dr. Li,

Please find enclosed our resubmission of “Polygenic prediction within and between families from a 3-million-person GWAS of educational attainment.” We believe that we have addressed the most important issues raised by the referees (as well as a number of more minor issues). As you noted, some of the referees’ comments are somewhat open-ended/exploratory. The revised manuscript reports the results from several new analyses of issues raised in the referee reports.

The main changes we made to the manuscript in response to the referees’ comments are as follows.

First, in response to R1’s major comment, we now report calculations of the predictive power of the polygenic index in African genetic ancestry individuals that would be expected if the reduction relative to European genetic ancestry individuals were entirely driven by allele frequency and linkage disequilibrium differences between populations. The observed reduction in predictive power is substantially greater than what we calculate. Therefore, we conclude that differences in heritability or differences in the true genetic associations across the populations, perhaps due to gene-environment interactions, likely contribute to the reduction in predictive power.

Second, in response to R2’s major comment, when we first discuss how we partition the polygenic index’s predictive power into direct and other effects, we now mention several pathways (namely, parents’ education, socioeconomic status, and behavior) by which indirect parental effects could matter for childrens’ educational attainment.

Third, in response to a comment by R3, in our assortative mating analyses we have added a further control for population stratification and geography-based social homogamy by residualizing each mate pair’s polygenic index on birth coordinates and assessment center in the UK Biobank. We

found a reduced correlation between mate pairs' polygenic indices when adding further controls for these geographic factors, indicating that geographic factors not well captured by top genetic principal components likely contribute to the high correlation between mate pairs' educational attainment polygenic indices.

In addition to these main changes, we have also made a number of additional, more minor edits, most importantly:

- All referees raised questions about how our within-family analyses related to analyses in existing papers (Kong *et al.* (2018) and Howe *et al.* (2021)). We have added brief explanations in the relevant section of the main text explaining the connections between the various analyses, and we now provide more details on the connection to Kong *et al.* (2018) in the Supplementary Note.
- Although not requested by referees, we have made two changes in terminology to avoid potential confusion. First, we have replaced “ancestry” with “genetic ancestry” to reflect the fact that we refer to genetic (rather than self-reported) ancestry throughout the paper. Secondly, in our assortative mating analyses we have replaced “spouses” and “spousal” with the terms “mate pairs” and “mate pair,” respectively, because the inclusion of these pairs in our analysis was not conditional on marital status (but instead based on jointly having a genotyped biological child in the sample).
- In response to a comment by R3, we have added confidence intervals to our predicted correlation between mate pairs' PGIs, computed via bootstrap. Doing so makes it clear that the observed correlation between mate pairs' EA PGIs is statistically distinguishable from the correlation predicted under the model of phenotypic assortment.
- In the original submission, we made an error in reporting the number of mate pairs that we used in our mate pair correlation analyses. Specifically, we double-counted genotyped mate pairs who had more than one genotyped offspring. This primarily affected the analysis in Generation Scotland, where many genotyped parent-sib quads are available. We have corrected this in the revised manuscript, revising down the number of genotyped spouse pairs in UKB from 894 to 862 and in Generation Scotland from 2964 to 1603.

Correcting this error has resulted in slightly larger standard errors than reported in the original submission, but has not changed the point estimates much nor the conclusions of the analysis.

- Although not requested by referees, we explored the predictive power of SBayesR, a newer method of constructing polygenic indexes. Since these polygenic indexes have higher predictive power than our benchmark polygenic indexes (constructed using LDpred), we now report those results as ex post analyses.
- Also not requested by referees, but because it may be of use in future GWAS as well powered as ours, we conducted further analyses comparing COJO versus clumping definitions of lead SNPs. These are reported in the Supplementary Note.

For each of the three referee reports, we have enclosed point-by-point responses.

Because almost all of the referee comments asked us to add additional analyses, the resulting revised manuscript is roughly 10% longer than the 4,000-word maximum for an Article. In addition to reducing the amount of space devoted to the dominance GWAS, we tried to trim the text wherever we could without adversely affecting the substance of the paper. Of course, if the 4,000-word maximum is a hard limit, we will find further ways to cut down the length.

Note that we have also added to the Supplementary Note a Frequency Asked Questions (FAQ) that explains, in a less technical way than the paper itself, what the paper finds and how the results should—and should not—be interpreted. It is standard practice for major papers by the Social Science Genetic Association Consortium to be accompanied by such a FAQ, which we view as particularly important for work on the genetics of behavior, given the potential sensitivity of the research. In the past, these FAQs were posted online along with the paper (including here: <https://www.thessgac.org/faqs>). While we intend to continue to post the FAQs online, we have received the feedback that in order to make sure a FAQ is not overlooked, we should also make it available along with the paper on the journal's website. In order to make that possible, we are including the FAQ as a section of the Supplementary Note.

Thank you for considering our manuscript.

Sincerely,

Daniel Benjamin on behalf of the authors

Response to Reviewer #1:

Remarks to the Author:

This 4th iteration of the GWAS of educational attainment by the SSGAC is a significant update to the previous version mainly in sample size. There are some incremental gains in variant discovery, polygenic prediction, and knowledge regarding the genetic architecture of educational attainment. Further refinement of direct/indirect components of polygenic score associations is a contribution. However, two significant issues in GWAS of educational attainment are only partly addressed. The data are there. But more interpretation is needed to realize the potential for impact.

We thank the referee for the positive remarks.

The first issue that requires more attention is the much smaller effect size for the polygenic score in Black vs. White HRS participants. The difference in R-squared is roughly an order of magnitude (PDF page 7, para 3). Two obvious potential sets of causes of this difference are (1) measurement error arising from differences in LD between GWAS SNPs and causal variants in the HRS Black sample vs. the White sample; and (2) gene-environment interactions arising from social-environmental constraints on phenotypic variance that are different in the HRS Black sample vs. the White sample. In past iterations of this GWAS, the authors could be forgiven for navigating around the thorny question of the relative contributions of these (and perhaps other) sources of low portability. But now it's time to take the question seriously. If these authors want to continue advancing the cause of genetic prediction algorithms for human socioeconomic attainments—and validating them in US samples, no less—they are going to have to confront the question of why their prediction algorithms work so poorly in a specific segment of the population that faces major barriers to socioeconomic attainment. I don't think the authors have to provide a definitive answer. But they have to at least frame the possibilities and offer us some kind of analysis.

Three approaches that could be taken are:

First, the prediction cohorts have been used by other researchers (and some of these authors) to study social forces that constrain attainments of Black vs. White Americans. Pick some of these dimensions and show us how R^2 varies across them within the Black samples of HRS and Add Health. Birth cohort, region of birth, as well as more granular features of the environment are all available for analysis.

Second, there is certainly heterogeneity in the extent of European ancestry within the samples of Black Americans. If portability is all about measurement error, then R^2 should scale with genetic similarity of the target to the discovery population. Does it?

Third, this group has expended significant effort projecting what gains in polygenic prediction R^2 should look like as GWAS sample sizes increase. An important observation in recent GWAS is that R^2 gains in Euro-descent samples are somewhat less than expected. What about in non-Euro descent samples? (and how does this compare to polygenic scores for stature or BMI?)

We agree that it is important to understand why predictive power tends to decline when the training and target populations differ in terms of genetic ancestry (and that the very large reductions in accuracy typically found in African genetic-ancestry validation samples seriously limit the utility of currently available PGIs). We appreciate the three concrete and constructive suggestions for alternative routes one could try to advance our understanding of these issues. We are also grateful to the referee for helpfully framing the question of limited portability in terms of two competing hypotheses: (1) differences in allele frequencies and linkage disequilibrium (LD) between populations, or (2) differences in environments, leading to differences in heritability between populations or differences in causal effects due to gene-environment interactions. (We note that gene-gene interactions (epistasis) could also generate differences in marginal causal effects between genetic ancestries, but recent findings suggest that epistatic genetic variance contributes little to total (broad sense) heritability of human complex traits (Hivert *et al.*, 2021)).

We interpreted the referee’s list as ideas about possible ways to shed light on these hypotheses, rather than as directives. Ultimately, we decided to conduct a number of analyses in the spirit of the second and third suggestions on the referee’s list. Our basic approach is to estimate the predictive power observed in individuals of African genetic ancestries and compare it to the predictions of a model that was developed to quantify the expected reduction in accuracy if the only factors reducing accuracy are differences in allele frequency and LD. In the revised manuscript, we report these calculations, which are based on theory developed by Wang *et al.* (2020):

PGIs like ours that are constructed from GWAS in samples of European genetic ancestries are generally found to have much lower predictive power in samples with other ancestries; for example, on average across phenotypes, estimates of relative accuracy (ratio of R^2) in African-genetic-ancestry to European-genetic-ancestry samples have been 22% (Martin *et al.*, 2019) and 36% (Duncan *et al.*, 2019). When we used our PGI to predict *EduYears* in samples with African genetic ancestries from the HRS ($N = 2,507$) and Add Health ($N = 1,716$), the incremental R^2 was 1.3% (95% CI: 0.6% to 2.2%) and 2.3% (95% CI: 1.1% to 3.7%), implying that the relative accuracies for EA in the HRS and Add Health are only 11% and 15%, respectively. Using the UKB, we find that the relative accuracy is smaller than would be predicted based on population differences in allele frequencies and LD alone (**Online Methods**), and this discrepancy is greater for EA than has been found in prior work (Wang *et al.*, 2020) for height, BMI, and six other phenotypes (**Supplementary Figure 8** and **Supplementary Table 5**). The remaining reduction in predictive power is due to factors including epistasis (although epistatic variance is likely small (Hill, Goddard and Visscher, 2008; Hivert *et al.*, 2021)), gene-environment interactions, and differences between populations in gene-environment correlations, assortative mating, and environmental variance.

We think our findings provide clear evidence that allele frequency and LD differences are important sources of non-portability, but that it is unlikely that these factors alone can fully account for the reductions in prediction accuracy that tend to be observed empirically.

We describe these analyses in a new section of the Supplementary Note (5.7). Per the referee's suggestion, the revised manuscript also includes a new Supplementary Figure 8 and Supplementary Table 5 that compare the observed and predicted losses in prediction accuracy for EA to those for the eight phenotypes examined by Wang et al. (which did not include EA). We believe these novel analyses have strengthened the paper and thank the referee for suggesting them.

While we also think it would be interesting to pursue some of the other avenues proposed by the referee, we have elected not to do so here. In most cases, the issue is data constraints. For example, we considered comparing SNP heritabilities across the European- and African-genetic-ancestry samples, but because the African genetic-ancestry sample available to us is small, the confidence interval for the SNP heritability estimate in the African-genetic-ancestry sample would be too wide to be informative. Similarly, we do not have access to a dataset that would enable well-powered analyses of how the predictive power of our PGIs varies across individuals of African genetic ancestries with different socioeconomic status. Of course, in the event that our paper is published, we anticipate that researchers with access to more suitable data sets will use our summary statistics to pursue these questions in greater depth.

The second issue is easier to deal with and has to do with the question of what traits genetic correlates of educational attainment influence on their way to producing differences in years of schooling. Since educational attainment is not a trait, per se, this question has always been in the background. Cognitive abilities are certainly a leading candidate. However, it is notable here that polygenic prediction R² values for measurements of cognitive abilities are roughly ½ of those for educational attainment phenotypes. A second, more striking observation is that spousal assortment on the polygenic score is not well explained by phenotypic assortment on education

and cognition, nor is it explained by assortment on genetic ancestry features (pdf page 9 para 3). The implication of this observation is that GWAS of education are turning up genetic signal for other traits. What might these be? There have been several papers on the network of phenotypes linked with the education polygenic score beyond cognition and education. What do these papers suggest?

We agree with the referee that the question of which traits explain the relationship between the EA PGI and EA has always been in the background, and several of our results—especially our findings on mate-pair assortment—make this question especially salient. In response to the referee’s comment, we undertook a literature review of the papers linking the EA PGI to phenotypes other than cognition and education. The most relevant papers we found are:

Reference	Phenotypes
de Zeeuw et al. (2014) https://onlinelibrary.wiley.com/doi/full/10.1002/ajmg.b.32254	attention deficit hyperactivity disorder (ADHD)
Belsky et al. (2016) https://journals.sagepub.com/doi/pdf/10.1177/0956797616643070	occupational choices, mobility, planfulness, self-control, interpersonal skills
Krapohl et al. (2016) https://www.nature.com/articles/mp2015126.pdf	behavioral problems
Marioni et al. (2016) https://www.pnas.org/content/pnas/113/47/13366.full.pdf	parental longevity
Möttus et al. (2017) https://journals.sagepub.com/doi/full/10.1177/0956797617719083	openness, neuroticism
Belsky et al. (2018)	social-class mobility

https://www.pnas.org/content/115/31/E7275	
Jansen et al. (2018) https://acamh.onlinelibrary.wiley.com/doi/abs/10.1111/jcpp.12759	behavioral problems
Niemi et al. (2018) https://www.nature.com/articles/s41586-018-0566-4	neurodevelopmental disorders
Wertz et al. (2018) https://journals.sagepub.com/doi/full/10.1177/0956797617744542	criminal record
Avinun (2019) https://www.biorxiv.org/content/10.1101/727552v2.full	SES, depression
Comes et al. (2019) https://www.nature.com/articles/s41398-019-0547-x	cognitive performance
Ding et al. (2019) https://www.sciencedirect.com/science/article/pii/S027795361930543X	cognitive decline
Ding, Barban and Mills (2019) https://www.sciencedirect.com/science/article/pii/S0091743519303421	allostatic load in later life
Elliott et al. (2019) https://academic.oup.com/cercor/article/29/8/3496/5095370	brain size
Huibregtse et al. (2021) https://academic.oup.com/psychsocgerontology/article/76/1/173/5541633	frailty in later life
Smith-Woolley, Selzam and Plomin (2019) https://psycnet.apa.org/fulltext/2019-16539-001.html	openness, conscientiousness, agreeableness
Verhoef et al. (2019)	ADHD

https://www.nature.com/articles/s41398-018-0324-2	
Zeng et al. (2019) https://pubmed.ncbi.nlm.nih.gov/31170283/	risk for coronary artery disease
Barth, Papageorge and Thom (2020) https://www.journals.uchicago.edu/doi/full/10.1086/705415	wealth accumulation, risk tolerance
Breinholt and Conley (2020) https://www.nber.org/papers/w28217	cognitively stimulating parenting behavior during early childhood
Judd et al. (2020) https://www.pnas.org/content/117/22/12411.short	cognitive and brain development
Mitchell et al. (2020) https://www.sciencedirect.com/science/article/pii/S1053811920301786	cortical measures
Papageorge and Thom (2020) https://academic.oup.com/jeea/article/18/3/1351/5677507?login=true	SES, labor earnings, non-routine analytic tasks
Salvatore et al. (2020) https://onlinelibrary.wiley.com/doi/abs/10.1111/add.14815	alcohol, nicotine and cannabis use disorders
Wertz et al. (2020) https://srcd.onlinelibrary.wiley.com/doi/full/10.1111/cde329	parenting
Bolyard and Savelyev (2021) https://papers.ssrn.com/sol3/Papers.cfm?abstract_id=3397735	health in young adulthood
Li et al. (2021) https://www.tandfonline.com/doi/full/10.1080/19485565.2020.1869919	obesity
Liu et al. (2021) https://link.springer.com/article/10.1007/s40865-021-00166-8	adolescent criminal justice involvement
Warrier et al. (2020)	autism

https://www.medrxiv.org/content/10.1101/2020.07.21.20159228	
---	--

v1	
--

Most plausibly, among the phenotypes we did not already mention in the paper, assortative mating on various personality traits and on socioeconomic status could contribute to the high mate-pair EA PGI correlation. Accordingly, following the referee’s suggestion, in the revised manuscript we have added to the Results section on assortative mating a more explicit mention of what these phenotypes may be:

This remainder is due to assortment on phenotypes correlated with the EA PGI other than EA, cognitive performance, and vocabulary—possibly including various personality traits (Belsky *et al.*, 2016; Möttus *et al.*, 2017; Smith-Woolley, Selzam and Plomin, 2019) —and sources of social homogamy other than genetic ancestry captured by the top 40 PCs—possibly including geographic location at courtship age (Abdellaoui *et al.*, 2019; Laidley, Vinneau and Boardman, 2019), socioeconomic status, and social class (Belsky *et al.*, 2018).

A few other, more minor comments:

The treatment of dominance effects in the main text was surprisingly extensive. Given the main result is that there is little evidence for dominance effects, this might be shifted to the supplement to make space for the issues raised above.

In the revised manuscript, we streamlined the discussion about the dominance results further, cutting several sentences. We believe we have reduced the length to the minimum necessary to convey what analyses we conducted and what their conclusions were.

Three main factors account for the originally submitted manuscript’s fairly extensive treatment of the dominance results:

- (i) dominance features prominently in many variance decompositions published in the behavior genetics (twin) literature;
- (ii) as far as we are aware, ours is the first large-scale GWAS of dominance deviations;
- (iii) the analysis plan was worked out before commencing the empirical data analysis.

While we think these are good reasons for not relegating all the discussion of our dominance GWAS to the Supplementary Note, we also agree with the referee that in the originally submitted manuscript, the treatment of dominance appeared disproportionate relative to the results. We hope the referee finds that we strike a more reasonable balance in the revised manuscript.

In a twin model, dominance is not distinguished from epistasis. Given the sample size, analysis of epistasis seems possible. At least the authors should offer some explanation of why they explore only dominance here.

We fully agree with the referee about the confounding between dominance and other non-additive genetic sources of variation in the classical twin design. In particular, dominance is fully confounded with additive-by-additive (AxA) effects in the classical twin design because both have a full-sibling correlation of $\frac{1}{4}$. In contrast, other components, such as additive-by-dominance and dominance-by-dominance, have smaller full-sibling correlations: $\frac{1}{8}$ and $\frac{1}{16}$, respectively. (See Table 7.2 in Lynch and Walsh (1998) for an overview of coefficients for different components of genetic covariance between various types of relatives that includes those referenced above.) Therefore, we focus our response on the detection of AxA effects and variance in GWAS.

There are two reasons we did not pursue AxA effects in our study. The first is lack of power of detection, both for specific SNP pairs and for overall variance explained by the AxA component. The second is logistical and computational constraints. We now elaborate on these issues.

Regarding power, existing theory and evidence indicates that for complex phenotypes, non-additive effects are likely to explain a much smaller fraction of variance than additive effects (Hill, Goddard and Visscher, 2008). Moreover, theory implies that when the effects of individual SNPs are smaller, the phenotype is likely to be more nearly additive (because dominance and AxA are both second-order effects)—leading us to expect these theoretical arguments to have substantial bite for EA. The theoretical reasons why AxA effects are expected to be small are similar to those for why dominance effects are expected to be small. Moreover, across 70 complex phenotypes in the UK Biobank (not including EA), Hivert *et al.* (2021) estimated that dominance and AxA effects both explain a small fraction of variance. Therefore, our prior was that AxA and dominance effects are likely to have small effect sizes on the same order of magnitude. However, holding constant the effect size, our power to detect AxA effects is far smaller than our power to detect dominance effects.

First, consider power for estimating the effects of specific SNP pairs. For detecting dominance effects, a p-value threshold of 5×10^{-8} can be justified on similar grounds as 5×10^{-8} for detecting additive effects, namely, as a Bonferroni-corrected significance threshold for a type-I error rate of 0.05 with ~ 1 million independent SNPs. Applying the same logic for detecting AxA effects, however, requires a much smaller p-value threshold; there are $\binom{10^6}{2} \approx 5 \times 10^{11}$ independent pairs of SNPs, implying a Bonferroni-corrected significance threshold of $\frac{0.05}{5 \times 10^{11}} = 10^{-13}$. Therefore, holding constant the effect size, we therefore expected to have far less power for detecting AxA effects than dominance effects.

Next, consider power for estimating the overall fraction of variance explained by AxA effects. To estimate this from SNP data, the most practical method is that of Hivert *et al.* (2021). Once again, power is much lower for estimating the fraction of variance explained by AxA effects than for that explained by dominance effects. Moreover, it is extremely unlikely that we could use data from 23andMe for this analysis (see below). Since we do not have access to the individual-level data from most of the cohorts in Okbay *et al.* (2016), our main sample available would therefore be the UK Biobank. This is the same sample analyzed by Hivert *et al.* (2021), who found that they were

underpowered for estimating the fraction of variance explained by AxA effects for any specific phenotype. Thus, we would almost surely also be underpowered for EA.

For generating the AxA summary statistics we would need for either type of analysis, we faced logistical and computational constraints. Both estimating the effects of specific SNP pairs and using Hivert *et al.*'s (2021) method of estimating the fraction of variance due to AxA effects would require a GWAS on AxA effects. With 10 million SNPs in a GWAS, an AxA GWAS requires $\binom{10^7}{2} \approx 5 \times 10^{13}$ statistical tests, which is like running a standard 10-million-SNP GWAS on 5 million phenotypes. Such a large GWAS in a sample the size of 23andMe's would require specialized hardware and software (e.g., based upon GPUs). 23andMe is generally reluctant to approve projects which require non-standard analyses and extensive computational resources. Thus 23andMe, which provided the largest sample size for the dominance and additive GWASs, would almost surely need to be excluded from these analyses. It would be an enormous effort to organize the ~70 cohorts other than 23andMe and UKB that contributed to Okbay *et al.* (2016) and Lee *et al.* (2018), and even if they had the analytical resources for the analyses, they would be reluctant to participate given the high cost and low expected payoff. We would therefore most likely need to rely solely on the UKB, which would compound the power challenges explained above.

In the revised manuscript, we now end the paper with: “even larger samples will enable other analyses that have not yet been adequately powered, such as estimating differences in SNP effect sizes across phenotypes or populations and estimating the fraction of variance explained by epistatic interactions (Hivert *et al.*, 2021).”

It would be helpful to provide some discussion of how the within-sib GWAS analysis reported here relates to the one reported by Howe *et al.* (2021 BioRxiv), on which a number of these authors are also listed as coauthors.

Howe *et al.* (2021) used siblings to estimate direct effects of individual SNPs, not PGIs. In this paper, we do not perform a within-sib GWAS: we use siblings and trios to estimate the direct and

population effects of PGIs, similar to previous analyses by Kong *et al.* (2018), Selzam *et al.* (2019), and Willoughby *et al.* (2019), as we now highlight in the first paragraph of the Results subsection on ‘Within-family Analyses’ in the revised manuscript. Our goal is to estimate how much of the PGI’s predictive power is due to direct effects. We also highlight that this is different from estimating the total contribution of the direct effects of genome-wide SNPs, for which summary statistics from within-family GWAS (such as by Howe *et al.* (2021)) could be used:

Our next set of analyses, like related prior work (Kong *et al.*, 2018; Selzam *et al.*, 2019; Willoughby *et al.*, 2019), aim to isolate the component of the PGI’s predictive power that is due to direct effects (Kong *et al.*, 2018; Walsh and Lynch, 2018): causal effects of an individual’s genetic material on that individual. When controls for both parents’ PGIs are included, we refer to the coefficient from a regression of an individual’s phenotype on the individual’s PGI as the *direct effect* of the PGI; when those controls are omitted, we refer to it as the *population effect*. (The regression controlling for parental PGIs gives an equivalent estimate of the direct effect of the PGI as a regression on PGIs constructed from transmitted and non-transmitted parental alleles (Kong *et al.*, 2018); see **Supplementary Note**.) The population effect captures the sum of the direct effect, indirect effects from relatives (e.g., genetic influences on parents’ education, socioeconomic status, and behavior), other gene-environment correlation (i.e., correlation between genotypes and environmental exposure, with population stratification being one possible cause), and a contribution from the genetic component of the phenotype that would be uncorrelated with the PGI under random mating but becomes correlated with the PGI due to the linkage disequilibrium between causal alleles induced by assortative mating (**Supplementary Note**) (Kong *et al.*, 2018; Howe *et al.*, 2021). Since the PGI is constructed from summary statistics that partly reflect indirect effects and other gene-environment correlation, estimating the direct effect of the PGI is different from estimating the total contribution of direct effects of SNPs (Trejo and Domingue, 2018; Fletcher *et al.*, 2021), for which relatedness

disequilibrium regression (Young *et al.*, 2018) or summary statistics from within-family GWAS (Howe *et al.*, 2021) could be used.

Response to Reviewer #2:

Remarks to the Author:

In the manuscript "Polygenic prediction within and between families from a 3-million-person GWAS of educational attainment" Okbay et al. investigate the genetics of educational attainment (EA). The authors QC-ed and analyzed the data with care. They use replication samples which validate the initial results. They did a good job of looking at possible confounders including stratification, assortative mating etc. This paper makes an important contribution to the genetics of EA and its part in the ongoing discussion about the "nature of nurture". But we raise a few moderate issues and several minor issues.

We thank the reviewer for the positive remarks.

Moderate and more major points:

First, the authors provide very few, if any, comments on what is a possible route for the indirect effect of EA on seemingly unrelated traits. We believe that the authors can at least work in the literature that shows that low SES is a significant risk factor in numerous diseases.

We agree that it would be valuable to comment more on possible routes for the indirect effects of EA on seemingly unrelated phenotypes. We also agree that the indirect effects may have connections with the literature that shows that low SES is a significant risk factor in numerous diseases. Indeed, for that reason, we refer to that literature in the first sentences of the paper:

Educational attainment (EA) is an important dimension of socioeconomic status that features prominently in research by social scientists, epidemiologists, and other medical researchers. EA is strongly related to a range of health behaviors and outcomes, including mortality (Marioni *et al.*, 2016).

In the revised manuscript, we now also give examples of routes for indirect effects from parents when first explaining our within-family analyses:

The population effect captures the sum of the direct effect, indirect effects from relatives (e.g., genetic influences on parents' education, socioeconomic status, and behavior), other gene-environment correlation (i.e., correlation between genotypes and environmental exposure, with population stratification being one possible cause), and a contribution from the genetic component of the phenotype that would be uncorrelated with the PGI under random mating but becomes correlated with the PGI due to the linkage disequilibrium between causal alleles induced by assortative mating (**Supplementary Note**) (Kong *et al.*, 2018; Howe *et al.*, 2021).

Before we turn to the novel analyses we ran, it is useful to clarify the key difficulty we encountered when we brainstormed about ways to make further progress on the indirect effects. Fundamentally, it is challenging to robustly identify indirect genetic effects using the data available to us. Our within-family analysis allows us to isolate the component of the predictive power of the PGI that is due to direct effects, but the remainder of the predictive power is due to some combination of indirect effects, other gene-environment correlation, and assortative mating. We believe our study provides clear evidence that these factors are jointly important, but cleanly separating out the contributions of these different factors is challenging without multi-generational data, which would allow for unbiased estimation of parental indirect genetic effects. We worked hard trying to make further progress, while at the same time being vigilant to avoid overinterpreting our results.

The referee's comment spurred us to conduct an additional analysis that makes some effort in the desired direction. Specifically, we started with our regression of 13 phenotypes on an individual's PGI and the individual's parents' PGIs in Generation Scotland. We then added three controls to these regressions: parental EA, parental cognitive performance, and parental vocabulary test

score. We examined to what extent adding these controls reduced the coefficient on the parents' PGIs. For this analysis, we used the sample of 2,964 individuals in Generation Scotland with both parents genotyped (we could not use UKB because it does not have the relevant phenotypic data on parents). The results are shown in the below figure. The green and orange bars, respectively, show the average coefficient of the mother's and father's PGIs without and with the parental phenotype controls. The white bars are the differences between the green and orange bars. For each estimate, the 95% confidence interval is shown.

Reductions in the coefficient on the parents' PGIs are visible for three phenotypes: an individual's own EA, own cognitive performance, and own vocabulary test score. These are the only three reductions that reach statistical significance at the stringent statistical significance threshold of $P < 0.005/13 = 0.0004$ (this includes a Bonferroni correction for 13 phenotypes, starting from a more-stringent-than-conventional threshold of 0.005 rather than 0.05), based on a two-sided Z-test.

These results are consistent with the possibility that indirect parental effects operate, at least in part, through factors relating to parental education and cognition. However, we ultimately decided against including these new results in the paper due to space constraints and interpretational challenges. First, as already noted, the coefficient on the parental PGI cannot be interpreted as the indirect effect. Second, the results could be reconciled with any of the other

possible interpretations of the coefficient on the parental PGI. For example, the results are consistent with population stratification driving the correlation between the parental PGI and the individual's phenotypes.

Of course, if the referee and editor feel that we should include these results in paper, we will readily do so. If we do add them, we will of course convey the interpretational challenges.

In the revised manuscript, we do cite related evidence by Kong *et al.* (2018), Selzam *et al.* (2019), and Willoughby *et al.* (2019). These papers conducted analyses similar to the above—testing whether the coefficient on the parental PGI is reduced when additional controls are added to a regression of an individual's phenotype on the individual's PGI and the parental PGI—but their control variables differed from ours (e.g., Kong *et al.* and Selzam *et al.* controlled for parental SES, rather than parental EA, parental cognitive performance, and parental vocabulary test score). Like in our analysis, these papers found that the coefficient on the parental PGI is indeed reduced, but we similarly feel that these results are difficult to interpret. Thus, we cite these papers in order to highlight that our empirical strategy follows theirs:

Our next set of analyses, like related prior work (Kong *et al.*, 2018; Selzam *et al.*, 2019; Willoughby *et al.*, 2019), aim to isolate the component of the PGI's predictive power that is due to direct effects (Kong *et al.*, 2018; Walsh and Lynch, 2018): causal effects of an individual's genetic material on that individual.

Second, LDscore intercept is notably larger than what we are used to seeing. I think this deserves more comments on the possible causes and comparisons with other GWAS (including some of the previous EA GWAS) in terms of lambda 1000.

We were also struck by the unusually large intercept at first. But it turns out that it is roughly of the magnitude that one should expect based on the unusually large sample size of our GWAS. All else equal, the LDSC intercept should increase as the GWAS sample size increases (Bulik-Sullivan

et al., 2015). A GWAS with a sample size of 3 million individuals—much larger than people are used to seeing—is therefore expected to produce an intercept larger than people are used to seeing.

More precisely, the expected intercept from an LD score regression is equal to $Na + 1$, where N is the sample size of the GWAS and a is the contribution of confounding biases (such as population stratification) to the χ^2 statistics from the GWAS. The intercept by itself is not a useful diagnostic of the amount of bias in the underlying GWAS (Loh *et al.*, 2018). The share of inflation in test statistics due to bias can be measured by the ratio of the intercept minus one (which is an estimate of Na) to the mean χ^2 statistic minus one. We should expect this ratio to be approximately invariant to the GWAS sample size. And in practice, it has remained roughly stable across the various iterations of our EA GWASs.

For example, the ratio was 5% in our previous large-scale GWAS of EA (Lee *et al.*, 2018), compared to 7% in the current study. It is also in the same ballpark as the ratio of 6% reported in the largest GWAS of height published to date (Yengo *et al.*, 2018). In the revised manuscript, we now remark explicitly about the similarity of this ratio between the current GWAS results and those of Lee *et al.* (2018): “According to the LD score regression (Bulik-Sullivan *et al.*, 2015) intercept (1.66), confounding accounts for 7% of the inflation, similar to previous GWAS of EA (Lee *et al.*, 2018)...”

(We interpret the referee’s reference to λ_{1000} as a request to compare the inflation in the χ^2 statistics in a way that adjusts for sample size, as we did above. To the best of our knowledge, the λ_{1000} was developed to facilitate comparability of results from case-control studies with different numbers of cases and/or controls. Moreover, unlike the LD score intercept, it does not distinguish between confounding biases and inflation that is due to true polygenic signal.)

Third, while they must have dealt with this elsewhere, is it true that all their data lists total years of education or do they get clumped data (8-11 years, finished high school, some college, graduated college, etc.). If the latter, how do they deal with different “clumping” of YOE where

the meaning of the thresholds likely change between samples and across birth cohorts? One of us has fitted multiple threshold models in pairs of relatives to such clumped YOE data in a large longitudinal sample covering 4 decades where YOE was rising rapidly. The model fitted terribly indicating that the “meaning” of the different categories was not constant across birth cohorts. How do they treat “technical schools” etc? We searched and were unable to find a discussion of these concerns in their material.

The data included in the GWAS meta-analysis come from over 70 datasets from at least 16 countries (16 is the number assuming the 23andMe data come only from countries represented also by other cohorts). Often, even cohorts from the same country used different survey questions to measure EA. Across datasets, the number of response options for the survey question ranged from 4 to 20, with a mean of 7.7 options (see Lee *et al.* (2018), Supplementary Table 22 and Supplementary Table 24 Panel A). Datasets with a large number of options require little clumping (as in the Estonian Biobank), but usually the data are clumped into a smaller number of options (as in 23andMe and the UK Biobank, which have 6 and 7 response options, respectively).

Dealing with differences in measures, as well as differences in the meaning of EA across countries and birth cohorts, has been a major challenge for GWAS meta-analyses of EA from the very beginning. In the first large-scale GWAS of EA (Rietveld *et al.*, 2013), our approach was to map country-specific educational categories onto the United Nation’s 1997 International Standard Classification of Education (ISCED), and the approach we ultimately settled on was developed in consultation with data providers and researchers familiar with the education systems and qualifications in countries with at least one GWAS cohort. The work was overseen by the Social Science Genetic Association Consortium principal investigators and an education researcher specializing on cross-country comparability of educational qualifications (Professor Roelande Hofman).

Because we were concerned about potential non-linearity in the relationship between genotypes and EA, we studied a binary phenotype, college completion, in addition to a continuous

phenotype, years of education. In subsequent GWAS of EA, we have continued to use the 1997 ISCED scale to harmonize the years-of-education phenotype. As we explain in Supplementary Note section 1.1 of Okbay *et al.* (2016), we decided to de-emphasize the college completion phenotype before that paper was completed:

... the original plan treated EduYears [the years-of-education phenotype] and College [the college completion phenotype] symmetrically whereas throughout the manuscript, we treat EduYears as the primary variable and de-emphasize College. After circulation of the Analysis Plan to our cohorts, a paper was posted on bioRxiv showing that the genetic correlation between the two measures is very high, with the point estimate suggesting a perfect genetic correlation (Bulik-Sullivan *et al.*, 2015). Previously, we had considered as plausible the possibility that College would have better power for detecting associations at the upper end of the distribution of EduYears. However, since College is constructed by dichotomizing EduYears, the very high genetic correlation suggests that the College phenotype is for all intents and purposes merely a coarsening of the EduYears phenotype.

This interpretation was borne out in the Okbay *et al.* (2016) results; as noted in their Supplementary Note section 1.6.2: “Overall, the results [for College] are similar to those from the EduYears analyses, but with higher P-values (consistent with the hypothesis that the College variable is a noisier measure of educational attainment than the EduYears variable).” We therefore dropped the college completion phenotype in Lee *et al.* (2018).

Our approaches to the issues raised by the referee’s comment have been to try to do the best we can given data and logistical constraints. We do not have access to individual-level data from the vast majority of the 70+ cohorts included in the meta-analysis, and cohort analysts could typically run only two specifications for GWAS (partly because of constraints build into the software that was used, such as Plink): ordinary least squares (OLS) and logistic regression. Later meta-analyses built on the results from earlier meta-analyses, and we could not re-run the GWAS that had already

been conducted. The largest contributor to our current meta-analysis, 23andMe, is extremely reluctant to run non-standard analyses (including anything that deviates from OLS or logistic regression) because their software is optimized for these analyses and their analyst time is very limited.

We agree with the referee that these issues are important, and we have tried to study them and improve our measurement of EA where possible. For example, in Lee *et al.* (2018, Supplementary Note section 3), we analyzed the genetic correlations (and SNP heritabilities) across different measures, countries, and cohorts. In the current paper, as described in Supplementary Note section 1, we improved our coding of years of education in the UK Biobank in this GWAS relative to our coding in previous GWAS.

In the current paper, instead of reporting how EA was coded in each of the cohorts in the GWAS meta-analysis, in (the slightly revised) Supplementary Note section 2.2.2. (“Phenotypes”), we refer the reader to the appropriate table to find information about how EA was coded within the paper where the GWAS for that cohort was initially reported:

As in our prior work, we analyze the *EduYears* phenotype obtained by mapping the highest level of education that a respondent achieved to an International Standard Classification of Education (ISCED) category and then imputing a years-of-education equivalent for each ISCED category (see **Supplementary Note** section 1.1.1 for the ISCED to years-of-education mapping). The phenotype measurement and distribution for the 23andMe cohort and the updated UKB GWAS (see Section 1) are summarized in Panel B of **Supplementary Table 15**. For analogous information on the remaining cohorts, see Supplementary Tables 17 and 1.3 in Lee *et al.* and Okbay *et al.*, respectively.

The exception is for the UK Biobank cohort, where we devote section 1 of the Supplementary Note to explaining the phenotype coding, since we changed it relative to earlier GWAS.

Fourth, given that parental PGIs are the (equal weighted) sum of transmitted and untransmitted alleles, can they be more specific about how their “controlling for parental PGI” is or is not equivalent to transmitted vs untransmitted analyses of Kong et al?

Kong *et al.* (2018) regress an individual’s phenotype on the individual’s PGI and a PGI constructed from the non-transmitted parental alleles, and they subtract the coefficient on the non-transmitted parental PGI from the coefficient on the transmitted PGI to estimate the direct genetic effect. In the current paper, we instead regress an individual’s phenotype on the individual’s PGI, controlling for parental PGI. These two approaches are equivalent, as we now explain.

The mother and the father have four alleles at each site in the genome, two of which are transmitted to the offspring and two of which are not transmitted. Let $PGI_{par(i)}$ denote the sum of maternal and paternal PGIs. This is therefore the sum of the individual’s PGI (calculated from the sum of transmitted alleles), denoted PGI_i , and the non-transmitted parental PGI, denoted PGI_i^{NT} : $PGI_{par(i)} = PGI_i + PGI_i^{NT}$. Consider the regression model for the individual’s phenotype, Y_i , corresponding to the regression approach in the current paper:

$$Y_i = \delta PGI_i + \alpha PGI_{par(i)} + \epsilon_i$$

Because $PGI_{par(i)} = PGI_i + PGI_i^{NT}$, we can rewrite this regression equation to correspond to the approach in Kong et al.:

$$Y_i = (\delta + \alpha) PGI_i + \alpha PGI_i^{NT} + \epsilon_i.$$

Note also that the estimate of the direct effect in Kong *et al.* is $(\delta + \alpha) - \alpha$, which equals δ , the estimate of the direct effect in the current paper. We have added a new Supplementary Note section 7.2 that makes this relationship to the method of Kong *et al.* explicit.

In our analysis of trios, we use the regression with parental PGIs rather than the (statistically equivalent) Kong *et al.* approach because the latter requires an additional step of determining parent-of-origin of alleles, which is both computationally costly and could introduce error.

Unlike in Kong *et al.*, in addition to our analysis of trios, we also use genetic differences between siblings to estimate the direct genetic effect (see Supplementary Note section 9 for a derivation of the method in terms of an explicit model). As in the trios analysis, the sibling analysis exploits the random segregation of genetic material during meiosis to identify the direct genetic effect. Conceptually, the main difference is that the approach that uses genetic differences between siblings can be biased when indirect genetic effects from siblings are present. However, a recent analysis in the UK Biobank found that indirect genetic effects from siblings are likely to be negligible for the EA PGI (Kong, Benonisdottir and Young, 2020). Therefore, this is unlikely to explain why our estimate of the fraction of the PGI's predictive power due to direct effects is smaller than Kong *et al.*'s. Instead, our finding is likely due to a stronger influence of indirect genetic effects, population stratification, and assortative mating in the UK and Sweden (where our samples are from) than in Iceland (where Kong *et al.*'s sample is from). It is notable that Iceland has a very low level of income inequality, with one of the lowest Gini coefficients in the OECD (<https://data.oecd.org/inequality/income-inequality.htm>), and much lower than in the UK.

To concisely mention these issues, we have added a parenthetical comment to the first paragraph of the Results section of “Within-Family analyses”: “(The regression controlling for parental PGIs gives an equivalent estimate of the direct effect of the PGI as a regression on PGIs constructed from transmitted and non-transmitted parental alleles (Kong *et al.*, 2018); see **Supplementary Note.**)” In addition, we have added a new section 7.2 to the Supplementary Note that explains the equivalence.

Firth, should they be discussing “assortative mating” or “spousal resemblance?” Do they have data on spousal correlations at marriage and if not, how can they justify their title? In their analyses of the causes of the high correlation, they don't seem to consider “spousal interaction”, that is that

one individual's EA could directly impact on the EA of their spouse by providing financial support, encouragement, etc. Doesn't that possibility merit discussion?

These are fair points that came up during numerous discussions as we were working on the paper. Before addressing the referee's comment, we note that we changed our language about "spouses" to "mates" throughout the paper because our data come from identifying genotyped pairs of individuals who jointly have a genotyped biological child in the same dataset. However, in the remainder of the response to the referee's comment, we mostly revert to the language of "spouses" because the referee's comment pertains to spouse pairs, and most of the mate pairs in our data are likely to be spouses.

We agree with the referee that for characteristics that vary over time, the distinction between spousal resemblance and assortative matching may be important. In our setting, we think the concern applies mostly to BMI, a phenotype for which it seems plausible that spousal interaction effects through shared meals and other shared lifestyles could cause convergence after the marriage. For height, cognitive performance, and educational attainment (usually completed prior to marriage) we find it less plausible that spousal interactions would substantially impact resemblance, even though we find it plausible that interaction effects are not exactly zero (at least not for cognitive performance and educational attainment). Furthermore, interaction effects, if they exist, almost certainly increase spousal resemblance, and therefore increase the predicted correlation between mate pairs' EA PGIs, so spousal interactions cannot explain why the mate-pair EA PGI correlation is higher than predicted under the model of phenotypic assortment.

In analyses inspired by the referee's query, we used a data set covering the entire Swedish population to estimate spousal EA correlations by year since marriage. Specifically, the data are merged from two administrative sources: the Longitudinal Integrated Database for Labour Market Research (LISA) for education and family identifiers, and the Total Population Register (RTB) for information on marital status and dates of marriages/divorces (Ekbom, 2011). For all heterosexual couples who got married in Sweden between 1990 and 2000, we have annual information about

the highest educational qualification of both spouses for at least fifteen years after the marriage. Although it barely matters for the estimates, we restrict the sample to only those who have stayed married for at least 15 years in order to keep the sample roughly the same across the different time points. Here are our estimates of the spousal phenotypic correlation for each t years after marriage ($t = 0$ is the year of marriage):

Years after marriage	Spousal EA correlation	S.E.	Number of spouse pairs
0	0.476	0.002	192353
1	0.476	0.002	199721
2	0.482	0.002	202150
3	0.484	0.002	203251
4	0.485	0.002	203331
5	0.487	0.002	202548
6	0.487	0.002	201124
7	0.487	0.002	198887
8	0.487	0.002	196169
9	0.484	0.002	192508
10	0.481	0.002	186973
11	0.480	0.002	180806
12	0.480	0.002	172962
13	0.481	0.002	163647
14	0.483	0.002	150168
15	0.487	0.002	129759

In this data, the spousal phenotypic correlation increases only very slightly over the 15 years subsequent to marriage. Unfortunately, analogous analyses for BMI or cognitive performance are not possible in this data set since those variables are not measured annually in the Swedish administrative records.

In the revised manuscript, we have added text discussing this to the Online Methods section on ‘Assortative mating analyses’: “We note that we use the correlation in phenotypes measured in mate pairs after they paired up, which could be inflated if mate pairs influence each other, leading to greater phenotypic similarity than at the time of pairing. This would have the effect of predicting a higher mate pair PGI correlation than if we had used phenotypes from the time of pairing.”

Minor issues

On line 162 “relatively weak enrichment” should probably read better as “weaker enrichment”. Re: **genes highly expressed in glial cells, what (MSigDB) pathway/gene sets they tend to significantly load on?**

We thank the referee for the improvement in wording, which we have implemented.

We investigated the overlap between the genes annotated as **astrocyte**, **oligodendrocyte**, and **neuron** respectively in the Cahoy *et al.* (2008) dataset and the gene sets employed in the Panther Overrepresentation Test (<http://geneontology.org/>). (We used Panther rather than the MSigDB enrichment tool (<https://www.gsea-msigdb.org/gsea/msigdb/annotate.jsp>) because the latter imposes a maximum number of genes in the input list that was exceeded by the **astrocyte** gene set.) We selected the GO (biological process, molecular function, cell compartment) and Reactome gene sets for overlap testing because these gene sets are also present in MSigDB and have proven useful in our previous work (Okbay *et al.*, 2016; Lee *et al.*, 2018). We selected Fisher’s exact test to detect the enrichment and $FDR < 0.05$ as our significance criterion. We downloaded the results for each combination of Cahoy *et al.* (2008) input list and Panther gene set; we can readily provide these results if requested. Here we try to provide a fair summary.

Many queries produced a long list of significant results with no clear overall interpretation. This was particularly true when the target gene sets were GO molecular function and Reactome. Text

searches of most tabulated results, however, did produce straightforward support for the validity of the input list.

The **astrocyte** input list enriches a number of GO biological processes defined by astrocytes: **negative regulation of astrocyte differentiation** (5.43-fold enrichment, $P < 0.002$), **regulation of astrocyte differentiation** (4.32-fold enrichment, $P < 2 \cdot 10^{-4}$), **astrocyte development** (3.08-fold enrichment, $P < 0.003$), and **astrocyte differentiation** (3.04-fold enrichment, $P < 4 \cdot 10^{-4}$). It also enriches a number of processes defined by glial cells: **negative regulation of glial cell differentiation** (3.88-fold enrichment, $P < 0.001$), **regulation of glial cell proliferation** (3.36-fold enrichment, $P < 0.002$), **glial cell differentiation** (2.68-fold enrichment, $P < 7 \cdot 10^{-8}$), and **glial cell development** (2.64-fold enrichment, $P < 2 \cdot 10^{-5}$). The **astrocyte** input list also enriches the GO cellular compartments **astrocyte projection** (5.55-fold enrichment, $P < 5 \cdot 10^{-5}$) and **glial cell projection** (3.56-fold enrichment, $P < 6 \cdot 10^{-4}$). The former represents the second largest enrichment in this query.

The **oligodendrocyte** input list enriches a number of GO biological processes defined by myelination, including **myelin assembly** (5.77-fold enrichment, $P < 7 \cdot 10^{-5}$), **axon ensheathment in central nervous system** (5.45-fold enrichment, $P < 3 \cdot 10^{-4}$), **axon ensheathment** (4.71-fold enrichment, $P < 7 \cdot 10^{-14}$), and **myelination** (4.57-fold enrichment, $P < 6 \cdot 10^{-13}$). These results are among the most strongly enriched in this query. The input list also enriches processes defined by oligodendrocytes: **oligodendrocyte development** (4.51-fold enrichment, $P < 6 \cdot 10^{-6}$) and **oligodendrocyte differentiation** (4.04-fold enrichment, $P < 3 \cdot 10^{-7}$). Among the top GO cellular compartments enriched by the **oligodendrocyte** input list are **compact myelin** (6.05-fold enrichment, $P < 7 \cdot 10^{-4}$) and **myelin sheath** (4.6-fold enrichment, $P < 7 \cdot 10^{-7}$).

The **neuron** input list enriches gene sets, including those in the GO molecular function and Reactome categories, that are clearly and predominantly defined by neuronal function.

We have not included these results in the paper because they represent validation of a data source already in wide use (Cahoy *et al.* (2008) has 2688 citations in Google Scholar, while Finucane *et al.* (2018) has 385). We view them as outside the scope of the paper. However, if the referee and editor feel that we should include these results in the paper, we will readily do so.

They should explain in their legend all abbreviations used in their tables.

We have carefully gone through all legends and double-checked that all abbreviations are now defined.

What exactly is “social homogamy based on genetic relatedness”? That deserves some explanation in the text.

We thank the reviewer for highlighting the need for better explanation here. In the revised manuscript, we now write: “Not all forms of social homogamy generate a mate-pair PGI correlation (Reynolds, Baker and Pedersen, 2000), but social homogamy that is related to genetic ancestry—for example, due to geographic proximity that tracks genetic structure in the population—will do so if there are components of genetic ancestry correlated with the PGI.”

They write “lead SNPs corresponds to 1.4 weeks of schooling per allele.” While cute, this is hardly a sustainable or sensible interpretation.

The purpose of the original sentence was to convey a rough sense of the distribution of effect sizes of our lead SNPs. The referee’s comment prompted us to make some edits to the text to ensure that our point comes across more clearly. Before we turn to the edits, it may be useful to start with a brief summary of what we intended to say.

We know from past experiences that some readers find it helpful when we supplement information about the effect sizes expressed as an R^2 or a standardized regression coefficient with

unstandardized regression coefficients. Readers who wish to gauge the effect of an additional copy of the reference allele of some lead SNP on years of schooling could in principle obtain the information from posted summary statistics. However, there are two complications:

1. Since the lead SNPs were selected using a process that filters on a p -value threshold, the estimated coefficients will generally be overestimates (due to the so-called winner's curse).
2. The estimated effect sizes provide the effect of an additional reference allele on *EduYears* measured in standard-deviation units. The standardized effect sizes are easier to work with in meta-analysis, but we have found that many readers find unstandardized effect sizes more intuitive.

Supplementary Note section 2.2.6 describes how we generated the winner's-curse adjusted estimates of the unstandardized effects of our lead SNPs.

The referee's comments prompted us to edit the sentence so that it now reads:

Adjusted for the winner's curse, we find that the effects of our lead SNPs are consistently quite small. On average, an additional copy of the reference allele of the median SNP is associated with 1.4 weeks of schooling more schooling: the effects at the 5th and 95th percentiles (in absolute value) are 0.9 and 3.5 weeks, respectively (see **Supplementary Note** for details on these calculations).

We hope this alternative phrasing, which avoids causal language when describing the coefficient, is an improvement.

They write "Our findings are fully consistent with earlier conclusions: SNP heritability due to the X chromosome of 0.4% and (using sex stratified association analyses in the UK Biobank) a male-

female genetic correlation close to unity ($r = 0.94$).” Is this correlation just for the X-chromosome or the entire genome?

In the revised manuscript, we have clarified that this genetic correlation refers only to the X chromosome: “a male-female genetic correlation on the X chromosome close to unity...”

(We did not run sex-stratified analyses for the autosomal meta-analysis because there is compelling evidence from our prior work that the male-female genetic correlation for *EduYears* is close to one. For example, Okbay *et al.*'s (2016) data yields an estimate of 0.98 (S.E. = 0.029).)

In figure 3 they use the term “depression.” It appears they mean major depression. It should be listed as such as generic “depression” can mean a variety of different things – e.g. self-report current symptoms above some cut-point.

Thanks for pointing this out. We have made this change.

There might not be room, but can they say something about the expected gain in predictive power as a function of sample size. Should it be linear as a function of log to base 10 sample size?

The originally submitted manuscript did not have a Methods section. In the revised manuscript, we now state in the Methods section:

We calculate the expected prediction accuracy of the EA PGI using a generalization of de Vlaming *et al.* (2017). In the Supplementary Note section 5.5, we show that the expected coefficient of determination, R^2 , can be expressed as the following function of the discovery sample size, N :

$$E(R^2) = \frac{A}{B+1/N}$$

where the parameters A and B are functions of the SNP heritabilities in the discovery and prediction samples, the genetic correlation between the samples, and the effective number of SNPs included in the PGI.

More details are in the Supplementary Note section 5.5.

Response to Reviewer #3:

Remarks to the Author:

The manuscript by Okbay et al. represents the largest GWAS effort of educational attainment (EA), with a 3-million predominantly European ancestry subjects. They reported 3,952 approximately uncorrelated genome-wide-significant SNPs. The polygenic score or polygenic index (PGI) explains 12-16% of EA variation and also other diseases. These results represent the increment advance of the previous study conducted by the same group (ref 2) and should be insightful for other large GWAS. The manuscript is well written.

We thank the reviewer for the positive remarks.

However, it is a little disappointed about how much biology we can learn from this large GWAS effort. Some statistical analyses may need careful investigation.

The referee is right to remark that for this iteration of our GWAS, there was a shift of emphasis toward family-based and assortative mating analyses. This reorientation was the result of some lengthy internal deliberations. We ultimately agreed that, in light of the threefold increase in sample size relative to the previous GWAS (Lee *et al.*, 2018), it made sense for us to prioritize analyses: (i) identifying novel associations and constructing, evaluating and disseminating more predictive PGIs, (ii) conducting novel and informative analyses of mate-pair assortment processes (e.g., testing if mate-pair PGI resemblance is consistent with assortment on phenotype) and (iii) conducting better-powered analyses of family-based data (e.g., assessing how much of the overall predictive power of our PGI for various phenotypes is due to direct effects). In our previous GWAS (Lee *et al.*, 2018), we conducted a comprehensive battery of biological annotation analyses. For the current paper, we reran one of those analyses—stratified LD score regression—and found that the results were generally very similar (albeit more precise). This similarity indicates to us that if we had run more bioinformatics pipelines, a likely outcome is that most of the analyses would

have produced only marginal new insights. Ultimately, we feel that our decision to focus on other analyses was justified, but we appreciate that decisions about how to resolve these sorts of tradeoffs are inherently subjective. No matter what is ultimately decided, there will always be some readers who would have preferred a somewhat different focus.

Comments:

1) The authors isolated the direct and indirect effects of the PGI in the population effect by controlling for both parents' PGIs in a regression model. They estimated 30.9% of the EA direct effect, which is significantly smaller than the estimate of 48.9% reported in the Icelandic data (Ref 4). In Discussion, they claimed that "much of the predictive power of the EA PGI is not explained by direct effect" (Page 10). However, it seems the regression approach used in this study is different from the previous study. Since parents transmit 50% of their genomes to offspring, adjusting for parents' PGIs likely underestimates the direct effects in this study. Thus, this claim seems unconvinced. It would be more reasonable to apply similar analysis approach of Ref 4.

The reviewer is correct that we use a different method than the method applied in Kong *et al.* to infer the direct effect of a PGI. Kong *et al.* regress an individual's phenotype on the individual's PGI and a PGI constructed from the non-transmitted parental alleles, and they subtract the coefficient on the non-transmitted parental PGI from the coefficient on the transmitted PGI to estimate the direct genetic effect. In the current paper, we instead regress an individual's phenotype on the individual's PGI, controlling for parental PGI. These two approaches are equivalent, as we now explain.

The mother and the father have four alleles at each site in the genome, two of which are transmitted to the offspring and two of which are not transmitted. Let $PGI_{par(i)}$ denote the sum of maternal and paternal PGIs. This is therefore the sum of the individual's PGI (calculated from the sum of transmitted alleles), denoted PGI_i , and the non-transmitted parental PGI, denoted PGI_i^{NT} : $PGI_{par(i)} = PGI_i + PGI_i^{NT}$. Consider the regression model for the individual's phenotype, Y_i , corresponding to the regression approach in the current paper:

$$Y_i = \delta PGI_i + \alpha PGI_{par(i)} + \epsilon_i$$

Because $PGI_{par(i)} = PGI_i + PGI_i^{NT}$, we can rewrite this regression equation to correspond to the approach in Kong *et al.*:

$$Y_i = (\delta + \alpha)PGI_i + \alpha PGI_i^{NT} + \epsilon_i.$$

Note also that the estimate of the direct effect in Kong *et al.* is $(\delta + \alpha) - \alpha$, which equals δ , the estimate of the direct effect in the current paper. We have added material to Supplementary Note Section 7 that makes the comparison to the method of Kong *et al.* explicit.

In our analysis of trios, we use the approach we do rather than the (statistically equivalent) Kong *et al.* approach because the latter requires an additional step of determining parent-of-origin of the offspring's alleles, which is computationally costly and could introduce error.

Unlike in Kong *et al.*, in addition to our analysis of trios, we also use genetic differences between siblings to estimate the direct genetic effect (see Supplementary Note section 9 for a derivation of the method in terms of an explicit model). As in the trios analysis, the sibling analysis exploits the random segregation of genetic material during meiosis to identify the direct genetic effect. Conceptually, the main difference is that the approach that uses genetic differences between siblings can be biased when indirect genetic effects from siblings are present. However, a recent analysis in the UK Biobank found that indirect genetic effects from siblings are likely to be negligible for the EA PGI (Young *et al.*, 2020). Therefore, this is unlikely to explain why our estimate of the fraction of the PGI's predictive power due to direct effects is smaller than Kong *et al.*'s. Instead, our finding is likely due to a stronger influence of indirect genetic effects, population stratification, and assortative mating in the UK and Sweden (where our samples are from) than in Iceland (where Kong *et al.*'s sample is from). It is notable that Iceland has a very low level of income inequality, with one of the lowest Gini coefficients in the OECD (<https://data.oecd.org/inequality/income-inequality.htm>), and much lower than in the UK.

To concisely mention these issues, we have added a parenthetical comment to the first paragraph of the Results section of “Within-Family analyses”: “(The regression controlling for parental PGIs gives an equivalent estimate of the direct effect of the PGI as a regression on PGIs constructed from transmitted and non-transmitted parental alleles (Kong *et al.*, 2018); see **Supplementary Note.**)

2) The authors performed assortative mating analysis and tried to answer how much PGI prediction power is due to assortative mating. They found that the observed spousal PGI correlation is substantially higher than the predicted PGI correlation, which is the product of the spousal phenotype correlation and both father and mother’s phenotype and PGI correlations. In Supplementary Table 12, father and mother’s phenotype and PGI correlations are missed. There are no standard errors of predicted PGI correlations listed. By controlling the ancestry, the spousal PGI correlation is much reduced. Furthermore, it is not clear why the spousal phenotype correlation reported in literature is much higher than that in the current study (0.412). Therefore, the predicted PGI correlation may be underestimated.

We thank the referee for highlighting these omissions in the previous Supplementary Table 12, which is Supplementary Table 14 in the revised manuscript. We have now added correlations between PGI and phenotype for both fathers and mothers. We did not originally include standard errors of predicted PGI correlations because it was not clear how to calculate them. In response to the referee’s comment, we calculated these standard errors by bootstrapping over mate pairs. We now report these standard errors in the table and in the main text, and we display the confidence intervals of the predicted correlation in Figure 5. We note that the standard errors for the predicted correlations are generally small: for example, the predicted correlation for EA is 0.031 with a standard error of 0.004. This is clearly statistically distinguishable from the observed correlation of 0.175 with a standard error of 0.020.

The referee is correct that the mate-pair EA correlation reported in our study is on the low end of typical estimates in the literature, which are sometimes as large as 0.6. However, much of the literature focuses on U.S. samples. The samples we use for our assortative mating analyses are from the U.K. For comparison with our estimate in our revised analysis (0.430, S.E. = 0.017), we were able to obtain two estimates of assortative mating on EA from nationally representative UK samples. In the literature, we found an estimate of 0.45 from the UK Household Longitudinal Study: Understanding Society (Hugh-Jones *et al.*, 2016), for which we estimate a standard error of 0.026 (using the approximation formula $\sqrt{(1 + r^2/2)/(N - 3)}$ from Bonnett and Wright (2000, p.23)). Because we have access to relevant data from the English Longitudinal Study of Ageing, we also calculated the mate pair correlation there: 0.513 (S.E. = 0.018). Relative to these estimates, we view ours as only slightly lower.

In the Online Methods of the revised manuscript, we have added a sentence to indicate that the correlation in our sample is close to that from a nationally representative UK sample: “Although the UK Biobank is not a representative sample, the correlation between mate pairs’ educational attainments in our sample (0.430, S.E. = 0.017) is not very different from those in representative UK samples: very close to the correlation of 0.45 estimated by Hugh-Jones *et al.* (2016) and only somewhat smaller than we estimate (0.513, S.E. = 0.018) in the English Longitudinal Study of Ageing (Stephens *et al.*, 2013) (we used the 3470 mate pairs identified in the harmonized ELSA data from the Gateway to Global Aging (g2aging.org) and our updated UK Biobank coding of *EduYears*).”

In response to the referee’s specific concern about our conclusion that “the observed spousal PGI correlation is substantially higher than the predicted PGI correlation,” we note that even if the mate-pair EA correlation in our sample were different from a nationally representative sample, this would not explain why the mate-pair EA PGI correlation is higher than predicted under a model of phenotypic assortment. If the model of phenotypic assortment is correct in our sample, then the mate-pair EA PGI correlation should follow the prediction based on the mate-pair phenotypic correlation in our sample. However, the mate-pair EA PGI correlation is much higher than the

prediction, taking into account both the uncertainty in the prediction and the observed mate-pair PGI correlation.

3) In the assortative mating analysis, the residuals of the father's and mother's PGIs were calculated after regression on their top 20 genetic principal components. Are the top 20 genetic principal components sufficient to control the effect by population stratification?

Controlling for the top 20 genetic principal components (PCs) is unlikely to fully control for population stratification. In the revised manuscript, we expanded the set of controls to the top 40 genetic PCs. The results are largely unchanged. However, we do not believe that simply controlling for more PCs can eliminate all effects of population stratification because, even when the PCs are estimated in a sample as large as the UKB, the higher order PCs are essentially just noise (see <https://www.youtube.com/watch?v=B7ub92OLw1g>). Moreover, aside from the issue of estimation error in the PCs, it is unlikely that we could ever fully control for geographic variation in the PGI using PCs constructed from common variants (Zaidi and Mathieson, 2020).

In the revised manuscript, we therefore attempted to go beyond geographic factors captured by PCs by taking advantage of the availability of north and east birth coordinates and the assessment center records in the UKB. (Unfortunately, these variables are not available in the other dataset we use for these analyses, Generation Scotland.) In the Online Methods, we explain how we used these variables:

In UKB, north and east birth coordinates in the UK (Data Fields 129-130) were recorded, in addition to the center where individuals were assessed (Data Field 54). To further assess the impact of geographic factors on the correlation between mate pairs' EA PGIs, we added north and east birth coordinates and the product of north and east birth coordinate, along with assessment center coded as a categorical variable, as regressors to the regression of the EA PGI onto EA and principal components in the UKB.

In the main text section ‘Assortative mating analyses’, we now discuss the results of this analysis (which we have added to what is now Supplementary Table 14):

After residualizing the EA PGI on 40 principal components (PCs) of the genomic relatedness matrix in addition to EA, we find that the mate-pair PGI correlation falls to 0.091 (S.E. = 0.021). This implies that some, but not most, of the mate-pair PGI correlation is due to assortment on genetic ancestry captured by the PCs (or some factor correlated with the PCs). In the UKB, further adjustment for birth coordinates and the center where participants were assessed (**Online Methods**) resulted in a slight reduction of the correlation between mate pairs’ PGIs (**Supplementary Table 14**), suggesting that geographic factors not captured by the top 40 PCs also contribute to the high mate-pair EA PGI correlation.

To summarize, *any* adjustment for geographic factors and population structure is likely to be imperfect and incomplete. Our analyses indicate that both population structure captured by the top 40 PCs and other geographic factors contribute to the high mate-pair EA PGI correlation. However, a substantial correlation between the mate pairs’ EA PGIs remains after accounting for PCs and educational and cognitive phenotypes, the origin of which we can only speculate about. We propose “assortment on phenotypes correlated with the EA PGI other than EA, cognitive performance, and vocabulary—possibly including various personality traits (Belsky *et al.*, 2016; Möttus *et al.*, 2017; Smith-Woolley, Selzam and Plomin, 2019)—and sources of social homogamy other than genetic ancestry captured by the top 40 PCs—possibly including geographic location at courtship age (Abdellaoui *et al.*, 2019; Laidley, Vinneau and Boardman, 2019), socioeconomic status, and social class (Belsky *et al.*, 2018)” as plausible possible explanations that we do not have the data to assess. Nevertheless, our results raise important questions about the processes of assortative mating, rejecting the commonly assumed model of phenotypic assortment and assortment on cognitive phenotypes as sufficient explanations. We anticipate that our results will

help spur future work that seeks to find the other factors that explain the correlation between mate pairs' EA PGIs.

4) The PGI was calculated by GWAS summary statistics of both males and females. However, it is known that gender is associated with EA, also year of birth. Have gender and year of birth and their interaction been including in the analysis?

The cohort-specific GWASs controlled for sex, year of birth, and their interaction. In the revised manuscript, we have now clarified this. In the explanation of the additive GWAS, we now write: "All analyses were conducted in samples of European genetic ancestries with controls for sex, year of birth, their interaction, and genetic principal components and applied a uniform set of quality-control procedures (see **Supplementary Note** for a comprehensive description)."

It seems more reasonable to calculate male and female specific PGIs by using male and female specific summary statistics.

We acknowledge that our originally submitted manuscript should have given some justification for our procedures.

Given our aims, we think there are two compelling arguments—one empirical and one theoretical—for our approach of relying on pooled summary statistics.

The *empirical* argument is that we have compelling evidence from prior work that the genetic correlation of EA across men and women is essentially one. We conducted sex-stratified association analyses in first two iterations of our EA GWAS (Rietveld *et al.*, 2013; Okbay *et al.*, 2016). In the Okbay *et al.* data, we estimated that the genetic correlation in EA across men and women is 0.98 (S.E. = 0.029), which is very close to, and not statistically distinguishable from, unity. We subsequently abandoned sex-stratified analyses since it was clear that any differences, if they exist, are tiny. Any degradation in predictive power due to imperfect genetic correlation between

males and females would be negligible in our setting and dwarfed by the degradation in predictive power one gets from halving the sample size of the underlying GWAS (which is effectively what happens when constructing sex-stratified PGIs).

The *theoretical* argument is that in Fisher's classical biometrical-genetic model, the genetic correlation between males and females is unity by assumption. Given our goal of testing a prediction that comes out of a model that assumes perfect genetic correlation, it is appropriate to construct PGIs that are aligned with the underlying theory.

In the Online Methods of the revised manuscript, we explain that we do not have summary statistics from analyses run separately in males and females and explain why:

We did not run sex-stratified analyses for the autosomal meta-analysis because there is compelling evidence from our prior work that the male-female genetic correlation for *EduYears* is close to one. For example, the Okbay *et al.* (2016) data yields an estimate of 0.98 (S.E. = 0.029).

Then, the correlation between EA and father's PGI can be less biased than that using father's PGI calculated using the summary statistics of males and females combined. This may be another reason of underestimating predicted spousal PGI correlation. The conclusion that "the spousal PGI correlation is far too strong to be consistent with assortative mating purely on phenotype" may need additional analysis.

Our understanding of the referee's concern is as follows: by calculating the father's PGI and mother's PGI using the summary statistics of males and females combined, the mate-pair PGI correlation is biased relative to what it would be if the PGIs were calculated using sex-specific summary statistics. If our understanding is correct, then we expect that the direction of bias would be toward zero, since both the father's PGI and mother's PGI that we use are measured with error

relative to the correct PGIs. Assuming we have not misunderstood the referee's concern, we therefore think this bias would go in the wrong direction for explaining our finding.

We apologize for any confusion caused and hope the responses to the items above ("**The PGI was calculated...**" and "**It seems more reasonable...**") do a better job explaining our analyses and justifying the conclusions we draw from them. The bottom lines are that our analyses controlled for sex and year of birth and that the male-female genetic correlation is close to one. Therefore, the bias from using the summary statistics of males and females combined will be negligible.

5) Page 10, line 348. What does gene-environment correlation mean? Does it suggest gene-environment interaction or something else?

In the revised manuscript, we now define this term when we first use it: "The population effect captures the sum of the direct effect, indirect effects from relatives (e.g., genetic influences on parents' education, socioeconomic status, and behavior), other gene-environment correlation (i.e., correlation between genotypes and environmental exposure, with population stratification being one possible cause), and a contribution from the genetic component of the phenotype that would be uncorrelated with the PGI under random mating but becomes correlated with the PGI due to the linkage disequilibrium between causal alleles induced by assortative mating (Supplementary Note) (Kong *et al.*, 2018; Howe *et al.*, 2021).

6) Reference 2 and 8 are the identical.

We thank the referee for catching this error, which is now fixed.

References

- Abdellaoui, A. *et al.* (2019) 'Genetic correlates of social stratification in Great Britain', *Nature Human Behaviour*, 3(12), pp. 1332–1342. doi: 10.1038/s41562-019-0757-5.
- Avinun, R. (2019) 'Educational Attainment Polygenic Score is Associated with Depressive Symptoms via Socioeconomic Status: A Gene-Environment-Trait Correlation', *bioRxiv*, p. 727552. doi: 10.1101/727552.
- Barth, D., Papageorge, N. W. and Thom, K. (2020) 'Genetic Endowments and Wealth Inequality', *Journal of Political Economy*, 128(4), pp. 1474–1522. doi: 10.1086/705415.
- Belsky, D. W. *et al.* (2016) 'The Genetics of Success: How Single-Nucleotide Polymorphisms Associated With Educational Attainment Relate to Life-Course Development', *Psychological Science*, 27(7), pp. 957–972. doi: 10.1177/0956797616643070.
- Belsky, D. W. *et al.* (2018) 'Genetic analysis of social-class mobility in five longitudinal studies.', *Proceedings of the National Academy of Sciences of the United States of America*, 115(31), pp. E7275–E7284. doi: 10.1073/pnas.1801238115.
- Bolyard, A. and Savelyev, P. A. (2021) *Understanding the Education Polygenic Score and Its Interactions with SES in Determining Health in Young Adulthood*, SSRN. doi: 10.2139/ssrn.3397735.
- Bonett, D. G. and Wright, T. A. (2000) 'Sample size requirements for estimating pearson, kendall and spearman correlations', *Psychometrika*, 65(1), pp. 23–28. doi: 10.1007/BF02294183.
- Breinholt, A. and Conley, D. (2020) *Child-Driven Parenting: Differential Early Childhood Investment by Offspring Genotype*, National Bureau of Economic Research Working Paper Series. Cambridge, MA. doi: 10.3386/w28217.
- Bulik-Sullivan, B. K. *et al.* (2015) 'LD Score regression distinguishes confounding from polygenicity in genome-wide association studies.', *Nature Genetics*, 47(3), pp. 291–295. doi: 10.1038/ng.3211.
- Cahoy, J. D. *et al.* (2008) 'A Transcriptome Database for Astrocytes, Neurons, and

Oligodendrocytes: A New Resource for Understanding Brain Development and Function', *Journal of Neuroscience*, 28(1), pp. 264–278. doi: 10.1523/JNEUROSCI.4178-07.2008.

Comes, A. L. *et al.* (2019) 'The genetic relationship between educational attainment and cognitive performance in major psychiatric disorders', *Translational Psychiatry*, 9(1), p. 210. doi: 10.1038/s41398-019-0547-x.

Ding, X. *et al.* (2019) 'The relationship between cognitive decline and a genetic predictor of educational attainment', *Social Science & Medicine*, 239, p. 112549. doi: 10.1016/j.socscimed.2019.112549.

Ding, X., Barban, N. and Mills, M. C. (2019) 'Educational attainment and allostatic load in later life: Evidence using genetic markers', *Preventive Medicine*, 129, p. 105866. doi: <https://doi.org/10.1016/j.ypmed.2019.105866>.

Duncan, L. *et al.* (2019) 'Analysis of polygenic risk score usage and performance in diverse human populations', *Nature Communications*, 10(1), pp. 1–9. doi: 10.1038/s41467-019-11112-0.

Ekbom, A. (2011) 'The Swedish Multi-generation Register', in Dillner, J. (ed.) *Methods in Biobanking*. Totowa, NJ: Humana Press. doi: 10.1007/978-1-59745-423-0_10.

Elliott, M. L. *et al.* (2019) 'A Polygenic Score for Higher Educational Attainment is Associated with Larger Brains', *Cerebral Cortex*, 29(8), pp. 3496–3504. doi: 10.1093/cercor/bhy219.

Finucane, H. K. *et al.* (2018) 'Heritability enrichment of specifically expressed genes identifies disease-relevant tissues and cell types', *Nature Genetics*, 50(4), pp. 621–629. doi: 10.1038/s41588-018-0081-4.

Fletcher, J. *et al.* (2021) 'Interpreting Polygenic Score Effects in Sibling Analysis', *bioRxiv*. doi: 10.1101/2021.07.16.452740.

Hill, W. G., Goddard, M. E. and Visscher, P. M. (2008) 'Data and theory point to mainly additive genetic variance for complex traits.', *PLoS Genetics*. Edited by T. F. C. Mackay, 4(2), p. e1000008.

doi: 10.1371/journal.pgen.1000008.

Hivert, V. *et al.* (2021) 'Estimation of non-additive genetic variance in human complex traits from a large sample of unrelated individuals', *American Journal of Human Genetics*, 108(5), pp. 786–798. doi: 10.1016/J.AJHG.2021.02.014.

Howe, L. J. *et al.* (2021) 'Within-sibship GWAS improve estimates of direct genetic effects', *bioRxiv*, p. 2021.03.05.433935. doi: 10.1101/2021.03.05.433935.

Hugh-Jones, D. *et al.* (2016) 'Assortative mating on educational attainment leads to genetic spousal resemblance for polygenic scores', *Intelligence*, 59, pp. 103–108. doi: 10.1016/j.intell.2016.08.005.

Huibregtse, B. M. *et al.* (2021) 'Genes Related to Education Predict Frailty Among Older Adults in the United States', *The Journals of Gerontology: Series B*. Edited by A. Zajacova, 76(1), pp. 173–183. doi: 10.1093/geronb/gbz092.

Jansen, I. *et al.* (2018) 'Genetic meta-analysis identifies 10 novel loci and functional pathways for Alzheimer's disease risk', *bioRxiv*. doi: 10.1101/258533.

Judd, N. *et al.* (2020) 'Cognitive and brain development is independently influenced by socioeconomic status and polygenic scores for educational attainment', *Proceedings of the National Academy of Sciences*, 117(22), pp. 12411–12418. doi: 10.1073/pnas.2001228117.

Kong, A. *et al.* (2018) 'The nature of nurture: Effects of parental genotypes', *Science*, 359(6374), pp. 424–428. doi: 10.1126/science.aan6877.

Kong, A., Benonisdottir, S. and Young, A. I. (2020) *Family Analysis with Mendelian Imputations*, *BioRxiv*. doi: 10.1101/2020.07.02.185181.

Krapohl, E. *et al.* (2016) 'Phenome-wide analysis of genome-wide polygenic scores', *Molecular Psychiatry*, 21(9), pp. 1188–1193. doi: 10.1038/mp.2015.126.

Laidley, T., Vinneau, J. and Boardman, J. D. (2019) 'Individual and Social Genomic Contributions to

Educational and Neighborhood Attainments: Geography, Selection, and Stratification in the United States', *Sociological Science*, 6(22), pp. 580–608. doi: 10.15195/v6.a22.

Lee, J. J. *et al.* (2018) 'Gene discovery and polygenic prediction from a genome-wide association study of educational attainment in 1.1 million individuals', *Nature Genetics*, 50(8), pp. 1112–1121. doi: 10.1038/s41588-018-0147-3.

Li, Y. *et al.* (2021) 'Achieved educational attainment, inherited genetic endowment for education, and obesity', *Biodemography and Social Biology*, 66(2), pp. 132–144. doi: 10.1080/19485565.2020.1869919.

Liu, H. *et al.* (2021) 'Adolescent Criminal Justice Involvement, Educational Attainment, and Genetic Inheritance: Testing an Integrative Model Using the Add Health Data', *Journal of Developmental and Life-Course Criminology*, 7(2), pp. 195–228. doi: 10.1007/s40865-021-00166-8.

Loh, P.-R. *et al.* (2018) 'Mixed-model association for biobank-scale datasets', *Nature Genetics*, 50(7), pp. 906–908. doi: 10.1038/s41588-018-0144-6.

Lynch, M. and Walsh, B. (1998) *Genetics and Analysis of Quantitative Traits*. 1st edn, *Genetics and Analysis of Quantitative Traits*. 1st edn. Sunderland, MA, MA: Sinauer Associates, Inc. Available at: http://www.invemar.org.co/redcostera1/invemar/docs/RinconLiterario/2011/febrero/AG_8.pdf %5Cnhttps://books.google.co.uk/books/about/Genetics_and_Analysis_of_Quantitative_Tr.html?id=UhCCQgAACAAJ&pgis=1 (Accessed: 29 August 2017).

Marioni, R. E. *et al.* (2016) 'Genetic variants linked to education predict longevity', *Proceedings of the National Academy of Sciences*, 113(47), pp. 13366–13371. doi: 10.1073/pnas.1605334113.

Martin, A. R. *et al.* (2019) 'Clinical use of current polygenic risk scores may exacerbate health disparities', *Nature Genetics*, 51(4), pp. 584–591. doi: 10.1038/s41588-019-0379-x.

Mitchell, B. L. *et al.* (2020) 'Educational attainment polygenic scores are associated with cortical total surface area and regions important for language and memory', *NeuroImage*, 212, p. 116691. doi: 10.1016/j.neuroimage.2020.116691.

Möttus, R. *et al.* (2017) 'Educational Attainment and Personality Are Genetically Intertwined', *Psychological Science*, 28(11), pp. 1631–1639. doi: 10.1177/0956797617719083.

Niemi, M. E. K. *et al.* (2018) 'Common genetic variants contribute to risk of rare severe neurodevelopmental disorders', *Nature*, 562(7726), pp. 268–271. doi: 10.1038/s41586-018-0566-4.

Okbay, A. *et al.* (2016) 'Genome-wide association study identifies 74 loci associated with educational attainment', *Nature*, 533(7604), pp. 539–542. doi: 10.1038/nature17671.

Papageorge, N. W. and Thom, K. (2020) 'Genes, Education, and Labor Market Outcomes: Evidence from the Health and Retirement Study', *Journal of the European Economic Association*, 18(3), pp. 1351–1399. doi: 10.1093/jeea/jvz072.

Reynolds, C. A., Baker, L. A. and Pedersen, N. L. (2000) 'Multivariate Models of Mixed Assortment: Phenotypic Assortment and Social Homogamy for Education and Fluid Ability', *Behavior Genetics*, 30(6), pp. 455–476. doi: 10.1023/A:1010250818089.

Rietveld, C. A. *et al.* (2013) 'GWAS of 126,559 individuals identifies genetic variants associated with educational attainment', *Science*, 340(6139), pp. 1467–1471. doi: 10.1126/science.1235488.

Salvatore, J. E. *et al.* (2020) 'Sibling comparisons elucidate the associations between educational attainment polygenic scores and alcohol, nicotine and cannabis', *Addiction*, 115(2), pp. 337–346. doi: 10.1111/add.14815.

Selzam, S. *et al.* (2019) 'Comparing within-and between-family polygenic score prediction', *The American Journal of Human Genetics*, 105(2), pp. 351–363.

Smith-Woolley, E., Selzam, S. and Plomin, R. (2019) 'Polygenic score for educational attainment captures DNA variants shared between personality traits and educational achievement.', *Journal of Personality and Social Psychology*, 117(6), pp. 1145–1163. doi: 10.1037/pspp0000241.

Stephoe, A. *et al.* (2013) 'Cohort Profile: The English Longitudinal Study of Ageing', *International*

Journal of Epidemiology, 42(6), pp. 1640–1648. doi: 10.1093/ije/dys168.

Trejo, S. and Domingue, B. W. (2018) ‘Genetic nature or genetic nurture? Introducing social genetic parameters to quantify bias in polygenic score analyses’, *Biodemography and Social Biology*, 64(3–4), pp. 187–215. doi: 10.1080/19485565.2019.1681257.

Verhoef, E. *et al.* (2019) ‘Disentangling polygenic associations between attention-deficit/hyperactivity disorder, educational attainment, literacy and language’, *Translational Psychiatry*, 9(1), p. 35. doi: 10.1038/s41398-018-0324-2.

de Vlaming, R. *et al.* (2017) ‘Meta-GWAS Accuracy and Power (MetaGAP) Calculator Shows that Hiding Heritability Is Partially Due to Imperfect Genetic Correlations across Studies’, *PLOS Genetics*. Edited by J. Marchini, 13(1), p. e1006495. doi: 10.1101/048322.

Walsh, B. and Lynch, M. (2018) ‘Associative Effects: Competition, Social Interactions, Group and Kin Selection’, in *Evolution and Selection of Quantitative Traits*. Oxford: Oxford University Press. Available at: <https://oxford.universitypressscholarship.com/view/10.1093/oso/9780198830870.001.0001/oso-9780198830870>.

Wang, Y. *et al.* (2020) ‘Theoretical and empirical quantification of the accuracy of polygenic scores in ancestry divergent populations’, *Nature Communications*, 11(1), pp. 1–9. doi: 10.1038/s41467-020-17719-y.

Warrier, V. *et al.* (2020) ‘Polygenic scores for intelligence, educational attainment and schizophrenia are differentially associated with core autism features, IQ, and adaptive behaviour in autistic individuals’, *medRxiv*, p. 2020.07.21.20159228. doi: 10.1101/2020.07.21.20159228.

Wertz, J. *et al.* (2018) ‘Genetics and Crime: Integrating New Genomic Discoveries Into Psychological Research About Antisocial Behavior’, *Psychological Science*, 29(5), pp. 791–803. doi: 10.1177/0956797617744542.

Wertz, J. *et al.* (2020) ‘Using DNA From Mothers and Children to Study Parental Investment in

Children's Educational Attainment', *Child Development*, 91(5), pp. 1745–1761. doi: 10.1111/cdev.13329.

Willoughby, E. A. *et al.* (2019) 'The role of parental genotype in predicting offspring years of education: evidence for genetic nurture', *Molecular Psychiatry*. doi: 10.1038/s41380-019-0494-1.

Yengo, L. *et al.* (2018) 'Meta-analysis of genome-wide association studies for height and body mass index in ~700000 individuals of European ancestry.', *Human molecular genetics*, 27(20), pp. 3641–3649. doi: 10.1093/hmg/ddy271.

Young, A. I. *et al.* (2018) 'Relatedness disequilibrium regression estimates heritability without environmental bias', *Nature Genetics*, 50(9), pp. 1304–1310. doi: 10.1038/s41588-018-0178-9.

Young, A. I. *et al.* (2020) *Mendelian imputation of parental genotypes for genome-wide estimation of direct and indirect genetic effects*, *BioRxiv*. doi: 10.1101/2020.07.02.185199.

Zaidi, A. A. and Mathieson, I. (2020) 'Demographic history mediates the effect of stratification on polygenic scores', *eLife*, 9, p. e61548. doi: 10.7554/eLife.61548.

de Zeeuw, E. L. *et al.* (2014) 'Polygenic scores associated with educational attainment in adults predict educational achievement and attention problems in children', *American Journal of Medical Genetics: Neuropsychiatric Genetics*, 165(6), pp. 510–520.

Zeng, L. *et al.* (2019) 'Genetically modulated educational attainment and coronary disease risk', *European Heart Journal*, 40(29), pp. 2413–2420. doi: 10.1093/eurheartj/ehz328.

Decision Letter, first revision:

Our ref: NG-A57578R

4th Nov 2021

Dear Dr. Benjamin,

Thank you for submitting your revised manuscript "Polygenic prediction within and between families from a 3-million-person GWAS of educational attainment" (NG-A57578R). It has now been seen by the original referees and their comments are below. The reviewers find that the paper has improved in revision, and therefore we'll be happy in principle to publish it in Nature Genetics, pending minor revisions to comply with our editorial and formatting guidelines.

Sincerely,

Wei

Wei Li, PhD
Senior Editor
Nature Genetics
New York, NY 10004, USA
www.nature.com/ng

Reviewer #1 (Remarks to the Author):

The authors have addressed my comments

Reviewer #2 (Remarks to the Author):

This dense paper is useful for beginning to understand the genetics of educational attainment (EA) and its influence on a raft of other traits, some of which might appear as unexpected a-priori. In our original review we stated "there is sufficient novelty, especially in their analysis of dominance – where

the results are rather remarkable, but also in the prediction of disease risk, direct vs indirect effects, and assortment to warrant publication in NG". Their thoughtful revision provides further support for our prior opinion. The authors have done an excellent job of responding to our concerns and we recommend publication. This review was done jointly by Kenneth S. Kendler MD and Silviu-Alin Bacanu PhD.

Reviewer #3 (Remarks to the Author):

I appreciate the authors' responses. I am satisfied with authors' responses.

Almost 2/3 of samples included in this study were obtained from 23andMe. However, their summary statistics are not available except by contacting 23andMe. Current GWAS summary statistics are either publicly available or can be obtained through dbGaP etc. The study can make further important contribution to the research community if the summary statistics are more accessible.

Author Rebuttal, first revision:

Response to Reviewer #3:

Remarks to the Author:

Almost 2/3 of samples included in this study were obtained from 23andMe. However, their summary statistics are not available except by contacting 23andMe. Current GWAS summary statistics are either publicly available or can be obtained through dbGaP etc. The study can make further important contribution to the research community if the summary statistics are more accessible.

We agree entirely that it is imperative that researchers do everything in their power to make the 23andMe summary statistics accessible to other researchers.

As we noted in our correspondence with the editor, the principal difficulty we face, along with all other researchers who work with 23andMe data, is that 23andMe's Data Agreement prohibits researchers from posting summary statistics for more than 10,000 SNPs per published paper (the restriction does not apply to any analyses that exclude 23andMe subjects altogether). The ostensible purpose of this rule is to safeguard the privacy of 23andMe study subjects. Since we share the referee's sentiment that 10K is overly conservative, we have raised the issue with our collaborators at 23andMe on a few occasions, but the answer has always been that the restriction is non-negotiable and a requirement from their legal department. Like all other published papers that use 23andMe data, dozens of which have appeared in *Nature Genetics*, we therefore try to our best to make the data accessible, subject to this constraint.

In practice, this boils down to:

1. Sharing full SNP-level summary statistics from all major GWAS meta-analyses with 23andMe excluded (but all other cohorts included).

2. Sharing summary statistics for as many SNPs as we can (10K) from the paper's major analyses that include 23andMe study subjects.
3. Explaining 23andMe's data access procedures to researchers who wish to access the complete, quality-controlled 23andMe summary statistics. Researchers who wish to access full summary statistics from any analyses that includes 23andMe can do so by (i) applying for permission from 23andMe to access the data (ii) meta-analyzing the 23andMe summary statistics with the summary statistics from (1) that we will make public.

The current draft of the paper lays out the issues in the Data Availability Statement:

...We provide association results for all SNPs that passed quality-control filters in autosomal, X chromosome, and dominance GWAS meta-analyses that excludes the research participants from 23andMe. SNP-level summary statistics from analyses based entirely or in part on 23andMe data can only be reported for up to 10,000 SNPs. For the complete dominance GWAS meta-analysis, which includes 23andMe, clumped results for the 1,000 SNPs with the smallest P values are provided. For the complete autosomal and X chromosome GWAS meta-analyses, respectively, clumped results for the 8,618 and 141 SNPs with $P < 10^{-5}$ are provided; this P value threshold was chosen such that the total number of SNPs across the analyses that include data from 23andMe does not exceed 10,000. The full GWAS summary statistics from 23andMe will be made available through 23andMe to qualified researchers under an agreement with 23andMe that protects the privacy of the 23andMe participants. Please visit <https://research.23andme.com/collaborate/#dataset-access/> for more information and to apply to access the data.

The paragraph quoted above explains why some of the data cannot be made public -- namely, the restriction from 23andMe -- and it provides information about how to access the non-public data by providing a URL to the web portal that researchers must use to apply for 23andMe data access. We believe this is the best we can do given our constraints. Based on our correspondence with the editor, we hope that is acceptable.

Final Decision Letter:

In reply please quote: NG-A57578R1 Benjamin

20th Jan 2022

Dear Dr. Benjamin,

I am delighted to say that your manuscript "Polygenic prediction of educational attainment within and between families from genome-wide association analyses in 3 million individuals" has been accepted for publication in an upcoming issue of Nature Genetics.

Your paper will be published online after we receive your corrections and will appear in print in the next available issue. You can find out your date of online publication by contacting the Nature Press Office (press@nature.com) after sending your e-proof corrections. Now is the time to inform your Public Relations or Press Office about your paper, as they might be interested in promoting its publication. This will allow them time to prepare an accurate and satisfactory press release. Include your manuscript tracking number (NG-A57578R1) and the name of the journal, which they will need when they contact our Press Office.

Acceptance is conditional on the data in the manuscript not being published elsewhere, or announced in the print or electronic media, until the embargo/publication date. These restrictions are not intended to deter you from presenting your data at academic meetings and conferences, but any

enquiries from the media about papers not yet scheduled for publication should be referred to us.

Please note that *Nature Genetics* is a Transformative Journal (TJ). Authors may publish their research with us through the traditional subscription access route or make their paper immediately open access through payment of an article-processing charge (APC). Authors will not be required to make a final decision about access to their article until it has been accepted. [Find out more about Transformative Journals](https://www.springernature.com/gp/open-research/transformative-journals)

Authors may need to take specific actions to achieve compliance with funder and institutional open access mandates. For submissions from January 2021, if your research is supported by a funder that requires immediate open access (e.g. according to [Plan S principles](https://www.springernature.com/gp/open-research/plan-s-compliance)) then you should select the gold OA route, and we will direct you to the compliant route where possible. For authors selecting the subscription publication route our standard licensing terms will need to be accepted, including our [self-archiving policies](https://www.springernature.com/gp/open-research/policies/journal-policies). Those standard licensing terms will supersede any other terms that the author or any third party may assert apply to any version of the manuscript.

Please note that Nature Research offers an immediate open access option only for papers that were first submitted after 1 January, 2021.

If you have not already done so, we invite you to upload the step-by-step protocols used in this manuscript to the Protocols Exchange, part of our on-line web resource, natureprotocols.com. If you complete the upload by the time you receive your manuscript proofs, we can insert links in your article that lead directly to the protocol details. Your protocol will be made freely available upon publication of your paper. By participating in natureprotocols.com, you are enabling researchers to more readily reproduce or adapt the methodology you use. [Natureprotocols.com](http://natureprotocols.com) is fully searchable, providing your protocols and paper with increased utility and visibility. Please submit your protocol to <https://protocolexchange.researchsquare.com/>. After entering your nature.com username and password you will need to enter your manuscript number (NG-A57578R1). Further information can be found at <https://www.nature.com/nprot/>.

Sincerely,

Wei Li, PhD
Senior Editor
Nature Genetics
New York, NY 10004, USA
www.nature.com/ng